# Lost in Latent Space: An Empirical Study of Latent Diffusion Models for Physics Emulation

**François Rozet**[1,2,3]    **Ruben Ohana**[1,2]    **Michael McCabe**[1,4]
**Gilles Louppe**[3]    **François Lanusse**[1,2,6]    **Shirley Ho**[1,2,4,5]

[1]Polymathic AI    [2]Flatiron Institute    [3]University of Liège
[4]New York University    [5]Princeton University
[6]Université Paris-Saclay, Université Paris Cité, CEA, CNRS, AIM

## Abstract

The steep computational cost of diffusion models at inference hinders their use as fast physics emulators. In the context of image and video generation, this computational drawback has been addressed by generating in the latent space of an autoencoder instead of the pixel space. In this work, we investigate whether a similar strategy can be effectively applied to the emulation of dynamical systems and at what cost. We find that the accuracy of latent-space emulation is surprisingly robust to a wide range of compression rates (up to $1000\times$). We also show that diffusion-based emulators are consistently more accurate than non-generative counterparts and compensate for uncertainty in their predictions with greater diversity. Finally, we cover practical design choices, spanning from architectures to optimizers, that we found critical to train latent-space emulators.

## 1 Introduction

Numerical simulations of dynamical systems are at the core of many scientific and engineering disciplines. Solving partial differential equations (PDEs) that describe the dynamics of physical phenomena enables, among others, weather forecasts [1, 2], predictions of solar wind and flares [3–5], or control of plasma in fusion reactors [6, 7]. These simulations typically operate on fine-grained spatial and temporal grids and require significant computational resources for high-fidelity results.

To address this limitation, a promising strategy is to develop neural network-based emulators to make predictions orders of magnitude faster than traditional numerical solvers. The typical approach [8–17] is to consider the dynamics as a function $f(x^i) = x^{i+1}$ that evolves the state $x^i$ of the system and to train a neural network $f_\phi(x)$ to approximate that function. In the context of PDEs, this network is sometimes called a neural solver [11, 18, 19]. After training, the autoregressive application of the solver, or rollout, emulates the dynamics. However, recent studies [11, 18–21] reveal that, while neural solvers demonstrate impressive accuracy for short-term prediction, errors accumulate over the course of the rollout, leading to distribution shifts between training and inference. This phenomenon is even more severe for stochastic or undetermined systems, where it is not possible to predict the next state given the previous one(s) with certainty. Instead of modeling the uncertainty, neural solvers produce a single point estimate, usually the mean, instead of a distribution.

The natural choice to alleviate these issues are generative models, in particular diffusion models, which have shown remarkable results in recent years. Following their success, diffusion models have been applied to emulation tasks [18, 19, 22–25] for which they were found to mitigate the rollout instability of non-generative emulators. However, diffusion models are much more expensive than deterministic alternatives at inference, due to their iterative sampling process, which defeats the purpose of using an emulator. To address this computational drawback, many works in the image and

video generation literature [26–32] consider generating in the latent space of an autoencoder. This approach has been adapted with success to the problem of emulating dynamical systems [33–37], sometimes even outperforming pixel-space emulation. In this work, we seek to answer a simple question: *What is the impact of latent-space compression on emulation accuracy?* To this end, we train and systematically evaluate latent-space emulators across a wide range of compression rates for challenging dynamical systems from TheWell [38]. Our results indicate that

i. Latent diffusion-based emulation is surprisingly robust to the compression rate, even when autoencoder reconstruction quality greatly degrades.

ii. Latent-space emulators match or exceed the accuracy of pixel-space emulators, while using fewer parameters and less training compute.

iii. Diffusion-based emulators consistently outperform their non-generative counterparts in both accuracy and plausibility of the emulated dynamics.

Finally, we dedicate part of this manuscript to design choices. We discuss architectural and modeling decisions for autoencoders and diffusion models that enable stable training of latent-space emulators under high compression. To encourage further research in this direction, we provide the code for all experiments at https://github.com/polymathicai/lola along with pre-trained model weights.

## 2 Diffusion models

The primary purpose of diffusion models (DMs) [39, 40], also known as score-based generative models [41, 42], is to generate plausible data from a distribution $p(x)$ of interest. Formally, continuous-time diffusion models define a series of increasingly noisy distributions

$$p(x_t) = \int p(x_t \mid x)\, p(x)\, \mathrm{d}x = \int \mathcal{N}(x_t \mid \alpha_t\, x, \sigma_t^2 I)\, p(x)\, \mathrm{d}x \tag{1}$$

such that the ratio $\alpha_t/\sigma_t \in \mathbb{R}_+$ is monotonically decreasing with the time $t \in [0, 1]$. For such a series, there exists a family of reverse-time stochastic differential equations (SDEs) [42–44]

$$\mathrm{d}x_t = \left[ f_t\, x_t - \frac{1 + \eta^2}{2} g_t^2\, \nabla_{x_t} \log p(x_t) \right] \mathrm{d}t + \eta\, g_t\, \mathrm{d}w_t \tag{2}$$

where $\eta \geq 0$ is a parameter controlling stochasticity, the coefficients $f_t$ and $g_t$ are derived from $\alpha_t$ and $\sigma_t$ [42–44], and for which the variable $x_t$ follows $p(x_t)$. In other words, we can draw noise samples $x_1 \sim p(x_1) \approx \mathcal{N}(0, \sigma_1^2 I)$ and obtain data samples $x_0 \sim p(x_0) \approx p(x)$ by solving Eq. (2) from $t = 1$ to 0. For high-dimensional samples, the terminal signal-to-noise ratio $\alpha_1/\sigma_1$ should be at or very close to zero [45]. In this work, we adopt the rectified flow [28, 46, 47] noise schedule, for which $\alpha_t = 1 - t$ and $\sigma_t = t$.

**Denoising score matching**    In practice, the score function $\nabla_{x_t} \log p(x_t)$ in Eq. (2) is unknown, but can be approximated by a neural network trained via denoising score matching [48, 49]. Several equivalent parameterizations and objectives have been proposed for this task [40–42, 47, 50, 51]. In this work, we adopt the denoiser parameterization $d_\phi(x_t, t)$ and its objective [51]

$$\arg\min_\phi \mathbb{E}_{p(x)p(t)p(x_t \mid x)} \left[ \lambda_t \left\| d_\phi(x_t, t) - x \right\|_2^2 \right] , \tag{3}$$

for which the optimal denoiser is the mean $\mathbb{E}[x \mid x_t]$ of $p(x \mid x_t)$. Importantly, $\mathbb{E}[x \mid x_t]$ is linked to the score function through Tweedie's formula [52–55]

$$\mathbb{E}[x \mid x_t] = \frac{x_t + \sigma_t^2 \nabla_{x_t} \log p(x_t)}{\alpha_t} , \tag{4}$$

which allows to use $s_\phi(x_t) = \sigma_t^{-2}(d_\phi(x_t, t) - \alpha_t\, x_t)$ as a score estimate in Eq. (2).

## 3 Methodology

In this section, we detail and motivate our experimental methodology for investigating the impact of compression on the accuracy of latent-space emulators. To summarize, we consider three challenging

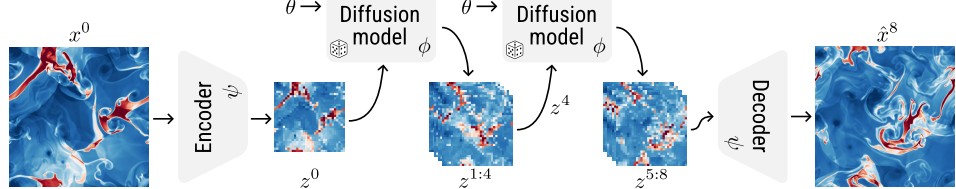

Figure 1. Illustration of the latent-space emulation process. At each step of the autoregressive rollout, the diffusion model generates the next $n = 4$ latent states $z^{i+1:i+n}$ given the current state $z^i$ and the simulation parameters $\theta$. After rollout, the generated latent states are decoded to pixel space.

datasets from TheWell [38]. For each dataset, we first train a series of autoencoders with varying compression rates. These autoencoders learn to map high-dimensional physical states $x^i \in \mathbb{R}^{H \times W \times C_{\text{pixel}}}$ to low-dimensional latent representations $z^i \in \mathbb{R}^{\frac{H}{r} \times \frac{W}{r} \times C_{\text{latent}}}$. Subsequently, for each autoencoder, we train two emulators operating in the latent space: a diffusion model (generative) and a neural solver (non-generative). Both are trained to predict the next $n$ latent states $z^{i+1:i+n}$ given the current latent state $z^i$ and simulation parameters $\theta$. This technique, known as temporal bundling [11], mitigates the accumulation of errors during rollout by decreasing the number of required autoregressive steps. After training, latent-space emulators are used to produce autoregressive rollouts $z^{1:L}$ starting from known initial state $z^0 = E_\psi(x^0)$ and simulation parameters $\theta$, which are then decoded to the pixel space as $\hat{x}^i = D_\psi(z^i)$.

## 3.1 Datasets

To study the effects of extreme compression rates, the datasets we consider should be high-dimensional and contain large amounts of data. Intuitively, the effective size of the dataset decreases in latent space, making overfitting more likely at fixed model capacity. According to these criteria, we select three datasets from TheWell [38]. Additional details are provided in Appendix B.

**Euler Multi-Quadrants**  The Euler equations model the behavior of compressible non-viscous fluids. In this dataset, the initial state presents multiple discontinuities which result in interacting shock waves as the system evolves for 100 steps. The 2d state of the system is represented with three scalar fields (energy, density, pressure) and one vector field (momentum) discretized on a $512 \times 512$ grid, for a total of $C_{\text{pixel}} = 5$ channels. Each simulation has either periodic or open boundary conditions and a different heat capacity $\gamma$, which constitutes their parameters $\theta$. We set a time stride $\Delta = 4$ between consecutive states $x^i$ and $x^{i+1}$, such that the simulation time $\tau = i \times \Delta$.

**Rayleigh-Bénard (RB)**  The Rayleigh-Bénard convection phenomenon occurs when an horizontal layer of fluid is heated from below and cooled from above. Over the 200 simulation steps, the temperature difference leads to the formation of convection currents where cooler fluid sinks and warmer fluid rises. The 2d state of the system is represented with two scalar fields (buoyancy, pressure) and one vector field (velocity) discretized on a $512 \times 128$ grid, for a total of $C_{\text{pixel}} = 4$ channels. Each simulation has different Rayleigh and Prandtl numbers as parameters $\theta$. We set a time stride $\Delta = 1$.

**Turbulence Gravity Cooling (TGC)**  The interstellar medium can be modeled as a turbulent fluid subject to gravity and radiative cooling. Starting from an homogeneous state, dense filaments form in the fluid, leading to the birth of stars. The 3d state of the system is represented with three scalar fields (density, pressure, temperature) and one vector field (velocity) discretized on a $64 \times 64 \times 64$ grid, for a total of $C_{\text{pixel}} = 6$ channels. Each simulation has different initial conditions function of their density, temperature, and metallicity. We set a time stride $\Delta = 1$.

## 3.2 Autoencoders

To isolate the effect of compression, we use a consistent autoencoder architecture and training setup across datasets and compression rates. We focus on compressing individual states $x^i$ into latent states $z^i = E_\psi(x^i)$, which are reconstructed as $\hat{x}^i = D_\psi(z^i)$.

**Architecture**  We adopt a convolution-based autoencoder architecture similar to the one used by Rombach et al. [26], which we adapt to perform well under high compression rates. Specifically, inspired by Chen et al. [31], we initialize the downsampling and upsampling layers near identity, which enables training deeper architectures with complex latent representations, while preserving reconstruction quality. For 2d datasets (Euler and RB), we set the spatial downsampling factor $r = 32$ for all autoencoders, meaning that a $32 \times 32$ patch in pixel space corresponds to one token in latent space. For 3d datasets (TGC), we set $r = 8$. The compression rate is then controlled solely by varying the number of channels per token in the latent representation. For instance, with the Euler dataset, an autoencoder with $C_{\text{latent}} = 64$ latent channels – f32c64 in the notations of Chen et al. [31] – transforms the input state with shape $512 \times 512 \times 5$ to a latent state with shape $16 \times 16 \times 64$, yielding a compression rate of 80. This setup ensures that the architectural capacity remains similar for all autoencoders and allows for fair comparison across compression rates. Further details as well as a short ablation study are provided in Appendix B.

**Training**  Latent diffusion models [26] often rely on a Kullback-Leibler (KL) divergence penalty to encourage latents to follow a standard Gaussian distribution. However, this term is typically down-weighted by several orders of magnitude to prevent severe reconstruction degradation. As such, the KL penalty acts more as a weak regularization than a proper variational objective [56] and post-hoc standardization of latents is often necessary. We replace this KL penalty with a deterministic saturating function

$$z \mapsto \frac{z}{\sqrt{1 + z^2/B^2}} \tag{5}$$

applied to the encoder's output. In our experiments, we choose the bound $B = 5$ to mimic the range of a standard Gaussian distribution. We find this approach simpler and more effective at structuring the latent space, without introducing a tradeoff between regularization and reconstruction quality. We additionally omit perceptual [57] and adversarial [58, 59] loss terms, as they are designed for natural images where human perception is the primary target, unlike physics. The training objective thus simplifies to a reconstruction loss

$$\arg \min_\psi \mathbb{E}_{p(x)} \left[ \ell(x, D_\psi(E_\psi(x))) \right] . \tag{6}$$

The loss $\ell$ is typically a variation of $L_1$ or $L_2$ regression, which we discuss in Appendix B. Finally, we find that preconditioned optimizers [60–62] greatly accelerate the training convergence of autoencoders compared to the widespread Adam [63] optimizer (see Table 4). We adopt the PSGD [60] implementation in the heavyball [64] library for its fewer number of tunable hyper-parameters and lower memory footprint than SOAP [62].

## 3.3  Diffusion models

We train diffusion models to predict the next $n$ latent states $z^{i+1:i+n}$ given the current state $z^i$ and simulation parameters $\theta$, that is to generate from $p(z^{i+1:i+n} \mid z^i, \theta)$. We parameterize our diffusion models with a denoiser $d_\phi(z_t^{i:i+n}, b, \theta, t)$ whose task is to denoise sequences of noisy states $z_t^i \sim p(z_t^i \mid z^i) = \mathcal{N}(z_t^i \mid \alpha_t z^i, \sigma_t^2 I)$ given the parameters $\theta$ of the simulation. Conditioning with respect to known elements in the sequence $z^{i:i+n}$ is tackled with a binary mask $b \in \{0,1\}^{n+1}$ concatenated to the input, as in MCVD [65]. For instance, $b = (1, 0, \ldots, 0)$ indicates that the first element $z^i$ is known, while $b = (1, \ldots, 1, 0)$ indicates that the first $n - 1$ elements $z^{i:i+n-1}$ are known. Known elements are provided to the denoiser without noise.

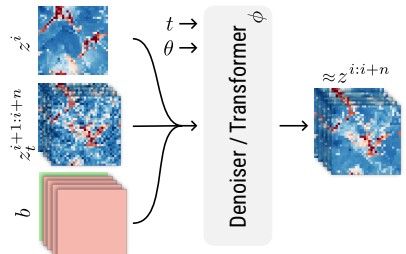

Figure 2. Illustration of the denoiser's inputs and outputs, while generating from $p(z^{i+1:i+n} \mid z^i, \theta)$.

**Architecture**  Drawing inspiration from recent successes in latent image generation [27–31], we use a transformer-based architecture for the denoiser. We incorporate several architectural refinements shown to improve performance and stability, including query-key normalization [66], rotary positional embedding (RoPE) [67, 68], and value residual learning [69]. The transformer operates on the spatial and temporal axes of the input $z_t^{i:i+n}$, while the parameters $\theta$ and diffusion time $t$ modulate the

transformer blocks. Thanks to the considerable ($r = 32$) spatial downsampling performed by the autoencoder, we are able to apply full spatio-temporal attention, avoiding the need for sparse attention patterns [70–72]. Finally, we fix the token embedding size (1024) and the number of transformer blocks (16) for all diffusion models. The only architectural variation stems from the number of input and output channels dictated by the corresponding autoencoder.

**Training**    As in Section 2, diffusion models are trained via denoising score matching [48, 49]

$$\arg\min_\phi \mathbb{E}_{p(\theta,z^{i:i+n},z_t^{i:i+n})p(b)} \left[ \left\| d_\phi(\underbrace{z^{i:i+n} \odot b}_{\text{clean}} + \underbrace{z_t^{i:i+n} \odot (1-b)}_{\text{noisy}}, b, \theta, t) - z^{i:i+n} \right\|_2^2 \right] \quad (7)$$

with the exception that the data does not come from the pixel-space distribution $p(\theta, x^{1:L})$ but from the latent-space distribution $p(\theta, z^{1:L})$ determined by the encoder $E_\psi$. Following Voleti et al. [65], we randomly sample the binary mask $b \sim p(b)$ during training to cover several conditioning tasks, including prediction with variable-length context $p(z^{i+c:i+n} \mid z^{i:i+c-1})$.

**Sampling**    After training, we sample from the learned distribution by solving Eq. (2) with $\eta = 0$, which corresponds to the probability flow ODE [42]. To this end, we implement a 3rd order Adams-Bashforth multi-step integration method, as proposed by Zhang et al. [73]. Intuitively, this method leverages information from previous integration steps to improve accuracy. We find this approach highly effective, producing high-quality samples with significantly fewer neural function evaluations (NFEs) than other widespread samplers [50, 51].

### 3.4  Neural solvers

We train neural solvers to perform the same task as diffusion models. Unlike the latter, however, solvers do not generate from $p(z^{i+1:i+n} \mid z^i, \theta)$, but produce a point estimate $f_\phi(z_i, \theta) \approx \mathbb{E}\left[z^{i+1:i+n} \mid z_i, \theta\right]$ instead. We also train a pixel-space neural solver, for which $z^i = x^i$, as baseline.

**Architecture**    For latent-space neural solvers, we use the same transformer-based architecture as for diffusion models. The only notable difference is that transformer blocks are only modulated with respect to the simulation parameters $\theta$. For the pixel-space neural solver, we keep the same architecture, but group the pixels into $16 \times 16$ patches, as in vision transformers [74]. We also double the token embedding size (2048) such that the pixel-space neural solver has roughly two times more trainable parameters than an autoencoder and latent-space emulator combined.

**Training**    Neural solvers are trained via mean regression

$$\arg\min_\phi \mathbb{E}_{p(\theta,z^{i:i+n})p(b)} \left[ \left\| f_\phi(z^{i:i+n} \odot b, b, \theta) - z^{i:i+n} \right\|_2^2 \right]. \quad (8)$$

Apart from the training objective, the training configuration (optimizer, learning rate schedule, batch size, epochs, masking, ...) for neural solvers is strictly the same as for diffusion models.

### 3.5  Evaluation metrics

We consider several metrics for evaluation, each serving a different purpose. We report these metrics either at a lead time $\tau = i \times \Delta$ or averaged over a lead time horizon $a : b$. If the states $x^i$ present several fields, the metric is first computed on each field separately, then averaged.

**Variance-normalized RMSE**    The root mean squared error (RMSE) and its normalized variants are widespread metrics to quantify the point-wise accuracy of an emulation [21, 38, 75]. Following Ohana et al. [38], we pick the variance-normalized RMSE (VRMSE) over the more common normalized RMSE (NRMSE), as the latter down-weights errors in non-negative fields such as pressure and density. Formally, for two spatial fields $u$ and $v$, the VRMSE is defined as

$$\text{VRMSE}(u, v) = \sqrt{\frac{\langle (u-v)^2 \rangle}{\langle (u - \langle u \rangle)^2 \rangle + \epsilon}} \quad (9)$$

where $\langle \cdot \rangle$ denotes the spatial mean operator and $\epsilon = 10^{-6}$ is a numerical stability term.

**Power spectrum RMSE** For chaotic systems such as turbulent fluids, it is typically intractable to achieve accurate long-term emulation as very small errors can lead to entirely different trajectories later on. In this case, instead of reproducing the exact trajectory, emulators should generate diverse trajectories that remain statistically plausible. Intuitively, even though structures are wrongly located, the types of patterns and their distribution should stay similar [76]. Following Ohana et al. [38], we assess statistical plausibility by comparing the power spectra of the ground-truth and emulated trajectories. For two spatial fields $u$ and $v$, we compute the isotropic power spectra $p_u$ and $p_v$ and split them into three frequency bands (low, mid and high) evenly distributed in log-space. We report the RMSE of the relative power spectra $p_v/p_u$ over each band.

**Spread-skill ratio** In earth sciences [25, 75], the skill of an ensemble of $K$ particles is defined as the RMSE of the ensemble mean. The spread is defined as the ensemble standard deviation. Under these definitions and the assumption of a perfect forecast where ensemble particles are exchangeable, Fortin et al. [75] show that

$$\text{Skill} \approx \sqrt{K+1/K} \ \text{Spread} . \tag{10}$$

This motivates the use of the (corrected) spread-skill ratio as a metric. Intuitively, if the ratio is smaller than one, the ensemble is biased or under-dispersed. If the ratio is larger than one, the ensemble is over-dispersed. It should be noted, however, that a spread-skill ratio of 1 is a necessary but insufficient condition for a perfect forecast.

## 4 Results

We start with the evaluation of the autoencoders. For all datasets, we train three autoencoders with respectively 64, 16, and 4 latent channels. These correspond to compression rates of 80, 320 and 1280 for the Euler dataset, 64, 256, and 1024 for the RB dataset, and 48, 192, 768 for the TGC dataset, respectively. In the following, we refer to models by their compression rate. Additional experimental details are provided in Section 3 and Appendix B.

For each autoencoder, we evaluate the reconstruction $\hat{x}^i = D_\psi(E_\psi(x^i))$ of all states $x^i$ in 64 test trajectories $x^{0:L}$. As expected, when the compression rate increases, the reconstruction quality degrades, as reported in Figure 3. For the Euler dataset, the reconstruction error grows with the lead time due to wavefront interactions and rising high-frequency content. For the RB dataset, the reconstruction error peaks mid-simulation during the transition from low to high-turbulence regime. Similar trends can be observed for the power spectrum RMSE in Tables 8, 9 and 10, where the high-frequency band is most affected by compression. These results so far align with what practitioners intuitively expect from lossy compression.

We now turn to the evaluation of the emulators. For each autoencoder, we train two latent-space emulators: a diffusion model and a neural solver. Starting from the initial state $z^0 = E_\psi(x^0)$ and simulation parameters $\theta$ of 64 test trajectories $x^{0:L}$, each emulator produces 16 distinct autoregressive rollouts $z^{1:L}$, which are then decoded to the pixel space as $\hat{x}^i = D_\psi(z^i)$. Note that for neural solvers, all 16 rollouts are identical. We compute the metrics of each prediction $\hat{x}^i$ against the ground-truth state $x^i$.

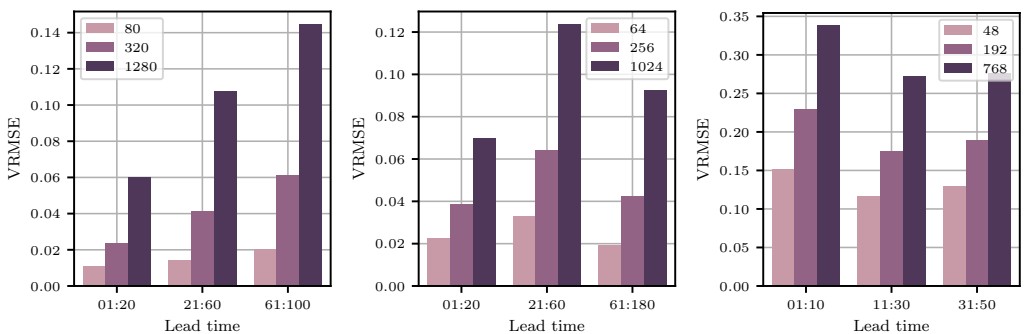

Figure 3. Average VRMSE of the autoencoder reconstruction at different compression rates and lead time horizons for the Euler (left), RB (center) and TGC (right) datasets. The compression rate has a clear impact on reconstruction quality.

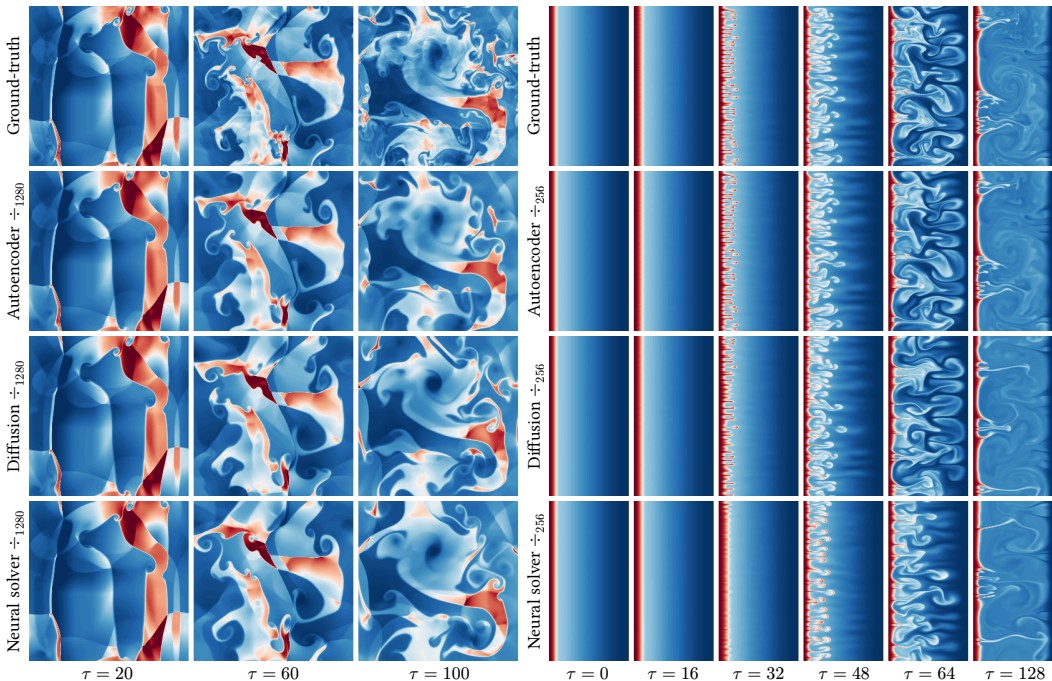

Figure 4. Examples of latent-space emulation for the Euler (left) and Rayleigh-Bénard (right) datasets. Even for large compression rates (÷), latent-space emulators are able to reproduce the dynamics surprisingly faithfully, despite significant reconstruction artifacts. For Euler, wavefronts are accurately propagated until the end of the simulation, while vortices are well located, but distorted. For Rayleigh-Bénard, diffusion-based emulators produce plumes that grow at the correct pace but diverge from the ground-truth. Similar observations can be made in Figures 10 to 21.

As expected from imperfect emulators, the emulation error grows with the lead time, as shown in Figures 5 and 8. However, the point-wise error of diffusion models, as measured by the VRMSE, does not grow (Euler, TGC) and sometimes decreases (RB) with higher compression rates. Even at extreme (> 1000) compression rates, latent-space emulators outperform the baseline pixel-space neural solver, despite the latter benefiting from more parameters and training compute. Similar observations can be made with the power spectrum RMSE over low and mid-frequency bands. High-frequency content, however, appears limited by the autoencoder's reconstruction capabilities. We confirm this hypothesis by recomputing the metrics relative to the auto-encoded state $D_\psi(E_\psi(x^i))$, which we report in Figure 9. This time, the power spectrum RMSE of the diffusion models is low for mid and high-frequency bands. These findings support a puzzling narrative: emulation accuracy exhibits strong resilience to latent-space compression, starkly contrasting with the clear degradation in reconstruction quality.

Our experiments also provide a direct comparison between generative (diffusion) and deterministic (neural solver) approaches to emulation within a latent space. Figures 8 and 9 indicate that diffusion-based emulators are consistently more accurate than their deterministic counterparts and generate trajectories that are statistically more plausible in terms of power spectrum. This can be observed qualitatively in Figure 4 or Figures 10 to 21 in Appendix C. In addition, the spread-skill ratio of diffusion models is close to 1, suggesting that the ensemble of trajectories they produce are reasonably well calibrated in terms of uncertainty. However, the ratio slightly decreases with the compression rate. This phenomenon is partially explained by the smoothing effect of $L_2$-driven compression, and is therefore less severe in Figure 9. Nonetheless, it remains present and could be a sign of overfitting due to the reduced amount of training data in latent space.

Table 1. Inference time per state for the Euler dataset, including generation and decoding.

| Method | Space | Time |
|---|---|---|
| simulator | pixel | $\mathcal{O}(10\,\text{s})$ |
| neural solver | pixel | $56\,\text{ms}$ |
| neural solver | latent | $13\,\text{ms}$ |
| diffusion | pixel | $\mathcal{O}(1\,\text{s})$ |
| diffusion | latent | $84\,\text{ms}$ |

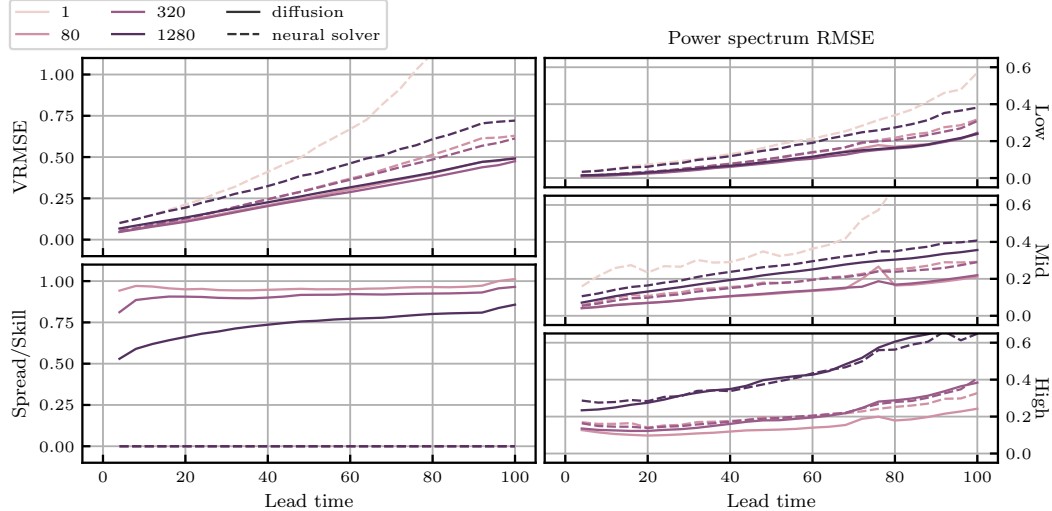

Figure 5. Average evaluation metrics of latent-space emulation for the Euler dataset. As expected from imperfect emulators, the emulation error grows with the lead time. However, the compression rate has little to no impact on diffusion-based emulation accuracy, beside high-frequency content. The spread-skill ratio [25, 75] drops slightly with the compression rate, which could be a sign of overfitting. Diffusion-based emulators are consistently more accurate than neural solvers.

In terms of computational cost, although they remain slower than latent-space neural solvers, latent-space diffusion models are much faster than their pixel-space counterparts and competitive with pixel-space neural solvers (see Table 1). With our latent diffusion models, generating and decoding a full (100 simulation steps, 7 autoregressive steps) Euler trajectory takes 3 seconds on a single A100 GPU, compared to roughly 1 CPU-hour with the original numerical simulation [38, 77].

A final advantage of diffusion models lies in their capacity to incorporate additional information during sampling via guidance methods [42, 78–81]. For example, if partial or noisy state observations are available, we can guide the emulation such that it remains consistent with these observations. We provide an illustrative example in Figure 6 where guidance is performed with the MMPS [78] method. Thanks to the additional information in the observations, the emulation diverges less from the ground-truth.

# 5   Related work

Data-driven emulation of dynamical systems has become a prominent research area [8–17] with diverse applications, including accelerating fluid simulations on uniform meshes using convolutional networks [8, 12], emulating various physics on non-uniform meshes with graph neural networks [9–11, 14], and solving partial differential equations with neural operators [13, 21, 82–84]. However, McCabe et al. [15] and Herde et al. [16] highlight the large data requirements of these methods and propose pre-training on multiple data-abundant physics before fine-tuning on data-scarce ones to improve data efficiency and generalization. Our experiments similarly suggest that large datasets are needed to train latent-space emulators.

A parallel line of work, related to reduced-order modeling [85], focuses on learning low-dimensional representations of high-dimensional system states. Within this latent space, dynamics can be emulated more efficiently [86–94]. Various embedding approaches have been explored: convolutional autoencoders for uniform meshes [88, 89], graph-based autoencoders for non-uniform meshes [90], and implicit neural representations for discretization-free states [34, 92]. Koopman operator theory [95] has also been integrated into autoencoder training to promote linear latent dynamics [91, 96]. Other approaches to enhance latent predictability include regularizing temporal derivatives [97], jointly optimizing the decoder and latent emulator [98], and self-supervised prediction [99]. While our work adopts this latent emulation paradigm, we do not impose structural biases on the latent space beside reconstruction quality.

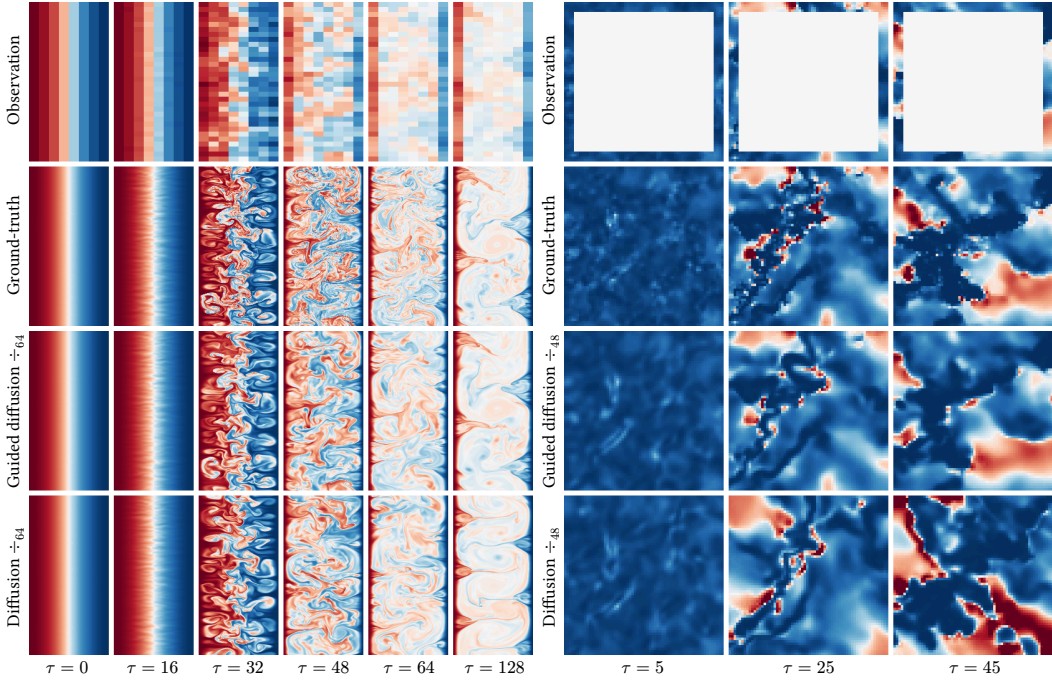

Figure 6. Example of guided latent-space emulation for the RB (left) and TGC (right) datasets. The observations are the states downsampled by a factor 16 for RB and a stripe along the domain boundaries for TGC. Guidance is performed using the MMPS [78] method. Thanks to the additional information in the observations, the emulation diverges less from the ground-truth.

A persistent challenge in neural emulation is ensuring temporal stability. Many models, while accurate for short-term prediction, exhibit long-term instabilities as errors accumulate, pushing the predictions out of the training data distribution [21]. Several strategies have been proposed to mitigate this issue: autoregressive unrolling during training [11, 86, 100], architectural modifications [21, 83], noise injection [12], and post-processing [18, 101]. Generative models, particularly diffusion models, have recently emerged as a promising approach to address this problem [18, 19, 22–25] as they produce statistically plausible states, even when they diverge from the ground-truth solution.

While more accurate and stable, diffusion models are computationally expensive at inference. Drawing inspiration from latent space generation in computer vision [26–32], recent studies have applied latent diffusion models to emulate dynamical systems: Gao et al. [33] address short-term precipitation forecasting, Zhou et al. [35] generate trajectories conditioned on text descriptions, Du et al. [34] generate trajectories within an implicit neural representation, and Li et al. [36] combine a state-wise autoencoder with a spatiotemporal diffusion transformer [27] for autoregressive emulation, similar to our approach. These studies report favorable or competitive results against pixel-space and deterministic baselines, consistent with our observations.

## 6  Discussion

Our results reveal key insights about latent physics emulation. First, diffusion-based emulation accuracy is surprisingly robust to latent-space compression, with performance remaining constant or even improving when autoencoder reconstruction quality significantly deteriorates. This observation is consistent with the latent generative modeling literature [26, 56], where compression serves a dual purpose: reducing dimensionality and filtering out perceptually irrelevant patterns that might distract from semantically meaningful information. Our experiments support this hypothesis as latent-space emulators outperform their pixel-space counterparts despite using fewer parameters and requiring less training compute. Yao et al. [102] similarly demonstrate that higher compression can sometimes improve generation quality despite degrading reconstruction. While our findings seem to violate the famous data processing inequality, they are well aligned with the theory of *usable* information [103], where a learned representation can hold more $\mathcal{V}$-information from the point

of view of a computationally constrained observer. Second, diffusion-based generative emulators consistently achieve higher ensemble accuracy than deterministic neural solvers while producing diverse, statistically plausible trajectories. This supports the idea that generative models mitigate distribution shift [18, 19, 22–25]. However, at the first prediction step, before distribution shift can take effect, diffusion models are already more accurate than deterministic neural solvers. This suggests an inherent modeling advantage, possibly lying in the iterative nature of diffusion sampling.

Despite the finite number of datasets, we believe that our findings are likely to generalize well across the broader spectrum of fluid dynamics. The Euler, RB and TGC datasets represent distinct fluid regimes that cover many key challenges in dynamical systems emulation: nonlinearities, multi-scale interactions, and complex spatio-temporal patterns. In addition, previous studies [33–36] come to similar conclusions for other fluid dynamics problems. However, we exercise caution about extending these conclusions beyond fluids. Systems governed by fundamentally different physics, such as chemical or quantum phenomena, may respond unpredictably to latent compression. Probing these boundaries represents an important direction for future research. Our empirical findings also prompt the need for theoretical explanations, which we leave to future work.

Apart from datasets, if compute resources were not a limiting factor, our study could be extended along several dimensions, although we anticipate that additional experiments would not fundamentally alter our conclusions. First, we could investigate techniques for improving the structure of the latent representation, such as incorporating Koopman-inspired losses [91, 96], regularizing temporal derivatives [97], or training shallow auxiliary decoders [102, 104]. Second, we could probe the behavior of different embedding strategies under high compression, including spatio-temporal embeddings [34, 35, 105], implicit neural representations [34, 92], and masked auto-encoders [104, 106]. Third, we could add the capability to trade speed for accuracy, analogous to running numerical solvers at finer resolutions, by training an auto-encoder with an adaptive latent dimensionality [107–109]. Forth, we could study the effects of autoencoder and emulator capacity by scaling either up or down their number of trainable parameters. Each of these directions represents a substantial computational investment, particularly given the scale of our datasets and models, but would help establish best practices for latent-space emulation.

Nevertheless, our findings lead to clear recommendations for practitioners wishing to implement physics emulators. First, try latent-space approaches before pixel-space emulation. The former offer reduced computational requirements, lower memory footprint, and comparable or better accuracy across a wide range of compression rates. Second, prefer diffusion-based emulators over deterministic neural solvers. Latent diffusion models provide more accurate, diverse and stable long-term trajectories, while narrowing the inference speed gap significantly.

Our experiments, however, reveal important considerations about dataset scale when training latent-space emulators. The decreasing spread-skill ratio observed at higher compression rates suggests potential overfitting. This makes intuitive sense: as compression increases, the effective size of the dataset in latent space decreases, making overfitting more likely at fixed model capacity. Benchmarking latent emulators on smaller (10-100 GB) datasets like those used by Kohl et al. [19] could therefore yield misleading results. In addition, because the latent space is designed to preserve pixel space content, observing overfitting in this compressed representation suggests that pixel-space models encounter similar issues that remain undetected. This points towards the need for large training datasets or mixtures of datasets used to pre-train emulators before fine-tuning on targeted physics, as advocated by McCabe et al. [15] and Herde et al. [16].

## Acknowledgments and Disclosure of Funding

We thank Géraud Krawezik and the Scientific Computing Core at the Flatiron Institute, a division of the Simons Foundation, for the compute facilities and support. We gratefully acknowledge use of the research computing resources of the Empire AI Consortium, Inc., with support from the State of New York, the Simons Foundation, and the Secunda Family Foundation. Polymathic AI acknowledges funding from the Simons Foundation and Schmidt Sciences, LLC. François Rozet is a research fellow of the F.R.S.-FNRS (Belgium) and acknowledges its financial support.

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

# A  Spread / Skill

The skill [25, 75] of an ensemble of $K$ particles $v_k$ is defined as the RMSE of the ensemble mean

$$\text{Skill} = \sqrt{\left\langle \left( u - \frac{1}{K} \sum_{k=1}^{K} v_k \right)^2 \right\rangle} \tag{11}$$

where $\langle \cdot \rangle$ denotes the spatial mean operator. The spread is defined as the ensemble standard deviation

$$\text{Spread} = \sqrt{\left\langle \frac{1}{K-1} \sum_{j=1}^{K} \left( v_j - \frac{1}{K} \sum_{k=1}^{K} v_k \right)^2 \right\rangle}. \tag{12}$$

Under these definitions and the assumption of a perfect forecast where ensemble particles are exchangeable, Fortin et al. [75] show that

$$\text{Skill} \approx \sqrt{\frac{K+1}{K}} \, \text{Spread} \, . \tag{13}$$

This motivates the use of the (corrected) spread-skill ratio as a metric. Intuitively, if the ratio is smaller than one, the ensemble is biased or under-dispersed. If the ratio is larger than one, the ensemble is over-dispersed. It should be noted however, that a spread-skill ratio of 1 is a necessary but not sufficient condition for a perfect forecast.

## B   Experiment details

**Datasets**   For all datasets, each field is standardized with respect to its mean and variance over the training set. For Euler, the non-negative scalar fields (energy, density, pressure) are transformed with $x \mapsto \log(x + 1)$ before standardization. For TGC, the non-negative scalar fields (density, pressure, temperature) are transformed with $x \mapsto \log(x + 10^{-6})$ before standardization. When the states are illustrated graphically, as in Figure 1, we represent the density field for Euler, the buoyancy field for RB, and a slice of the temperature field for TGC.

Table 2. Details of the selected datasets. We refer the reader to Ohana et al. [38] for more information.

|  | Euler Multi-Quadrants | Rayleigh-Bénard | Turbulence Gravity Cooling |
|---|---|---|---|
| Software | Clawpack [77] | Dedalus [110] | ASURA-FDPS [111] |
| Size | 5243 GB | 367 GB | 849 GB |
| Fields | energy, density, pressure, velocity | buoyancy, pressure, momentum | density, pressure, temperature, velocity |
| Channels $C_{\text{pixel}}$ | 5 | 4 | 6 |
| Resolution | $512 \times 512$ | $512 \times 128$ | $64 \times 64 \times 64$ |
| Discretization | Uniform | Chebyshev | Uniform |
| Trajectories | 10000 | 1750 | 2700 |
| Time steps $L$ | 100 | 200 | 50 |
| Stride $\Delta$ | 4 | 1 | 1 |
| $\theta$ | heat capacity $\gamma$, boundary conditions | Rayleigh number, Prandtl number | hydrogen density $\rho_0$, temperature $T_0$, metallicity $Z$ |

**Autoencoders**   The encoder $E_\psi$ and decoder $D_\psi$ are convolutional networks with residual blocks [112], SiLU [113] activation functions and layer normalization [114]. The output of the encoder is transformed with a saturating function (see Section 3). We provide a schematic illustration of the autoencoder architecture in Figure 7. Following McCabe et al. [15], we use a field-weighted loss $\ell$, and choose the variance-normalized MSE (VMSE)

$$\text{VMSE}(u, v) = \frac{\langle (u - v)^2 \rangle}{\langle (u - \langle u \rangle)^2 \rangle + \epsilon} \tag{14}$$

averaged over fields, where $\epsilon = 10^{-2}$ mitigates training instabilities. We train the encoder and decoder jointly for $1024 \times 256$ steps of the PSGD [60] optimizer. To mitigate overfitting we use random spatial axes permutations, flips and rolls as data augmentation. Each autoencoder takes 1 (RB), 2 (Euler) or 4 (TGC) days to train on 8 H100 GPUs. Other hyperparameters are provided in Table 3.

**Caching**   The entire dataset is encoded with each trained autoencoder and the resulting latent trajectories are cached permanently on disk. The latter can then be used to train latent-space emulators, without needing to load and encode high-dimensional samples on the fly. Depending on hardware and data dimensionality, this approach can make a huge difference in I/O efficiency.

**Emulators**   The denoiser $d_\phi$ and neural solver $f_\phi$ are transformers with query-key normalization [66], rotary positional embedding (RoPE) [67, 68], and value residual learning [69]. The 16 blocks are modulated by the simulation parameters $\theta$ and the diffusion time $t$, as in diffusion transformers [27]. We train the emulator for $4096 \times 64$ steps of the Adam [63] optimizer. Each latent-space emulator takes 2 (RB) or 5 (Euler, TGC) days to train on 8 H100 GPUs. Each pixel-space emulator takes 5 (RB) or 10 (Euler) days to train on 16 H100 GPUs. We do not train a pixel-space emulator for TGC. Other hyperparameters are provided in Table 5.

During training we randomly sample the binary mask $b$. The number of context elements $c$ follows a Poisson distribution $\text{Pois}(\lambda = 2)$ truncated between 1 and $n$. Hence, the masks $b$ take the form

$$b = (\underbrace{1, \ldots, 1}_{c}, 0, \ldots, 0) \tag{15}$$

implicitly defining a distribution $p(b)$.

Table 3. Hyperparameters for the autoencoders.

|  | Euler & RB | TGC |
|---|---|---|
| Architecture | Conv | Conv |
| Parameters | $3.1 \times 10^8$ | $7.2 \times 10^8$ |
| Pixel shape | $C_{\text{pixel}} \times H \times W$ | $C_{\text{pixel}} \times H \times W \times Z$ |
| Latent shape | $C_{\text{latent}} \times \frac{H}{32} \times \frac{W}{32}$ | $C_{\text{latent}} \times \frac{H}{8} \times \frac{W}{8} \times \frac{Z}{8}$ |
| Residual blocks per level | (3, 3, 3, 3, 3, 3) | (3, 3, 3, 3) |
| Channels per level | (64, 128, 256, 512, 768, 1024) | (64, 256, 512, 1024) |
| Kernel size | $3 \times 3$ | $3 \times 3 \times 3$ |
| Activation | SiLU | SiLU |
| Normalization | LayerNorm | LayerNorm |
| Dropout | 0.05 | 0.05 |
| Loss | VMSE | VMSE |
| Optimizer | PSGD | PSGD |
| Learning rate | $10^{-5}$ | $10^{-5}$ |
| Weight decay | 0.0 | 0.0 |
| Scheduler | cosine | cosine |
| Gradient norm clipping | 1.0 | 1.0 |
| Batch size | 64 | 64 |
| Steps per epoch | 256 | 256 |
| Epochs | 1024 | 1024 |
| GPUs | 8 | 8 |

Table 4. Short ablation study on the autoencoder architecture and training configurations. We pick the Rayleigh-Bénard dataset and an architecture with 64 latent channels to perform this study. The two major modifications that we propose are (1) the initialization of the downsampling and upsampling layers near identity, inspired by Chen et al. [31], and (2) the use of a preconditioned optimizer, PSGD [60], instead of Adam [63]. We report the mean absolute error (MAE) on the validation set during training. The combination of both proposed modifications leads to order(s) of magnitude faster convergence.

| Optimizer | Id. init | Epoch | | | Time |
|---|---|---|---|---|---|
|  |  | 10 | 100 | 1000 |  |
| Adam | w/o | 0.065 | 0.029 | 0.017 | 19 h |
| Adam | w/ | 0.039 | 0.023 | 0.014 | 19 h |
| PSGD | w/ | 0.023 | 0.015 | 0.011 | 25 h |

**Evaluation** For each dataset, we randomly select 64 trajectories $x^{0:L}$ with various parameters $\theta$ in the test set. For each latent-space emulator, we encode the initial state $z_0 = E_\psi(x_0)$ and produce 16 distinct autoregressive rollouts $z^{1:L}$. For the diffusion models, sampling is performed with 16 steps of the 3rd order Adams-Bashforth multi-step integration method [73]. The metrics (VRMSE, power spectrum RMSE, spread-skill ratio) are then measured between the predicted states $\hat{x}^i = D_\psi(z^i)$ and the ground-truth states $x^i$ or the auto-encoded states $D_\psi(E_\psi(x^i))$.

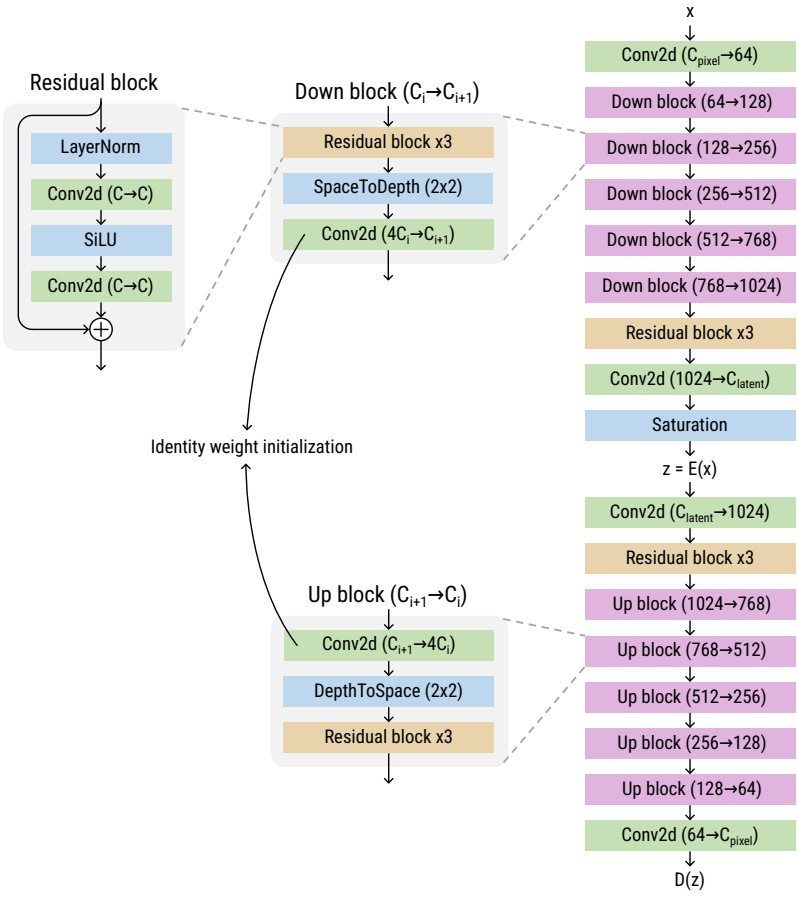

Figure 7. Schematic representation of the autoencoder architecture. Downsampling (resp. upsampling) is performed with a space-to-depth (resp. depth-to-space) operation followed (resp. preceded) with a convolution initialized near identity.

Table 5. Hyperparameters for the emulators.

|  | Latent-space | Pixel-space |
|---|---|---|
| Architecture | Transformer | Transformer |
| Parameters | $2.2 \times 10^8$ | $8.6 \times 10^8$ |
| Input shape | $C_{\text{latent}} \times (n+1) \times \frac{H}{32} \times \frac{W}{32}$ | $C_{\text{pixel}} \times (n+1) \times H \times W$ |
| Patch size | $1 \times 1 \times 1$ | $1 \times 16 \times 16$ |
| Tokens | $(n+1) \times \frac{H}{32} \times \frac{W}{32}$ | $(n+1) \times \frac{H}{16} \times \frac{W}{16}$ |
| Embedding size | 1024 | 2048 |
| Blocks | 16 | 16 |
| Positional embedding | Absolute + RoPE | Absolute + RoPE |
| Activation | SiLU | SiLU |
| Normalization | LayerNorm | LayerNorm |
| Dropout | 0.05 | 0.05 |
| Optimizer | Adam | Adam |
| Learning rate | $10^{-4}$ | $10^{-4}$ |
| Weight decay | 0.0 | 0.0 |
| Scheduler | cosine | cosine |
| Gradient norm clipping | 1.0 | 1.0 |
| Batch size | 256 | 256 |
| Steps per epoch | 64 | 64 |
| Epochs | 4096 | 4096 |
| GPUs | 8 | 16 |

# C Additional emulation results

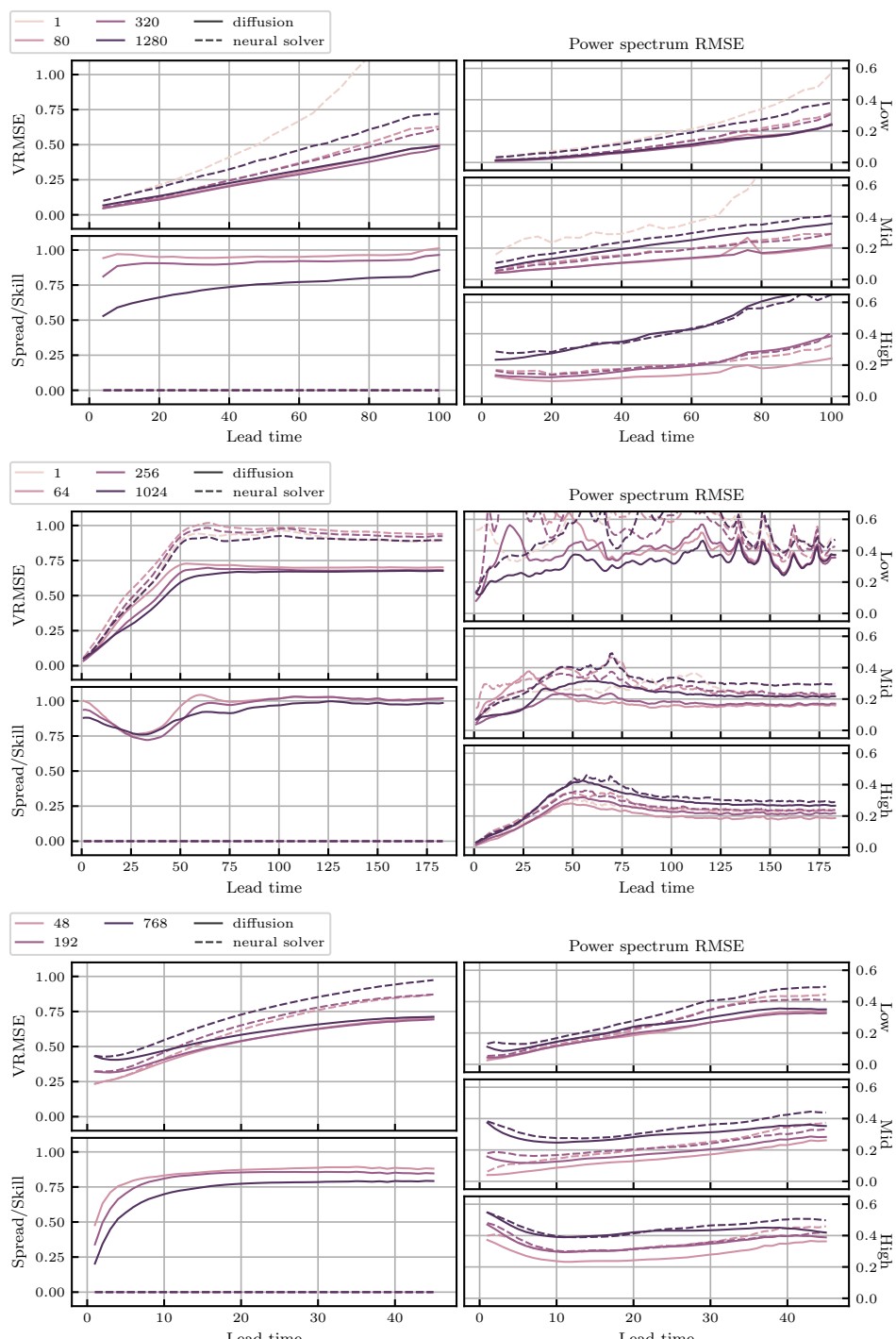

Figure 8. Average evaluation metrics of latent-space emulation for the Euler (top), RB (center) and TGC (bottom) datasets. As expected from imperfect emulators, the emulation error grows with the lead time. However, increasing the compression rate does not degrade (Euler, TGC) and sometimes improves (RB) the accuracy of diffusion models. The spread-skill ratio [25, 75] drops slightly with the compression rate, which could be a sign of overfitting. Diffusion-based emulators are consistently more accurate than neural solvers.

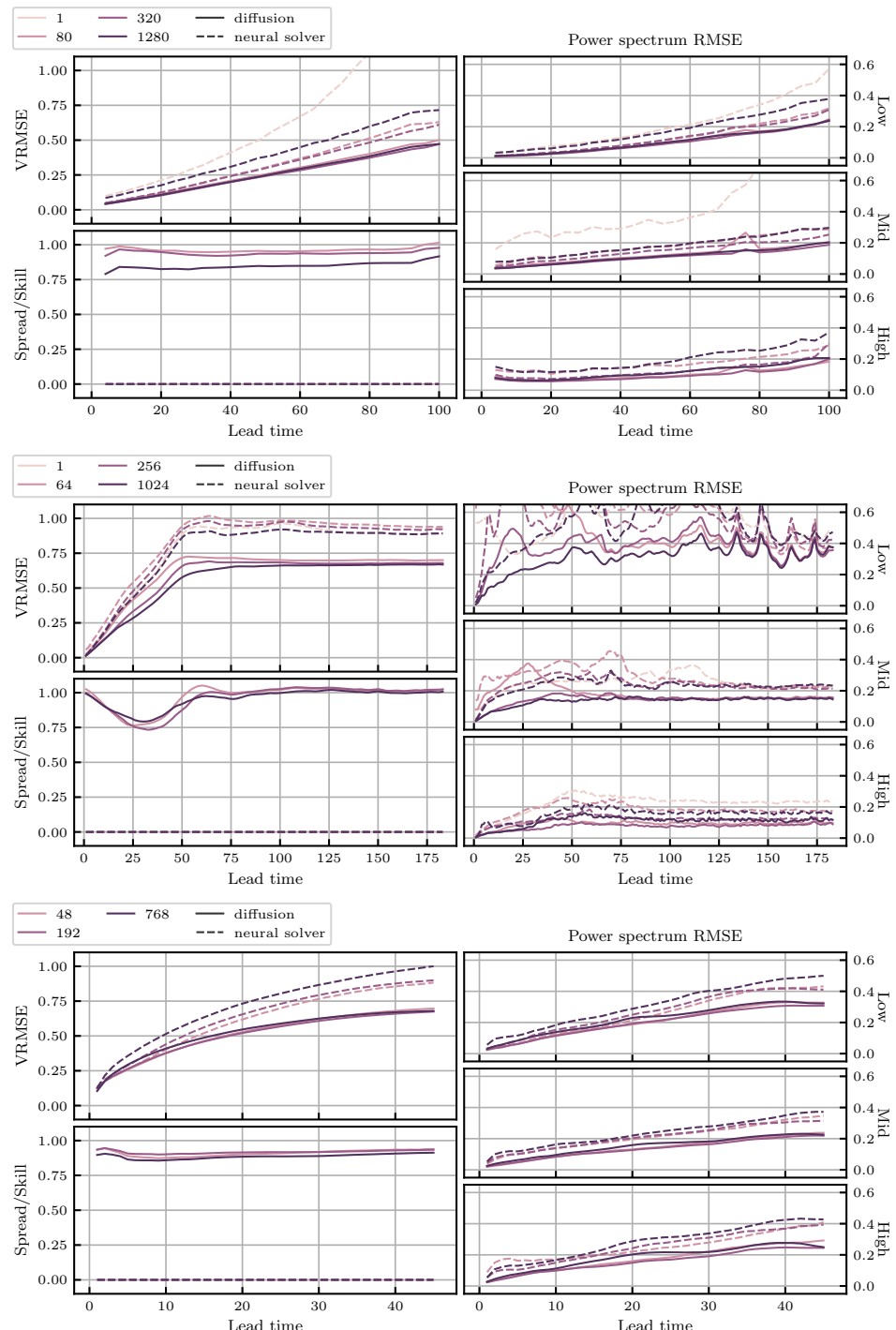

Figure 9. Average evaluation metrics of latent-space emulation relative to the auto-encoded states $D_\psi(E_\psi(x^i))$ for the Euler (top), RB (center) and TGC (bottom) datasets. As expected from imperfect emulators, the emulation error grows with the lead time. However, increasing the compression rate does not degrade (Euler, TGC) and sometimes improves (RB) the accuracy of diffusion models. The spread-skill ratio [25, 75] drops slightly with the compression rate, which could be a sign of overfitting. Diffusion-based emulators are consistently more accurate than neural solvers.

Table 6. Average VRMSE of autoencoder reconstruction and latent-space emulation at different compression rates ($\div$) and lead time horizons for the Euler, RB and TGC datasets. Increasing the compression rate has a clear impact on reconstruction quality, but does not degrade significantly (Euler, TGC) and sometimes improves (RB) the accuracy of diffusion models.

| Method | Euler | | | | Method | RB | | | |
|---|---|---|---|---|---|---|---|---|---|
| | $\div$ | 1:20 | 21:60 | 61:100 | | $\div$ | 1:20 | 21:60 | 61:180 |
| autoencoder | 80 | 0.011 | 0.014 | 0.020 | autoencoder | 64 | 0.023 | 0.033 | 0.019 |
| | 320 | 0.023 | 0.041 | 0.061 | | 256 | 0.039 | 0.064 | 0.042 |
| | 1280 | 0.060 | 0.107 | 0.144 | | 1024 | 0.070 | 0.124 | 0.092 |
| diffusion | 80 | 0.075 | 0.199 | 0.395 | diffusion | 64 | 0.171 | 0.582 | 0.704 |
| | 320 | 0.070 | 0.192 | 0.371 | | 256 | 0.141 | 0.509 | 0.683 |
| | 1280 | 0.093 | 0.217 | 0.400 | | 1024 | 0.146 | 0.457 | 0.670 |
| neural solver | 1 | 0.138 | 0.397 | 1.102 | neural solver | 1 | 0.185 | 0.681 | 0.918 |
| | 80 | 0.077 | 0.232 | 0.500 | | 64 | 0.244 | 0.761 | 0.968 |
| | 320 | 0.080 | 0.232 | 0.476 | | 256 | 0.197 | 0.716 | 0.945 |
| | 1280 | 0.137 | 0.314 | 0.592 | | 1024 | 0.195 | 0.665 | 0.903 |

| Method | TGC | | | |
|---|---|---|---|---|
| | $\div$ | 1:10 | 11:20 | 21:50 |
| autoencoder | 48 | 0.151 | 0.116 | 0.129 |
| | 192 | 0.229 | 0.175 | 0.189 |
| | 768 | 0.338 | 0.272 | 0.276 |
| diffusion | 48 | 0.296 | 0.522 | 0.673 |
| | 192 | 0.342 | 0.527 | 0.665 |
| | 768 | 0.425 | 0.575 | 0.694 |
| neural solver | 48 | 0.302 | 0.599 | 0.826 |
| | 192 | 0.361 | 0.632 | 0.835 |
| | 768 | 0.462 | 0.710 | 0.920 |

Table 7. Average VRMSE of latent-space emulation at different context lengths ($c$) and lead time horizons for the Euler, RB and TGC datasets. We can test different context lengths without retraining as our models were trained for different conditioning tasks (see Section 3). Perhaps surprisingly, context lengths does not have a significant impact on emulation accuracy.

| Method | Euler | | | | Method | RB | | | |
|---|---|---|---|---|---|---|---|---|---|
| | $c$ | 1:20 | 21:60 | 61:100 | | $c$ | 1:20 | 21:60 | 61:180 |
| diffusion | 1 | 0.085 | 0.204 | 0.393 | diffusion | 1 | 0.152 | 0.510 | 0.683 |
| | 2 | 0.074 | 0.200 | 0.383 | | 2 | 0.150 | 0.511 | 0.685 |
| | 3 | 0.078 | 0.203 | 0.389 | | 3 | 0.157 | 0.527 | 0.689 |
| neural solver | 1 | 0.108 | 0.266 | 0.526 | neural solver | 1 | 0.208 | 0.705 | 0.932 |
| | 2 | 0.092 | 0.253 | 0.513 | | 2 | 0.209 | 0.708 | 0.943 |
| | 3 | 0.094 | 0.260 | 0.529 | | 3 | 0.220 | 0.728 | 0.940 |

| Method | TGC | | | |
|---|---|---|---|---|
| | $c$ | 1:10 | 11:20 | 21:50 |
| diffusion | 1 | 0.362 | 0.550 | 0.681 |
| | 2 | 0.351 | 0.535 | 0.669 |
| | 3 | 0.350 | 0.539 | 0.683 |
| neural solver | 1 | 0.376 | 0.632 | 0.837 |
| | 2 | 0.371 | 0.641 | 0.855 |
| | 3 | 0.378 | 0.669 | 0.888 |

Table 8. Average power spectrum RMSE of autoencoder reconstruction and latent-space emulation at different compression rates (÷) and lead time horizons for the Euler dataset. The high-frequency content of diffusion-based emulators is limited by the autoencoder's reconstruction capabilities.

| Method | ÷ | Low | | | Mid | | | High | | |
|---|---|---|---|---|---|---|---|---|---|---|
| | | 1:20 | 21:60 | 61:100 | 1:20 | 21:60 | 61:100 | 1:20 | 21:60 | 61:100 |
| autoencoder | 80 | 0.001 | 0.001 | 0.001 | 0.006 | 0.008 | 0.014 | 0.072 | 0.069 | 0.096 |
| | 320 | 0.002 | 0.003 | 0.004 | 0.022 | 0.047 | 0.085 | 0.112 | 0.141 | 0.240 |
| | 1280 | 0.009 | 0.017 | 0.025 | 0.074 | 0.167 | 0.264 | 0.240 | 0.355 | 0.577 |
| diffusion | 80 | 0.017 | 0.063 | 0.168 | 0.054 | 0.100 | 0.178 | 0.112 | 0.116 | 0.184 |
| | 320 | 0.014 | 0.058 | 0.157 | 0.052 | 0.102 | 0.171 | 0.128 | 0.155 | 0.275 |
| | 1280 | 0.019 | 0.065 | 0.163 | 0.096 | 0.187 | 0.300 | 0.246 | 0.349 | 0.569 |
| neural solver | 1 | 0.046 | 0.128 | 0.339 | 0.227 | 0.297 | 0.754 | 0.821 | 0.984 | 2.666 |
| | 80 | 0.021 | 0.074 | 0.212 | 0.085 | 0.151 | 0.245 | 0.164 | 0.173 | 0.249 |
| | 320 | 0.020 | 0.075 | 0.204 | 0.074 | 0.144 | 0.234 | 0.151 | 0.169 | 0.271 |
| | 1280 | 0.045 | 0.116 | 0.274 | 0.131 | 0.227 | 0.349 | 0.283 | 0.345 | 0.545 |

Table 9. Average power spectrum RMSE of autoencoder reconstruction and latent-space emulation at different compression rates (÷) and lead time horizons for the Rayleigh-Benard dataset. The high-frequency content of diffusion-based emulators is limited by the autoencoder's reconstruction capabilities.

| Method | ÷ | Low | | | Mid | | | High | | |
|---|---|---|---|---|---|---|---|---|---|---|
| | | 1:20 | 21:60 | 61:180 | 1:20 | 21:60 | 61:180 | 1:20 | 21:60 | 61:180 |
| autoencoder | 64 | 0.043 | 0.004 | 0.001 | 0.011 | 0.013 | 0.012 | 0.026 | 0.159 | 0.148 |
| | 256 | 0.061 | 0.011 | 0.004 | 0.028 | 0.080 | 0.075 | 0.050 | 0.220 | 0.212 |
| | 1024 | 0.121 | 0.033 | 0.018 | 0.063 | 0.186 | 0.197 | 0.076 | 0.294 | 0.294 |
| diffusion | 64 | 1.751 | 0.850 | 0.386 | 0.197 | 0.266 | 0.159 | 0.054 | 0.220 | 0.199 |
| | 256 | 0.328 | 0.399 | 0.396 | 0.084 | 0.195 | 0.177 | 0.065 | 0.239 | 0.232 |
| | 1024 | 0.193 | 0.292 | 0.344 | 0.095 | 0.243 | 0.240 | 0.083 | 0.314 | 0.297 |
| neural solver | 1 | 0.467 | 0.520 | 0.650 | 0.151 | 0.255 | 0.264 | 0.076 | 0.232 | 0.242 |
| | 64 | 3.625 | 0.915 | 0.566 | 0.268 | 0.351 | 0.275 | 0.099 | 0.279 | 0.257 |
| | 256 | 0.575 | 0.675 | 0.526 | 0.165 | 0.317 | 0.264 | 0.091 | 0.275 | 0.257 |
| | 1024 | 0.285 | 0.496 | 0.560 | 0.152 | 0.338 | 0.321 | 0.090 | 0.320 | 0.326 |

Table 10. Average power spectrum RMSE of autoencoder reconstruction and latent-space emulation at different compression rates (÷) and lead time horizons for the TGC dataset. The high-frequency content of diffusion-based emulators is limited by the autoencoder's reconstruction capabilities.

| Method | ÷ | Low | | | Mid | | | High | | |
|---|---|---|---|---|---|---|---|---|---|---|
| | | 1:10 | 11:30 | 31:50 | 1:10 | 11:30 | 31:50 | 1:10 | 11:30 | 31:50 |
| autoencoder | 48 | 0.011 | 0.016 | 0.025 | 0.023 | 0.026 | 0.044 | 0.275 | 0.188 | 0.195 |
| | 192 | 0.028 | 0.033 | 0.045 | 0.108 | 0.091 | 0.114 | 0.359 | 0.273 | 0.282 |
| | 768 | 0.072 | 0.068 | 0.080 | 0.285 | 0.235 | 0.254 | 0.454 | 0.476 | 0.367 |
| diffusion | 48 | 0.064 | 0.185 | 0.319 | 0.058 | 0.128 | 0.220 | 0.296 | 0.247 | 0.331 |
| | 192 | 0.069 | 0.191 | 0.311 | 0.128 | 0.164 | 0.252 | 0.369 | 0.316 | 0.384 |
| | 768 | 0.107 | 0.294 | 0.425 | 0.289 | 0.305 | 0.360 | 0.456 | 0.419 | 0.444 |
| neural solver | 48 | 0.070 | 0.221 | 0.424 | 0.110 | 0.197 | 0.324 | 0.357 | 0.320 | 0.427 |
| | 192 | 0.086 | 0.228 | 0.402 | 0.172 | 0.201 | 0.295 | 0.391 | 0.317 | 0.395 |
| | 768 | 0.138 | 0.277 | 0.465 | 0.322 | 0.305 | 0.407 | 0.471 | 0.418 | 0.493 |

Table 11. Average sliced earth mover's distance (SEMD) [115, 116] of the density field of autoencoder reconstruction and latent-space emulation at different compression rates ($\div$) and lead time horizons for the Euler dataset. The SEMD is small and is not significantly impacted by the compression rate, especially for diffusion models. For reference, the density fields of two consecutive states $x^i$ and $x^{i+1}$ have a typical SEMD of $0.0025$. *Why this metric?* The Euler equations are sometimes used in aerodynamics to model flow around objects and one is typically interested in the global fluid displacement. The rationale for using this metric is that a small drift in the density field would not significantly affect the (S)EMD, while it could affect point-wise metrics heavily.

| Method | | EMD (density field) | | |
|---|---|---|---|---|
| | $\div$ | 1:20 | 21:60 | 61:100 |
| autoencoder | 80 | 0.0000 | 0.0000 | 0.0000 |
| | 320 | 0.0001 | 0.0001 | 0.0001 |
| | 1280 | 0.0002 | 0.0003 | 0.0005 |
| diffusion | 80 | 0.0004 | 0.0010 | 0.0023 |
| | 320 | 0.0003 | 0.0009 | 0.0022 |
| | 1280 | 0.0004 | 0.0010 | 0.0023 |
| neural solver | 1 | 0.0011 | 0.0031 | 0.0066 |
| | 80 | 0.0005 | 0.0012 | 0.0028 |
| | 320 | 0.0004 | 0.0012 | 0.0027 |
| | 1280 | 0.0008 | 0.0020 | 0.0041 |

Table 12. Average Wasserstein distance of the distribution of vertical velocity values of autoencoder reconstruction and latent-space emulation at different compression rates ($\div$) and lead time horizons for the RB dataset. The Wasserstein distance is smaller for diffusion models and decreases with the compression rate. For reference, the distributions of vertical velocity values of two consecutive states $x^i$ and $x^{i+1}$ have a typical Wasserstein distance of $0.004$. *Why this metric?* One interesting quantity in buoyancy-driven convection is the growth speed of plumes in the fluid. The distribution of the (vertical) velocity values is a good summary statistic for tracking the growth of plumes.

| Method | | Wasserstein (vertical velocity field) | | |
|---|---|---|---|---|
| | $\div$ | 1:20 | 21:60 | 61:180 |
| autoencoder | 64 | 0.0000 | 0.0002 | 0.0002 |
| | 256 | 0.0001 | 0.0007 | 0.0005 |
| | 1024 | 0.0002 | 0.0020 | 0.0018 |
| diffusion | 64 | 0.0003 | 0.0104 | 0.0141 |
| | 256 | 0.0003 | 0.0092 | 0.0141 |
| | 1024 | 0.0004 | 0.0063 | 0.0139 |
| neural solver | 1 | 0.0003 | 0.0153 | 0.0247 |
| | 64 | 0.0009 | 0.0272 | 0.0223 |
| | 256 | 0.0007 | 0.0197 | 0.0187 |
| | 1024 | 0.0007 | 0.0157 | 0.0206 |

Table 13. Average Wasserstein distance of the distribution of density values of autoencoder reconstruction and latent-space emulation at different compression rates ($\div$) and lead time horizons for the TGC dataset. The Wasserstein distance is smaller for diffusion models, but grows significantly with the lead time, even for the autoencoder reconstruction. For reference, the distributions of density values of two consecutive states $x^i$ and $x^{i+1}$ have a typical Wasserstein distance of $0.01$. *Why this metric?* In the interstellar medium, gravity forms clusters of matter that eventually lead to the birst of stars. The kind of clusters (compact, diffuse, or anything in between) and their proportions is of interest for domain-scientists. The distribution of the density values is a good summary statistic for clustering dynamics.

| Method | Wasserstein (density field) | | | |
|---|---|---|---|---|
| | $\div$ | 1:10 | 11:30 | 31:50 |
| autoencoder | 48 | 0.0034 | 0.0048 | 0.0089 |
| | 192 | 0.0082 | 0.0110 | 0.0183 |
| | 768 | 0.0181 | 0.0236 | 0.0338 |
| diffusion | 48 | 0.0044 | 0.0138 | 0.0266 |
| | 192 | 0.0089 | 0.0172 | 0.0310 |
| | 768 | 0.0186 | 0.0274 | 0.0425 |
| neural solver | 48 | 0.0074 | 0.0253 | 0.0524 |
| | 192 | 0.0091 | 0.0171 | 0.0329 |
| | 768 | 0.0220 | 0.0296 | 0.0492 |

Table 14. Average VRMSE results from various studies using TheWell [38] datasets. Even though our latent diffusion models (LDMs) outperform most published baselines, we emphasize that we do not position our models as state-of-the-art, notably due to the discrepancies in parameters count, training and evaluation.

| Source | Method | Dataset | Parameters | Lead time | VRMSE |
|---|---|---|---|---|---|
| Ohana et al. [38] | FNO | Euler | $2 \times 10^7$ | 6:12 | 1.13 |
| Ohana et al. [38] | U-Net | Euler | $2 \times 10^7$ | 6:12 | 1.02 |
| Ours | ViT | Euler | $8.6 \times 10^8$ | 1:20 | 0.138 |
| Ours | LDM $\div_{80}$ | Euler | $3.1 \times 10^8 + 2.2 \times 10^8$ | 1:20 | 0.075 |
| Ohana et al. [38] | FNO | RB | $2 \times 10^7$ | 6:12 | 10+ |
| Ohana et al. [38] | U-Net | RB | $2 \times 10^7$ | 6:12 | 10+ |
| Nguyen et al. [117] | PhysiX | RB | $4.5 \times 10^9$ | 2:8 | 1.067 |
| Wu et al. [118] | TANTE | RB | $10^8$ | 1:16 | 0.609 |
| Mukhopadhyay et al. [119] | ViT + CSM | RB | $10^8$ | 10 | 0.140 |
| Ours | ViT | RB | $8.6 \times 10^8$ | 1:20 | 0.185 |
| Ours | LDM $\div_{256}$ | RB | $3.1 \times 10^8 + 2.2 \times 10^8$ | 1:20 | 0.141 |
| Ohana et al. [38] | FNO | TGC | $2 \times 10^7$ | 6:12 | 3.55 |
| Ohana et al. [38] | U-Net | TGC | $2 \times 10^7$ | 6:12 | 7.14 |
| Mukhopadhyay et al. [119] | ViT + CKM | TGC | $10^8$ | 10 | 0.527 |
| Ours | LDM $\div_{48}$ | TGC | $7.2 \times 10^8 + 2.2 \times 10^8$ | 1:10 | 0.296 |

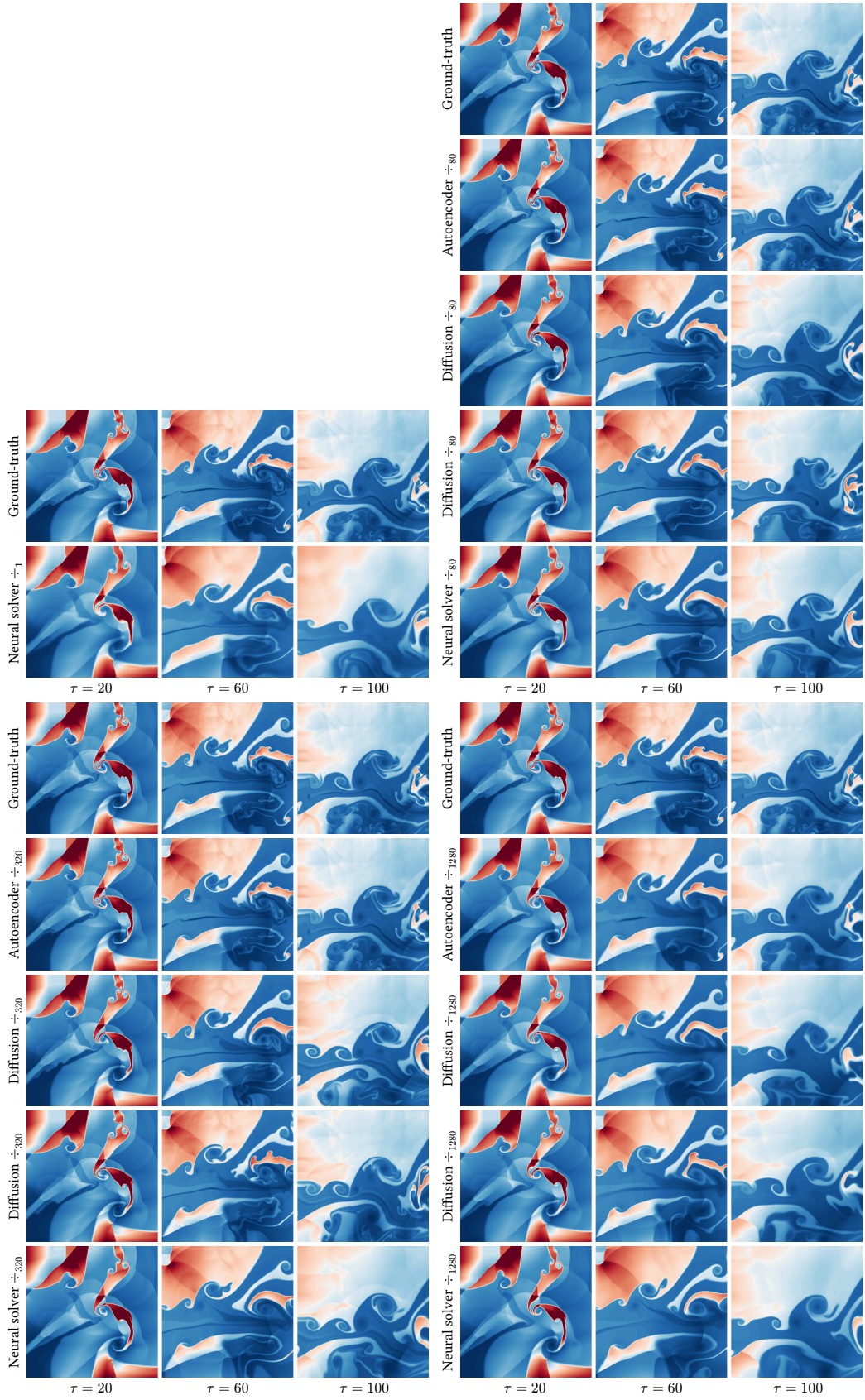

Figure 10. Examples of emulation at different compression rates (÷) for the Euler dataset. In this simulation, the system has open boundary conditions.

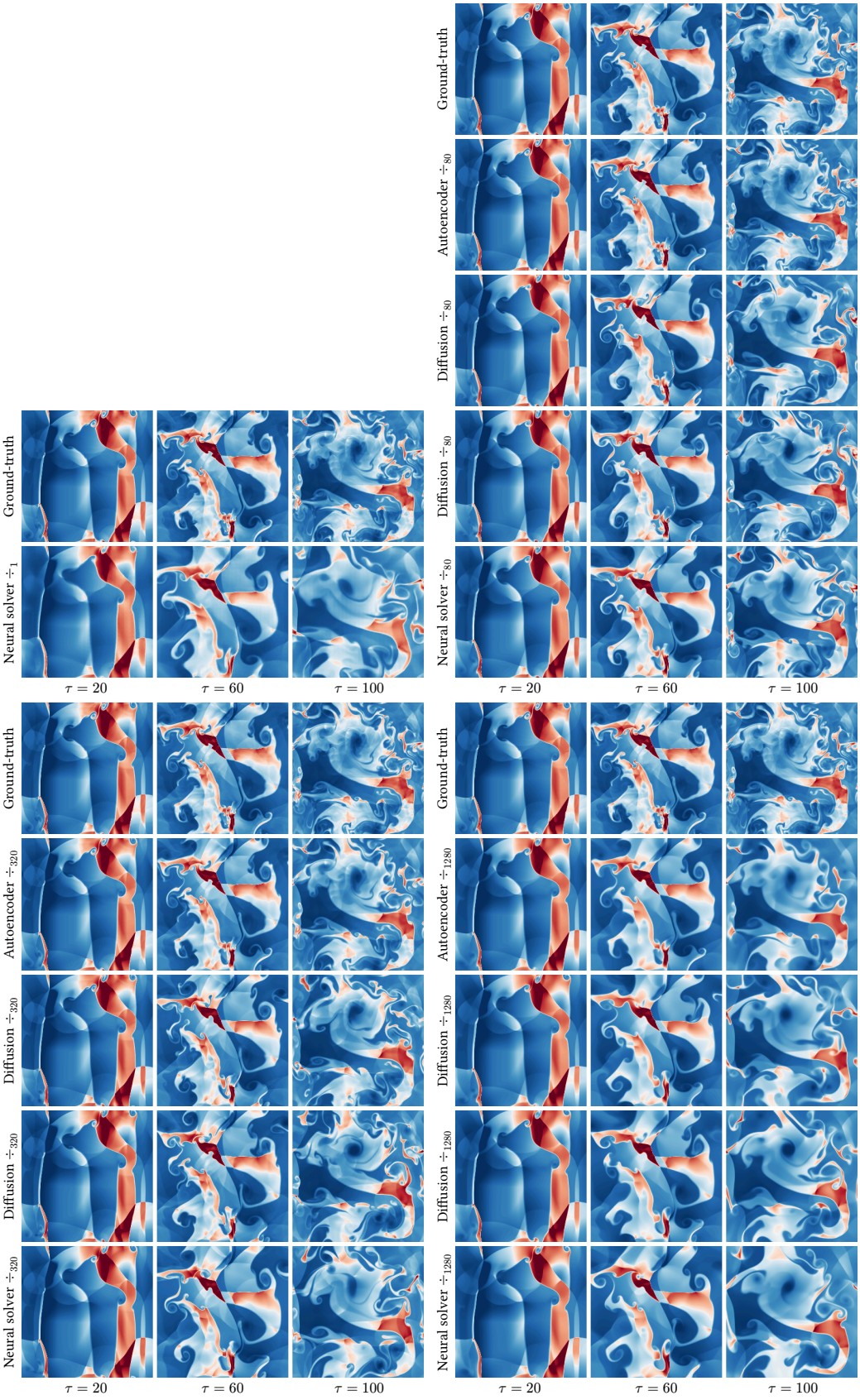

Figure 11. Examples of emulation at different compression rates ($\div$) for the Euler dataset. In this simulation, the system has periodic boundary conditions.

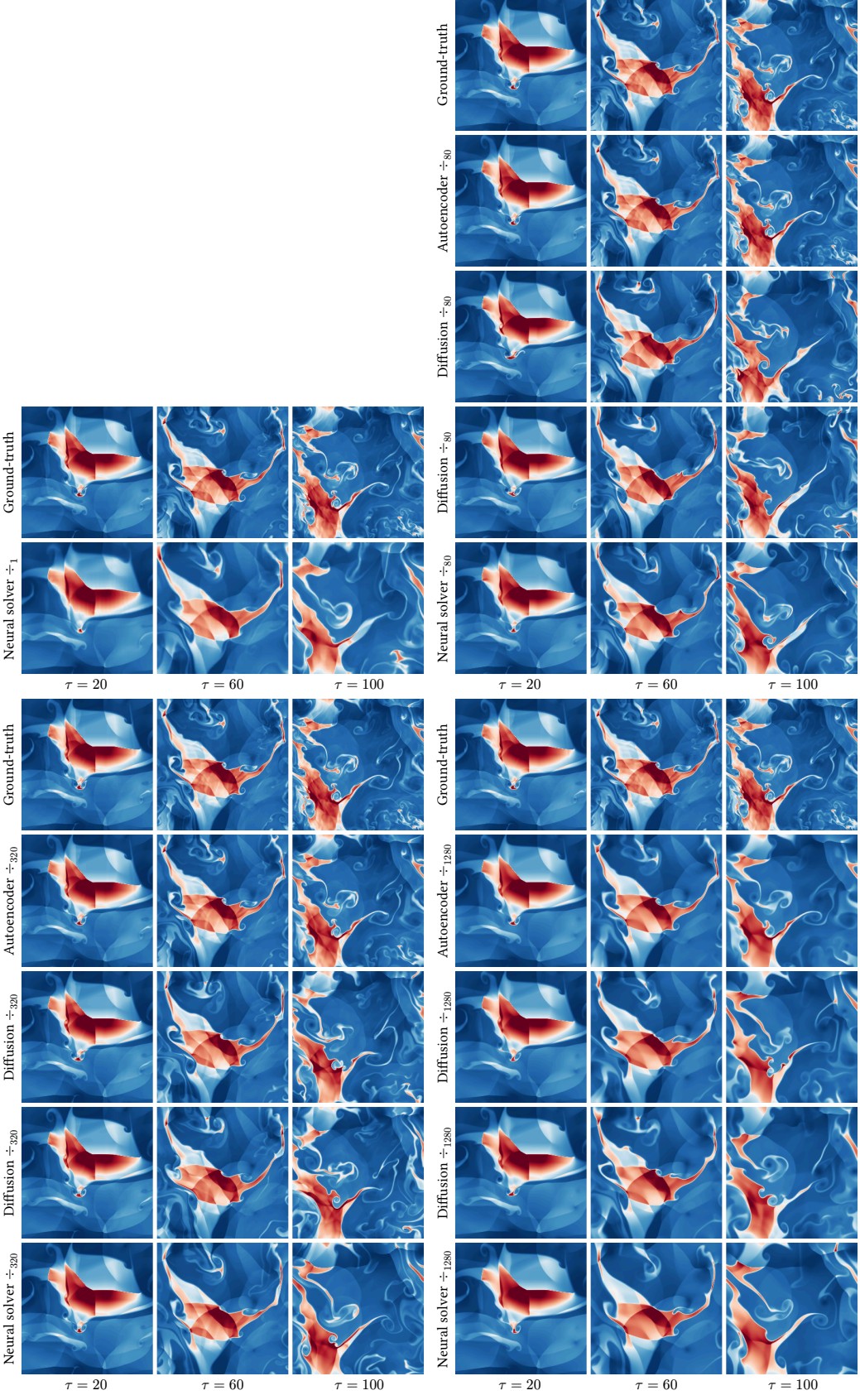

Figure 12. Examples of emulation at different compression rates ($\div$) for the Euler dataset. In this simulation, the system has periodic boundary conditions.

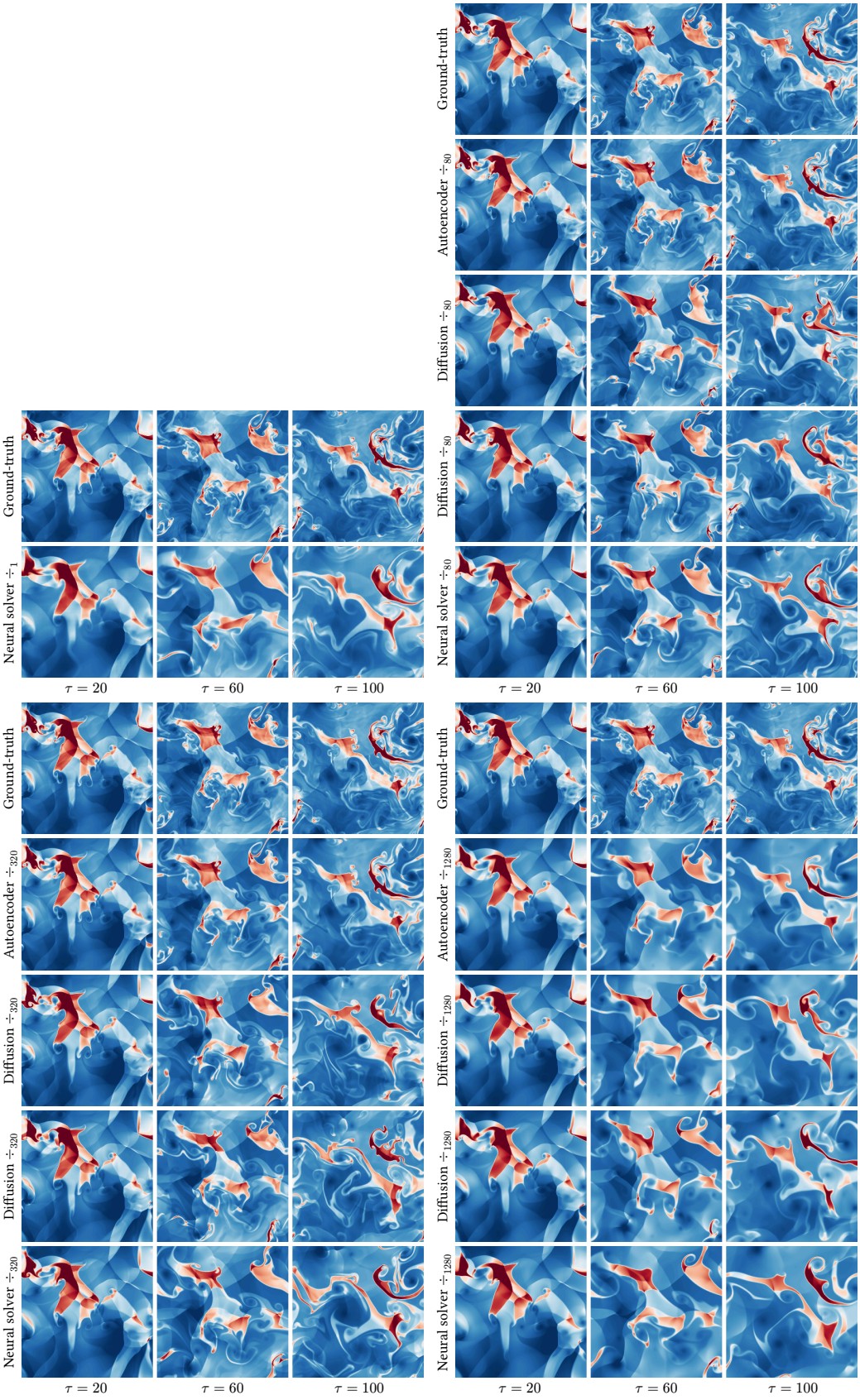

Figure 13. Examples of emulation at different compression rates (÷) for the Euler dataset. In this simulation, the system has periodic boundary conditions.

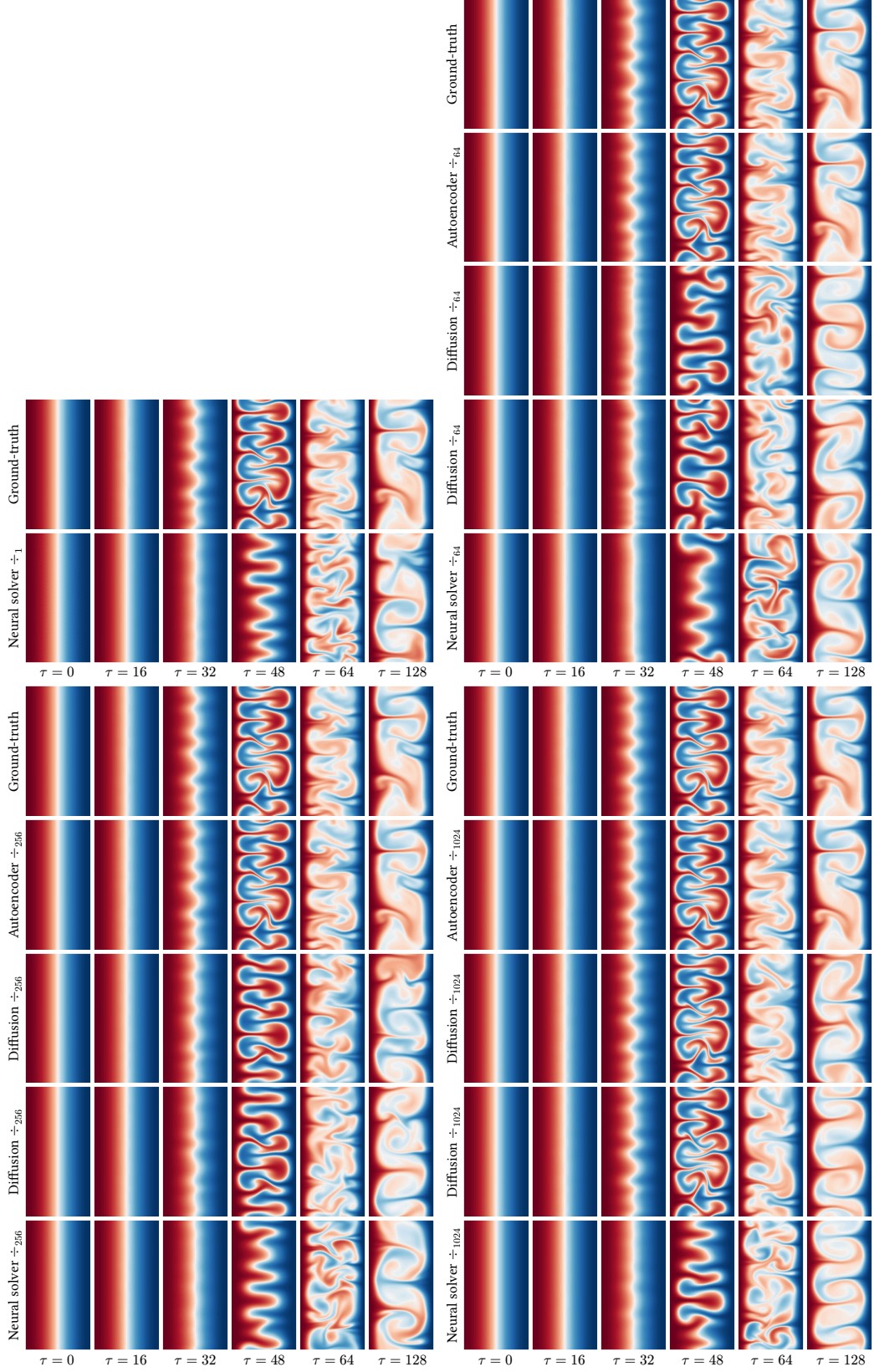

Figure 14. Examples of emulation at different compression rates ($\div$) for the Rayleigh-Bénard dataset. In this simulation, the fluid is in a low-turbulence regime ($\mathrm{Ra} = 10^6$).

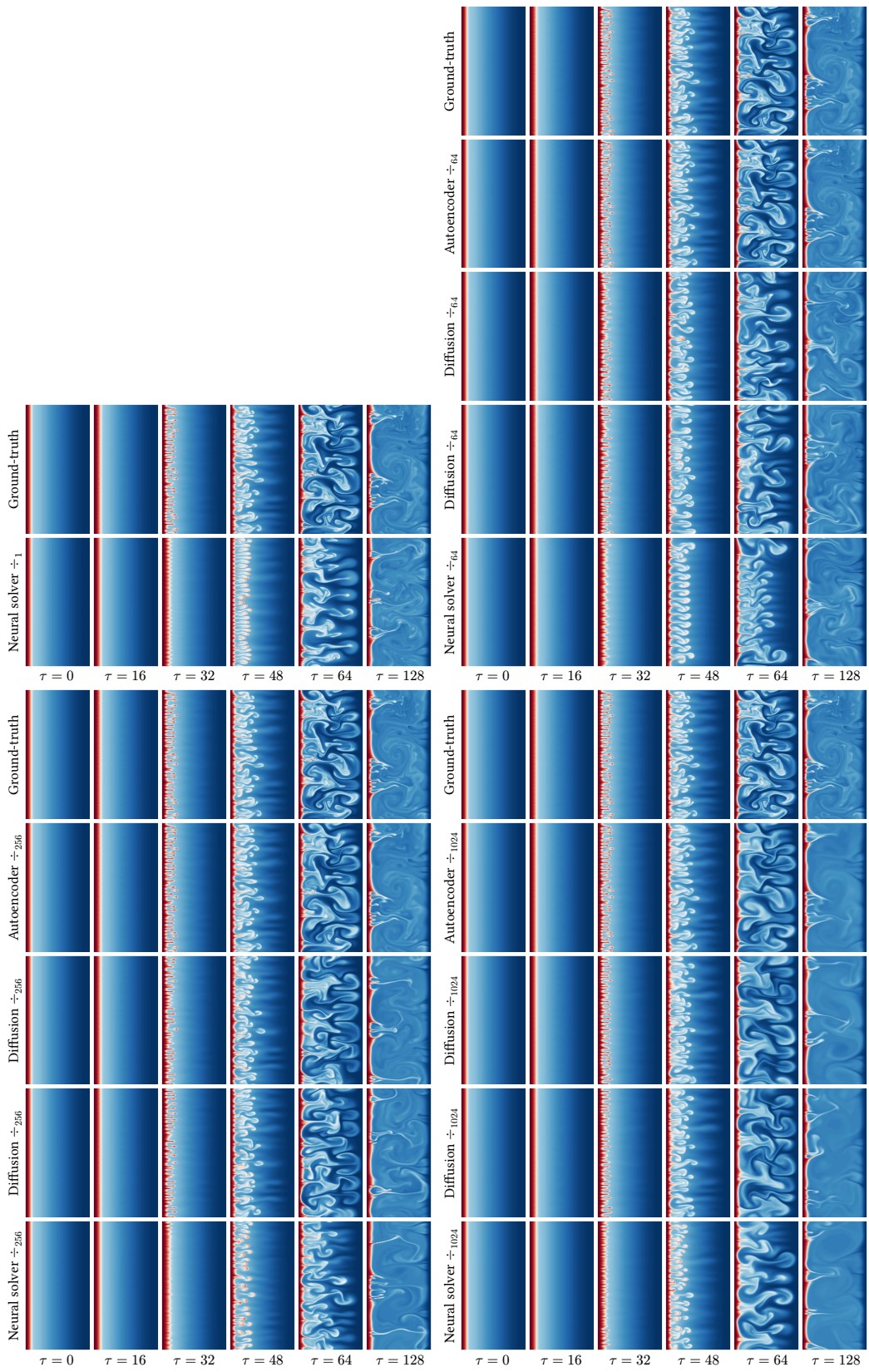

Figure 15. Examples of emulation at different compression rates ($\div$) for the Rayleigh-Bénard dataset. In this simulation, the fluid is in a high-turbulence regime ($\mathrm{Ra} = 10^8$).

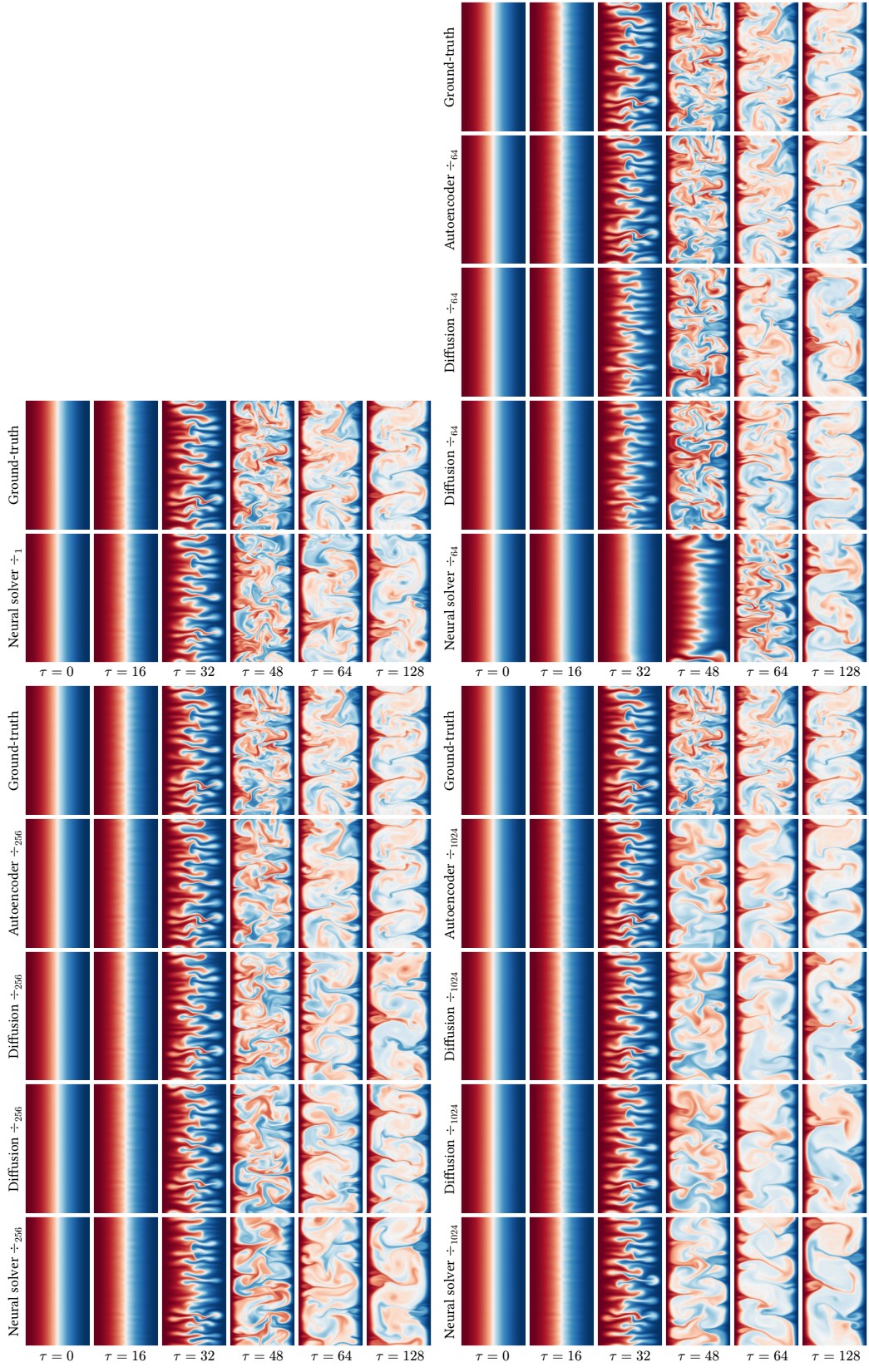

Figure 16. Examples of emulation at different compression rates ($\div$) for the Rayleigh-Bénard dataset. In this simulation, the fluid is in a low-turbulence regime ($\mathrm{Ra} = 10^6$).

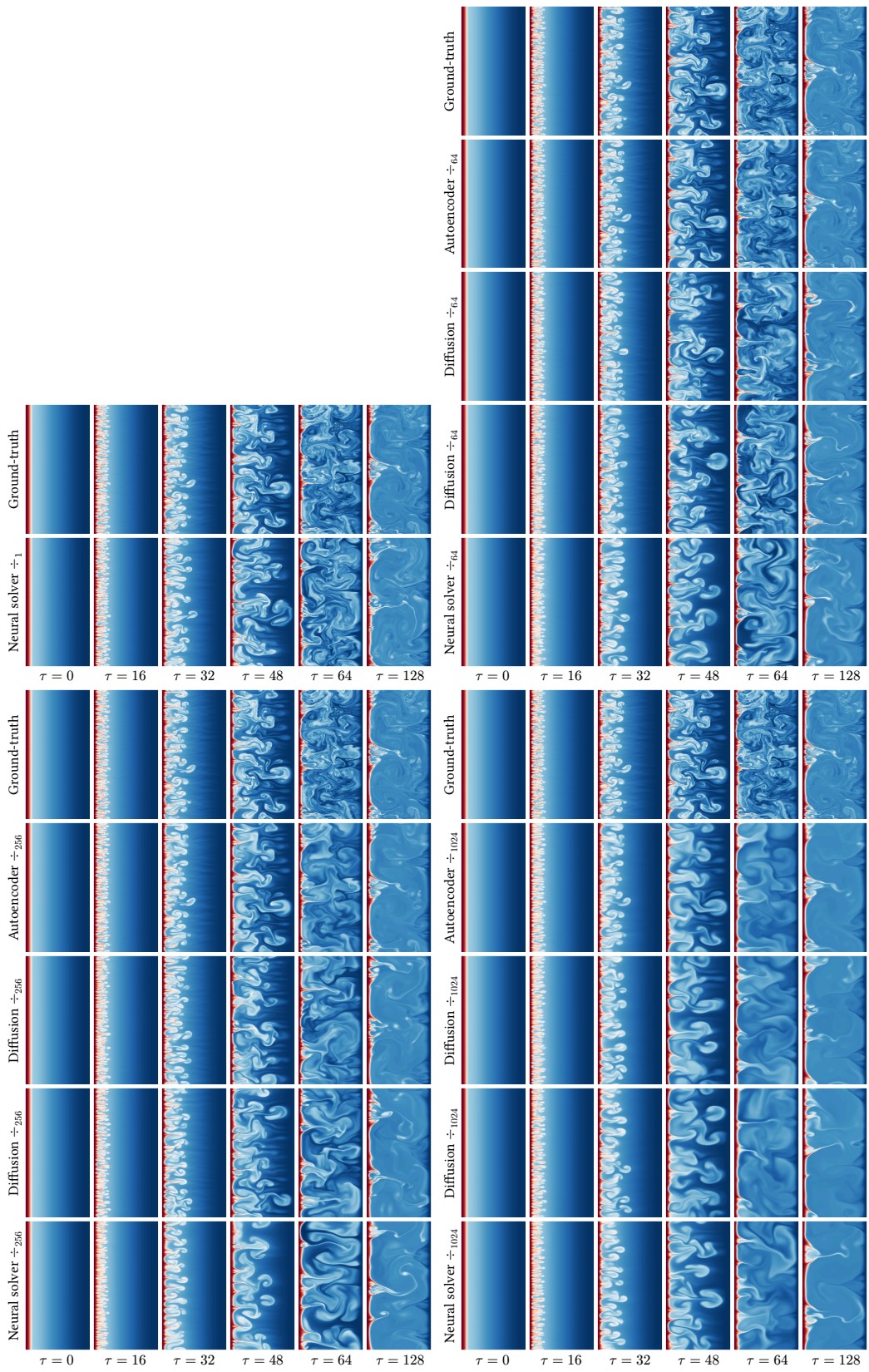

Figure 17. Examples of emulation at different compression rates (÷) for the Rayleigh-Bénard dataset. In this simulation, the fluid is in a high-turbulence regime ($\mathrm{Ra} = 10^8$).

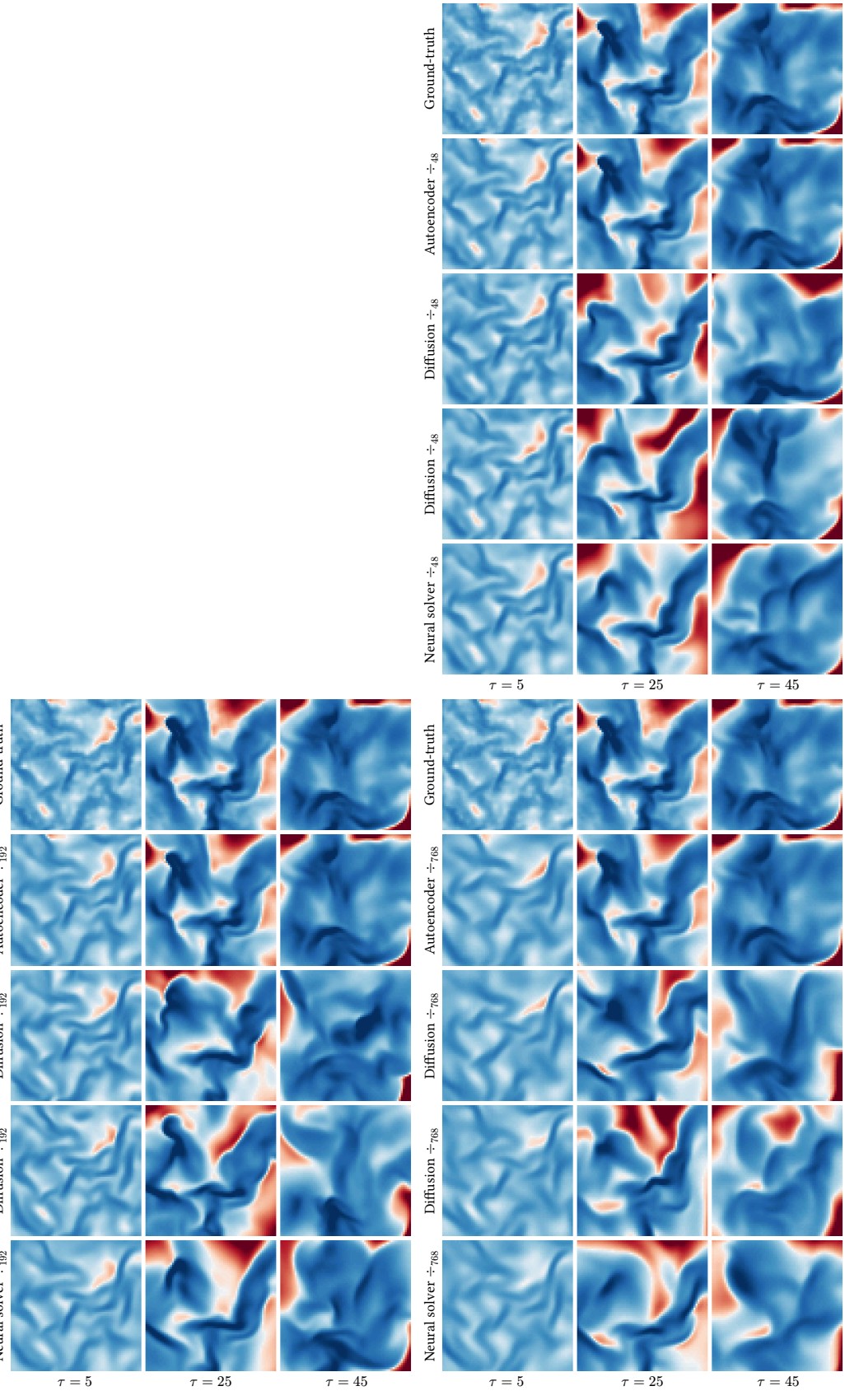

Figure 18. Examples of emulation at different compression rates (÷) for the TGC dataset. In this simulation, the initial density is low and the initial temperature is low ($\rho_0 = 0.445$, $T_0 = 10.0$).

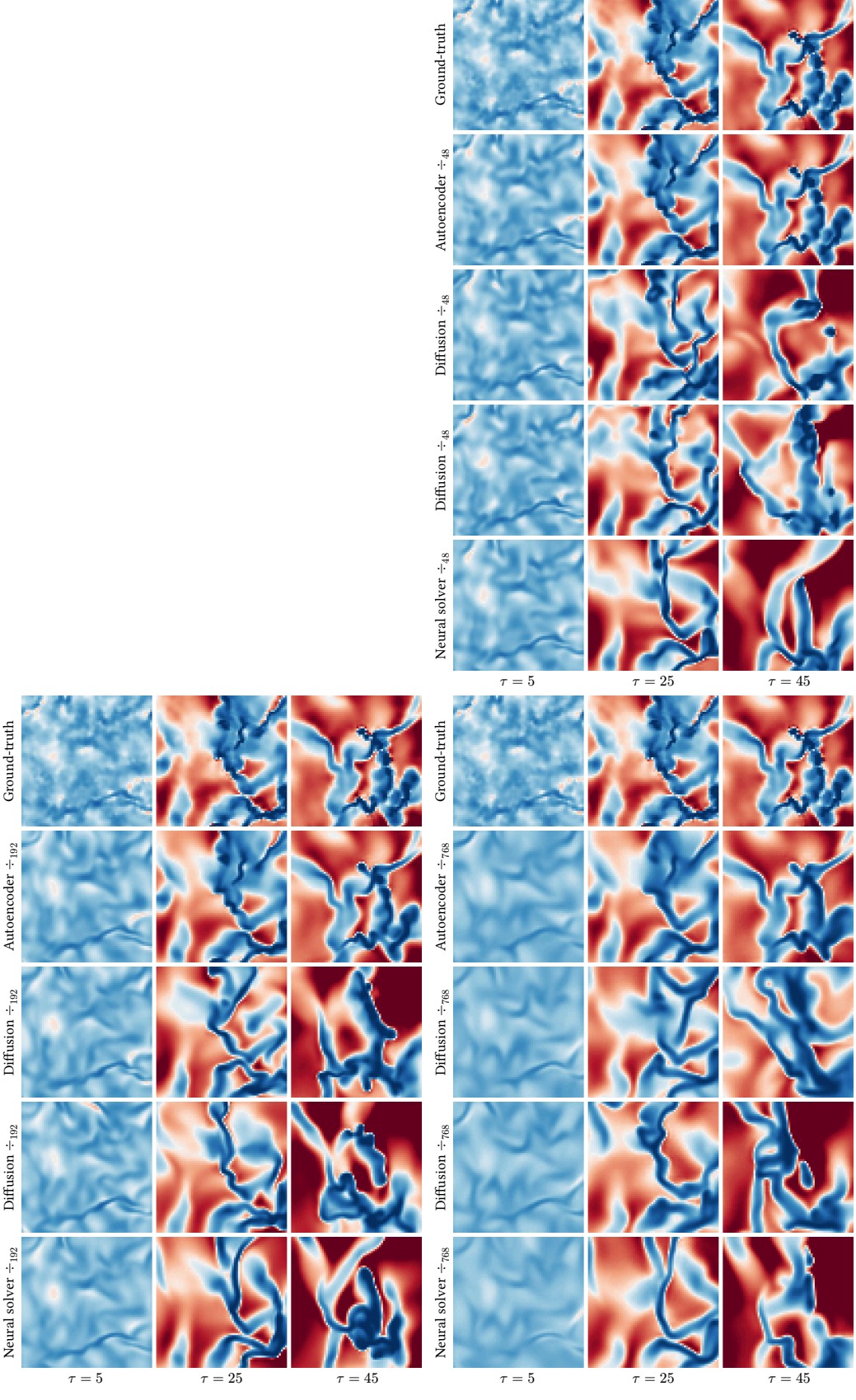

Figure 19. Examples of emulation at different compression rates ($\div$) for the TGC dataset. In this simulation, the initial density is medium and the initial temperature is high ($\rho_0 = 4.45$, $T_0 = 1000.0$).

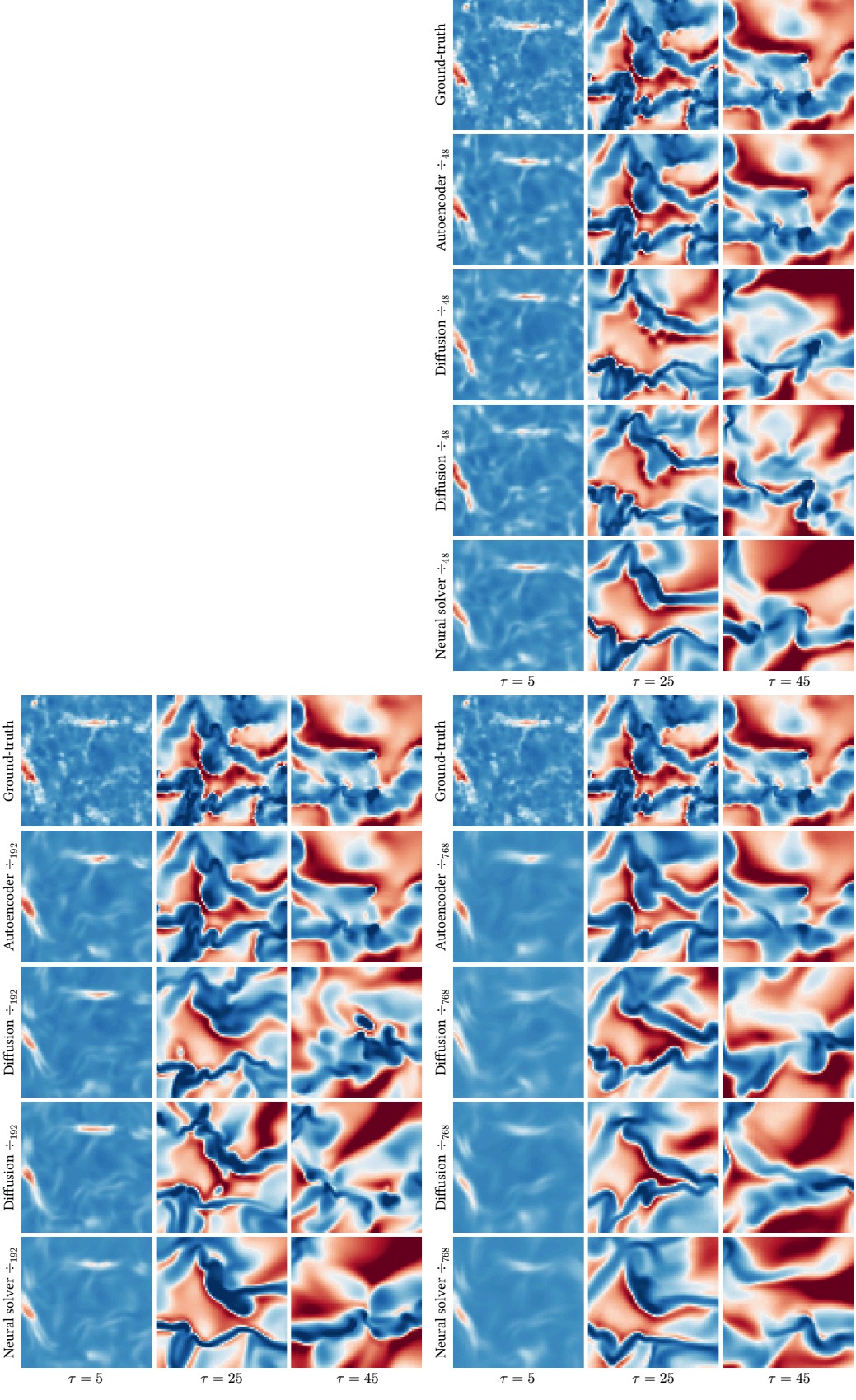

Figure 20. Examples of emulation at different compression rates ($\div$) for the TGC dataset. In this simulation, the initial density is high and the initial temperature is low ($\rho_0 = 44.5$, $T_0 = 10.0$).

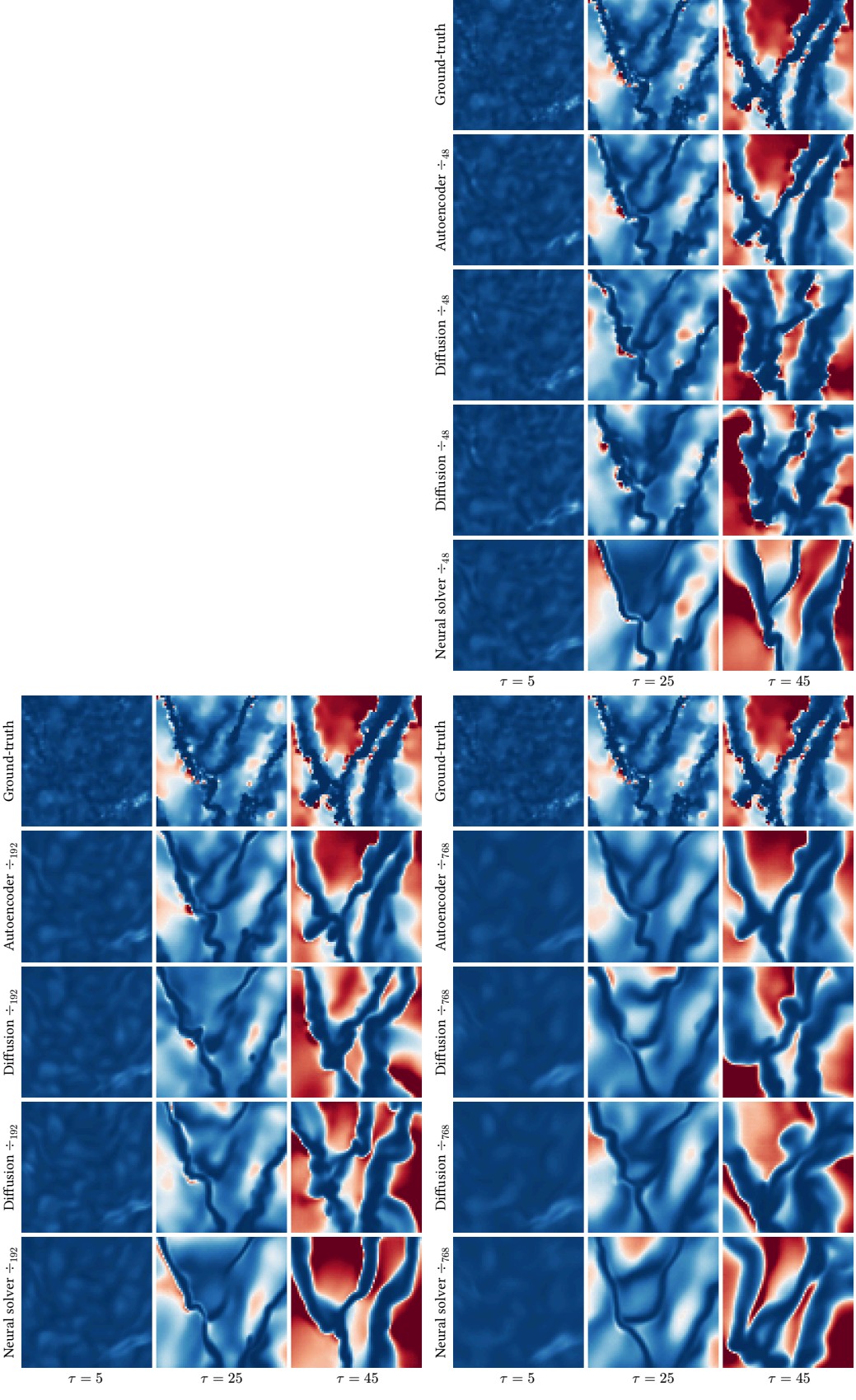

Figure 21. Examples of emulation at different compression rates ($\div$) for the TGC dataset. In this simulation, the initial density is high and the initial temperature is medium ($\rho_0 = 44.5$, $T_0 = 100.0$).

# D   Latent space analysis

In this section, we conduct a short analysis of the learned latent representations. We are notably interested in the separability of the latent representation with respect to different parameteres $\theta$.

For our first experiment, we select a random initial state $x^1$ from the test split of the Euler dataset. We compute the initial state $z^1 = E_\psi(x^1)$ for the $\div_{80}$ auto-encoder. For each heat capacity $\gamma \in \{1.2, 1.3, 1.4, 1.5, 1.6\}$, we generate one latent trajectory $z^{1:L}$ with the diffusion-based emulator. Afterwards we compute the Euclidean distance $||z_a^i - z_b^i||_2$ for each pair $(\gamma_a, \gamma_b)$ of heat capacities. We report the results in Table 15 and represent the trajectories in Figure 22. As expected, trajectories with similar heat capacity $\gamma$ are close to each others.

For our second experiment, we compute the latent representations $z^i = E_\psi(x^i) \in \mathbb{R}^{16 \times 4 \times 64}$ of the $\div_{64}$ auto-encoder for randomly selected states $x^i$ of the Rayleigh-Bénard dataset. We then train a small multi-layer perceptron (MLP) to predict the simulation parameters $\theta$ (Rayleigh and Prandtl numbers) from the latent state's central token $z^i[8, 2] \in \mathbb{R}^{64}$. We extract the activations of the MLP's last layer and visualize them with t-SNE [120] in Figure 23. We observe that t-SNE [120] continuously separates latent states with respect to their parameters $\theta$, indicating that our auto-encoders learn to distinguish between different physics. We further validate this result by computing the pairwise Bures-Wasserstein distances [121] between the distributions of central tokens $z^i[8, 2]$ for different Rayleigh and Prandtl numbers. The distances, reported in Tables 16 and 17, are anti-correlated with the similarity of simulation parameters $\theta$.

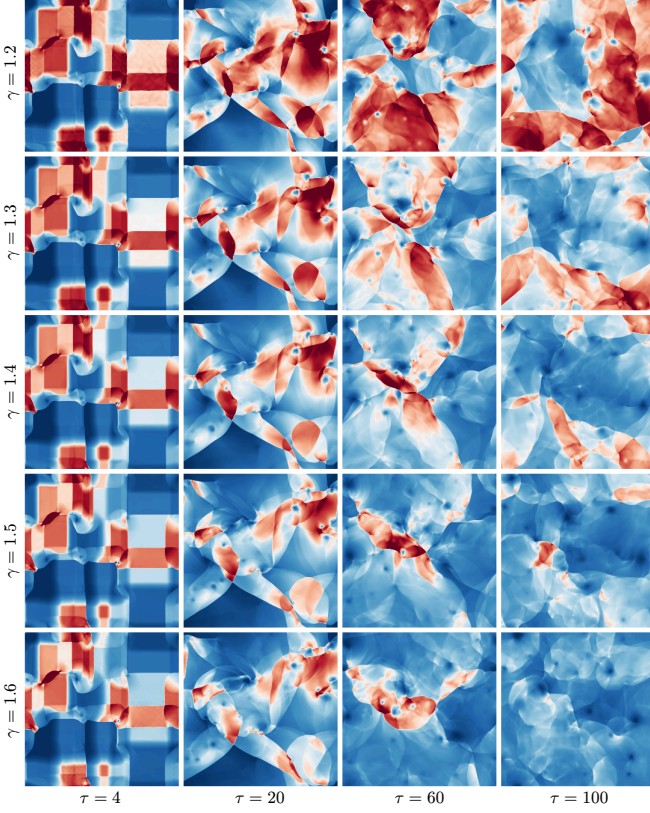

Figure 22. Example of emulated trajectories with different heat capacities $\gamma \in \{1.2, 1.3, 1.4, 1.5, 1.6\}$ but starting at the same initial state $x^1$ for the Euler dataset. The energy field is visualized instead of the density field to emphasize the differences.

Table 15. Euclidean distance matrix between emulated trajectories with different heat capacities $\gamma \in \{1.2, 1.3, 1.4, 1.5, 1.6\}$ but starting at the same initial state $x^1$ for the Euler dataset.

| $\tau = 1$ | 1.2 | 1.3 | 1.4 | 1.5 | 1.6 | $\tau = 20$ | 1.2 | 1.3 | 1.4 | 1.5 | 1.6 |
|---|---|---|---|---|---|---|---|---|---|---|---|
| 1.2 | 0.00 | 26.61 | 32.45 | 38.46 | 46.28 | 1.2 | 0.00 | 55.95 | 64.92 | 71.06 | 78.85 |
| 1.3 | 26.61 | 0.00 | 14.72 | 22.09 | 32.33 | 1.3 | 55.95 | 0.00 | 38.93 | 55.03 | 66.19 |
| 1.4 | 32.45 | 14.72 | 0.00 | 15.26 | 25.62 | 1.4 | 64.92 | 38.93 | 0.00 | 44.00 | 59.93 |
| 1.5 | 38.46 | 22.09 | 15.26 | 0.00 | 18.52 | 1.5 | 71.06 | 55.03 | 44.00 | 0.00 | 52.14 |
| 1.6 | 46.28 | 32.33 | 25.62 | 18.52 | 0.00 | 1.6 | 78.85 | 66.19 | 59.93 | 52.14 | 0.00 |
| $\tau = 60$ | 1.2 | 1.3 | 1.4 | 1.5 | 1.6 | $\tau = 100$ | 1.2 | 1.3 | 1.4 | 1.5 | 1.6 |
| 1.2 | 0.00 | 74.68 | 84.37 | 90.41 | 96.04 | 1.2 | 0.00 | 74.71 | 82.09 | 90.16 | 94.79 |
| 1.3 | 74.68 | 0.00 | 67.06 | 75.22 | 82.20 | 1.3 | 74.71 | 0.00 | 66.72 | 74.16 | 81.68 |
| 1.4 | 84.37 | 67.06 | 0.00 | 67.42 | 76.49 | 1.4 | 82.09 | 66.72 | 0.00 | 67.51 | 74.72 |
| 1.5 | 90.41 | 75.22 | 67.42 | 0.00 | 71.58 | 1.5 | 90.16 | 74.16 | 67.51 | 0.00 | 69.75 |
| 1.6 | 96.04 | 82.20 | 76.49 | 71.58 | 0.00 | 1.6 | 94.79 | 81.68 | 74.72 | 69.75 | 0.00 |

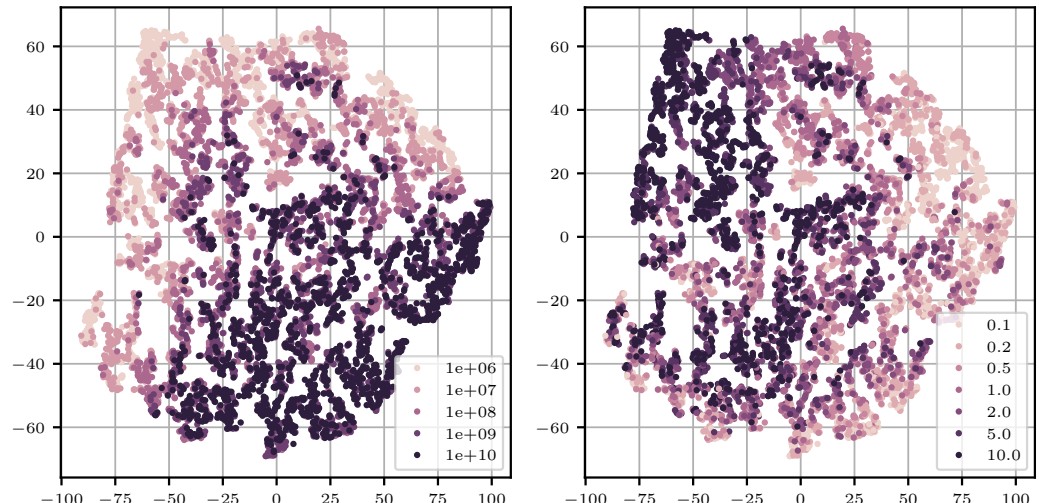

Figure 23. t-SNE [120] visualization of the latent states $z^i = E_\psi(x^i)$. The projections are colored with respect to their Rayleigh (left) and Prandtl (right) numbers.

Table 16. Bures-Wasserstein distance matrix between the distributions of latent states $z^i = E_\psi(x^i)$ with different Rayleigh numbers.

| | $10^6$ | $10^7$ | $10^8$ | $10^9$ | $10^{10}$ |
|---|---|---|---|---|---|
| $10^6$ | 0.000 | 1.045 | 1.708 | 2.279 | 2.489 |
| $10^7$ | 1.045 | 0.000 | 0.965 | 1.537 | 1.794 |
| $10^8$ | 1.708 | 0.965 | 0.000 | 0.915 | 1.180 |
| $10^9$ | 2.279 | 1.537 | 0.915 | 0.000 | 0.714 |
| $10^{10}$ | 2.489 | 1.794 | 1.180 | 0.714 | 0.000 |

Table 17. Bures-Wasserstein distance matrix between the distributions of latent states $z^i = E_\psi(x^i)$ with different Prandtl numbers.

| | 0.1 | 0.2 | 0.5 | 1.0 | 2.0 | 5.0 | 10.0 |
|---|---|---|---|---|---|---|---|
| 0.1 | 0.000 | 1.367 | 2.042 | 2.631 | 3.244 | 3.884 | 4.210 |
| 0.2 | 1.367 | 0.000 | 1.269 | 1.839 | 2.381 | 3.007 | 3.331 |
| 0.5 | 2.042 | 1.269 | 0.000 | 0.986 | 1.479 | 2.093 | 2.398 |
| 1.0 | 2.631 | 1.839 | 0.986 | 0.000 | 0.930 | 1.472 | 1.766 |
| 2.0 | 3.244 | 2.381 | 1.479 | 0.930 | 0.000 | 0.988 | 1.251 |
| 5.0 | 3.884 | 3.007 | 2.093 | 1.472 | 0.988 | 0.000 | 0.711 |
| 10.0 | 4.210 | 3.331 | 2.398 | 1.766 | 1.251 | 0.711 | 0.000 |

