# OpenReview forum: "Lost in Latent Space: An Empirical Study of Latent Diffusion Models for Physics Emulation"
_NeurIPS.cc/2025/Conference — NeurIPS 2025 poster_

### Official Review · Reviewer_mPpC · 2025-06-11

**Clarity:** 3
**Significance:** 3
**Originality:** 3
**Rating:** 4
**Confidence:** 2

**Summary:**

This paper investigates the potential of *latent diffusion models* for 2-dimensional fluids emulations. For the aforementioned problem setting, the authors demonstrate 1) latent space models match or exceed the performance of pixel (raw data) space models and 2) diffusion significantly improves emulation performance compared to deterministic point-regression neural solvers and 3) latent modelling brings considerable speedup at inference time. They also investigate the reconstruction ability of autoencoders for various compression rates.

**Questions:**

In addition to the weaknesses already mentioned (which affect my rating), I have a few questions that will not affect my rating, but I hope that the authors can clarify to me.

1. To what extent are such neural emulators actually useful for current applications? Can you provide an example of a concrete application that would benefit from the trade-off in accuracy for speed?
2. In the case of low-resolution conditioned generation, would it be possible to use a simulator to predict a spatially-coarse-grained but accurate sequence of states and then upscale it using a diffusion model?
3. How do neural emulators perform at "OOD" generalisation like on $\theta$ or $i$ not seen during training time? Do they collapse? Time extrapolation seems like an important application but the paper does not even consider it.

**Ethical Concerns:**

["NO or VERY MINOR ethics concerns only"]

**Final Justification:**

The authors clarified my concerns, however, as I am not familiar with the application (although I am familiar with the algorithms employed), I do not feel like I can give a firm recommendation. As such I will keep my score of weak accept.

**Limitations:**

Yes

**Quality:**

3

**Strengths And Weaknesses:**

**Note that I am familiar with diffusion but not physics emulation, so I will only assign low confidence to this review. I also welcome the authors to correct any mistakes/misunderstandings on my part that appear in this review.**

## Strengths
- The writing is clear, straightfoward and easy to follow/understand (for the most part).
- The takeaways seem to have clear impact in the field the paper is targeting. Various design choices are explored, leading ultimately to the advantages of the proposed latent diffusion approach being clearly presented. (Although I am personally not familiar with this application).
## Weaknesses
1. The paper lacks a clear intuition as to why diffusion is superior for this application (although perhaps it lies in the literature). Could the authors provide such an intuition?
2. This application seems very similar to video generation, especially considering the data is spatially 2D dimensional and the diffusion model is not trained with any "physics-informed" inductive biases. The authors would do better to link the performance of LDMs for video generation to this work. In fact, perhaps answers to why diffusion is effective for physics emulation can be found in the literature for image/video generation ([see this blog post](https://sander.ai/2024/09/02/spectral-autoregression.html]) and its accompanying references).
3. Although the authors present the result that the compression level of the autoencoder doesn't seem to affect diffusion model performance much, they don't suggest any potential reasons. Could this be due to e.g. *emulation* VRMSE being dominated by lower frequency information (larger blocks of "colour") which arises from the diffusion model and not the autoencoder? I would encourage the authors to provide at least an informed suggestion to provide some additional insight to the reader.
4. **Minor**:
    1. Compression rate would be better visualised using a colour map that changes in intensity according to the compression value. Figures 5/7 are difficult to interpret currently.
    2. The use of "well calibrated" in conjunction with "uncertainty" L249 is a bit confusing, since it refers typically to quite a specific concept in the uncertainty literature. I would suggest using different wording.
    3. Various equations would benefit from some underbraces for readability e.g. indicating which latents are noised and which are clean in Eq. 7.
    4. The use of eq 5. seems somewhat arbitrary (although the use of KL regularisation is similarly arbitrary in my experience). Was this based on eyeballing a histrogram of the element values of $z$? Did the authors ablate this choice? If I recall correctly DCAE doesn't use KL regularisation or this tanh-style squashing.
    4. Figures 1 and 2 are a little unclear. It would be better to clearly separate the emulation timesteps to better illustrate how the masking is applied, as well as which frames serve as conditioning. For example, in Figure 1 it is unclear whether $z^{5:8}$ is conditioned on  $z^{1:4}$ or just $z^{4}$. My understanding is that it is the latter, although I wouldn't be surprised if the former produced better results.

---

> ### Author Rebuttal · Authors · 2025-07-30
>
> Thank you for the pertinence of your review. Your feedback is very valuable, especially since your background gives us different perspectives on our work.
>
> **W1 & W3 (Insights)** Indeed, our empirical findings raise many questions, which prompt the need for theoretical explanations. We believe that this is precisely the goal of empirical research, and hence a strength of our work. Nevertheless, we agree that the lack of explanation could leave the reader puzzled. We provide two non-exclusive hypotheses that could explain our findings:
>
> 1. As explained in Section 5, many models, while accurate for short-term prediction, exhibit long-term instabilities as errors accumulate, pushing the predictions out of the training data distribution. Generative models, particularly diffusion models, partially address this problem [18, 19, 22–25] as they produce statistically plausible states, even when they diverge from the ground-truth solution.
>
>     However, distribution shift doesn't explain everything. At the first prediction step, before distribution shift can take effect, diffusion models are already slightly better than deterministic neural solvers. This could be explained by the iterative nature of DM sampling which allows the model to refine its prediction.
>
> 2. For many physics, and especially fluids, dynamics are mainly explained by large-scale/low-frequency content, while small-scale/high-frequency content influences the (very) long-term evolution of the system. Learning to process small-scale content and predict its influence on long-term dynamics might require very large networks, as well as huge amounts of data and compute. When the latter are limited, it becomes more tractable to focus on larger scales. Therefore, by discarding some of the small-scale content, latent diffusion models are able to learn and reproduce the dynamics more easily than pixel-space models.
>
> Validating these hypotheses is an interesting avenue for future research. We would appreciate your feedback on these hypotheses and whether we should include them in the manuscript.
>
> **W2 (Link to Video Generation)** There are definitely links with the image and video generation literatures. We already make connections in the discussion, especially regarding latent image generation which works much better than in pixel-space. Thank you for the blog post, we read it with attention. Note that we already refer to a blog post by Dieleman [56], although a different one. We will further look into the video diffusion model literature regarding the effects of compression.
>
> However, there is one very important difference between dynamical systems and videos. Dynamical systems are Markovian, meaning that knowing the current state is enough to predict the future (or the past). Therefore, there is no point in remembering the history, unlike for videos where it is very important.
>
> **W4.1 (Color map)** Thank you for your feedback. We will try to use a color map that changes in intensity.
>
> **W4.2 (Well Calibrated)** We are not sure to understand what you find confusing, but we are appreciate this opportunity to clarify our manuscript. Could you explain your perspective more precisely?
>
> From our perspective, the spread-skill ratio is a measure of calibration. Spread represents the uncertainty of the model. Skill represents the error of the model. For a well calibrated model, spread should be equal to skill (and the ratio be 1), meaning that the model should be as uncertain as it is wrong.
>
> **W4.3 (Equations Readability)** We will make sure to improve the readability of the equations in the camera-ready version.
>
> **W4.4 (Latent Saturation)** Yes, this is quite arbitrary. We tried a few saturating functions (which you can find in the code) in our preliminary experiments and all worked similarly. We used this one as it has a smooth derivative and saturates very slowly, which prevents vanishing gradients. $B = 5$ is chosen such that the values range between $-5$ and $5$, which is close to the range of a standard Gaussian. With these choices the mean and variance of the latents are close to 0 and 1 (empirically). Therefore, we do not need to re-scale the latents before training the diffusion model. Conversely, DCAE does not use this saturating function, but relies on rescaling the latents to train the latent diffusion model.
>
> **W4.5 (Figures)** Thank you again for your feedback. Indeed, these figures could be misleading. We will improve them in the camera-ready version, but cannot provide them in this rebuttal, as per NeurIPS guidelines.
>
> In most of our experiments, the latent emulators are only conditioned on a single past frame. However thanks to our masking strategy, we are able to test whether conditioning on several past frames would help. We already do so in Table 7, where we vary the number of context frames from 1 to 3. We find that context length has little to no impact on emulation accuracy with our latent models.
>
> This is likely due to dynamical systems being Markovian, as explained previously. In a Markov chain, knowing the current state is enough to predict the future. One could argue, however, that in the latent space, the system is not Markovian anymore as some information is lost. The Markov chain becomes a hidden Markov model (HMM). However, the fact that emulation accuracy is robust to compression and independent from context length, contradicts this argument.
>
> **Q1 (Applications)** There exists many applications for fast emulators. Common ones are weather forecasting and aerodynamic design (cars, airplanes, ...), for which traditional numerical solvers are extremely slow and expensive. Anything relying on simulations for real-time decision making too.
>
> We would like to note, however, that neural networks have other advantages than being fast emulators. They enable other tasks which are impossible with traditional numerical solvers. Notably, neural networks are differentiable, which can be very useful in the case of inverse problems. Diffusion-based emulators can also be guided with partial/noisy observations. Perhaps most importantly, data-driven emulators do not require knowledge of the underlying physics/PDEs, which is extremely valuable in real-world settings.
>
> **Q2 (Upscaling)** This is called "downscaling" in numerical weather forecasting. This generally doesn't work well because for most fluid systems, numerical solvers fail at low resolution. Therefore "spatially-coarse-grained but accurate" numerical simulations typically do not exist. Besides, upscaling with a diffusion model could lead to "good looking" states, but the additional high-frequencies would not be linked to the past states.
>
> **Q3 (Generalization)** Whether our models generate to out-of-distribution tasks is very relevant. We propose to add an experiment where the models are used to emulate with parameters $\theta$ that are not present in the training distribution. Our preliminary results suggest that the models generalize to new parameters $\theta$ when these are close to or in between parameters seen during training. However, they fail for completely new parameters, including new types of boundary conditions.
>
> Regarding time extrapolation, unrolling the trajectory past the training length ($i > L$) works quite well qualitatively, although we are not able to assess it quantitatively as we don't have access to data past this length.
>
> We hope that our answers address your concerns. If so, we kindly ask you to reconsider your score.

---

> > ### Comment · Reviewer_mPpC · 2025-08-03
> >
> > I thank the authors for their response and clarification, as well as for kindly answering my additional questions. I apologise for the late response as I have been travelling this weekend.
> >
> > > Validating these hypotheses is an interesting avenue for future research. We would appreciate your feedback on these hypotheses
> >
> > As discussed in the referenced blogpost, diffusion models generation low-frequency data components first, followed by high frequencies. Folding this idea into your suggested hypotheses could potentially work.
> >
> > > We are not sure to understand what you find confusing, but we are appreciate this opportunity to clarify our manuscript. Could you explain your perspective more precisely?
> >
> > I am used to the concept of "confidence calibration" in uncertainty estimation where the probability predicted for an event matched its empirical frequency. I appreciate the authors' clarification that there is an alternative use of the terminology. Perhaps a small footnote in the paper to clarify could help?
> >
> > Although I appreciate the authors' rebuttal, I think I will keep my score, mainly due to my personal lack of confidence judging research that is in an application outside of my area of expertise.

---

> > > ### Author Response · Authors · 2025-08-04
> > >
> > > Thank you very much for reading and answering to our rebuttal. Your feedback is very valuable and we will make sure to include it in the final manuscript.
> > >
> > > We appreciate your honesty regarding your lack of confidence. Nevertheless, given that we have addressed your concerns, we are genuinely surprised by your decision to keep your rating unchanged (4, borderline accept). We would greatly appreciate it if you could describe how to further improve our submission.
> > >
> > > We would like to note that increasing your rating would not affect your reported confidence (2).

---

### Official Review · Reviewer_v3JQ · 2025-06-16

**Clarity:** 4
**Significance:** 3
**Originality:** 3
**Rating:** 6
**Confidence:** 4

**Summary:**

The authors propose the use of latent diffusion models as emulators for physical systems governed by partial differential equations (PDEs). The primary objective of the paper is to investigate whether compression impacts the emulation capacity. The main finding is that the emulation quality remains high even in high-compression regimes, even when the autoencoder provides poor reconstructions. The authors evaluate their methods across several canonical PDES and provide qualitative and quantitative results.

**Questions:**

- Why did you not include comparisons to more established neural solvers such as Fourier Neural Operators (FNO), U-Nets, or Physics-Informed Neural Networks (PINNs)?

- It would be good to understand what physical properties are preserved in the latent space. For instance, are the latent variables learning specific quantities, such as velocity, pressure, etc., and are these preserved despite the poor reconstruction?

- Given that the decoder can hallucinate perceptually plausible outputs, how can we be sure that the generated fields are physically meaningful? Have you considered incorporating physics-based constraints or projection steps post-decoding? This also raises the question of how this model performs under different boundary conditions. Do I need to retrain the model, or does it generalize?

- Can you provide an intuitive explanation or a theoretical perspective of why the emulation remains strong despite poor reconstructions?

- Latent diffusion models can be very slow to train. Can you comment on the generalization capabilities of your model and how it would hold up in practice against more traditional alternatives?

- Your model generates statistically correct trajectories. In applications requiring deterministic repeatability (e.g. control), how would you address the stochastic nature of diffusion outputs?

**Ethical Concerns:**

["NO or VERY MINOR ethics concerns only"]

**Final Justification:**

I believe that the method is sound, and the authors in the rebuttal went beyond the original metrics to prove that the learned latents are meaningful. I recommend acceptance.

**Limitations:**

See questions and weaknesses.

**Quality:**

3

**Strengths And Weaknesses:**

### Strengths
- I believe that the experiments are well-motivated and extensive. In this regard, the paper's main finding, that good emulation does not require good reconstruction, is important, as it is, at least in my opinion, unexpected.
- The paper is well written and quite easy to understand, which is great. The numerical experiments/applications are interesting and validate the research question.
### Weaknesses
- I believe a major weakness of this approach is the reduced interpretability, as the method does not incorporate any explicit physical knowledge. This issue is crucial when dealing with safety-critical applications, for instance, in medical imaging, where certain applications may lead to physically invalid outputs.
- This is compounded with some missing baselines, for instance, https://arxiv.org/abs/2305.07671 proposes a PINN hybrid approach it would be good to compare against the physics-informed models counterparts.

---

> ### Author Rebuttal · Authors · 2025-07-29
>
> Thank you for the quality of your review. We greatly appreciate your interest in our findings. We are thankful for the opportunity to improve and clarify our manuscript.
>
> **W1 (Interpretability)** We believe that the lack of interpretability is a limitation of data-driven emulators, rather than a weakness of our work. Whether interpretability is necessary for emulators, is an open debate, which we believe is outside of the scope of this work. An example where interpretability is not necessary is weather forecasting, where speed and accuracy are much more relevant for consumers.
>
> **W2 & Q1 (Missing Baselines)** At the time of submission, no emulation results for the datasets of TheWell were available in the literature, other than the preliminary baselines from TheWell's paper [38]. Training our own competing baselines would be unreasonably expensive and lead to fairness concerns. Nevertheless, we agree that a more comprehensive set of baselines would help the reader assess the significance of our results within the context of data-driven emulators. Fortunately, in the past months, several studies have published results for TheWell datasets, which we compile here and will add to the manuscript.
>
> | Source | Method | Dataset | Parameters | lead time -> VRMSE |
> |:---|:---:|:---:|:---:|:---:|
> | Ohana et al. [38] | FNO | Euler | ~20M | 6:12 -> 1.13 |
> | Ohana et al. [38] | U-Net | Euler | ~20M | 6:12 -> 1.02 |
> | Ours | ViT | Euler | 860M | 1:20 -> 0.138 |
> | Ours | LDM $\div_{80}$ | Euler | 240M + 220M | 1:20 -> 0.075 |
> | Ohana et al. [38] | FNO | RB | ~20M | 6:12 -> 10+ |
> | Ohana et al. [38] | U-Net | RB | ~20M | 6:12 -> 10+ |
> | Nguyen et al. (2025) | PhysiX | RB | 4.5B | 2:8 -> 1.067 |
> | Wu et al. (2025) | COAST | RB | ? | 5 -> 0.282 |
> | Mukhopadhyay et al. (2025) | ViT + CSM | RB | 100M | 10 -> 0.140 |
> | Ours | ViT | RB | 860M | 1:20 -> 0.210 |
> | Ours | LDM $\div_{64}$ | RB | 240M + 220M | 1:20 -> 0.162 |
> | Ohana et al. [38] | FNO | TGC | ~20M | 6:12 -> 3.55 |
> | Ohana et al. [38] | U-Net | TGC | ~20M | 6:12 -> 7.14 |
> | Mukhopadhyay et al. (2025) | ViT + CKM | TGC | 100M | 10 -> 0.527 |
> | Ours | LDM $\div_{48}$ | TGC | 720M + 220M | 1:10 -> 0.297 |
>
> > Table A. Average VRMSE results from different studies using TheWell datasets. Even though our latent diffusion models (LDMs) outperform most published baselines, we emphasize that we are not making any claims regarding the novelty of our method and do not position our models as state-of-the-art, notably due to the discrepancies in parameters count, training and evaluation.
>
> Thank you for the PINN reference you provided, we will read it with attention. PINNs are very interesting and a developing area of research. However, we believe that it is unfair to compare PINNs with purely data-driven emulators, which do not require any knowledge of the physics.
>
> **Q2 & Q4 (Insights)** Indeed, our empirical findings raise many questions, which prompt the need for theoretical explanations. We believe that this is precisely the goal of empirical research, and hence a strength of our work. Nevertheless, we agree that the lack of explanation could leave the reader puzzled. We provide two non-exclusive hypotheses that could explain our findings:
>
> 1. As explained in Section 5, many models, while accurate for short-term prediction, exhibit long-term instabilities as errors accumulate, pushing the predictions out of the training data distribution. Generative models, particularly diffusion models, partially address this problem [18, 19, 22–25] as they produce statistically plausible states, even when they diverge from the ground-truth solution.
>
>     However, distribution shift doesn't explain everything. At the first prediction step, before distribution shift can take effect, diffusion models are already slightly better than deterministic neural solvers. This could be explained by the iterative nature of DM sampling which allows the model to refine its prediction.
>
> 2. For many physics, and especially fluids, dynamics are mainly explained by large-scale/low-frequency content, while small-scale/high-frequency content influences the (very) long-term evolution of the system. Learning to process small-scale content and predict its influence on long-term dynamics might require very large networks, as well as huge amounts of data and compute. When the latter are limited, it becomes more tractable to focus on larger scales. Therefore, by discarding some of the small-scale content, latent diffusion models are able to learn and reproduce the dynamics more easily than pixel-space models.
>
> Validating these hypotheses is an interesting avenue for future research. We would appreciate your feedback on these hypotheses and whether we should include them in the manuscript.
>
> **Q3 (Hallucinations)** There are two ways our models can "hallucinate": either the decoder introduces artifacts that are not plausible or the latent emulator generates implausible dynamics.
>
> 1. Our decoder is not trained with a perceptual loss. We therefore expect the decoding to have a "smoothing" behavior on the fields, which we observe in practice (high-frequency cutoff). Due to this degradation of small-scale features, some physicality is indeed lost.
>
>     To address this issue, we considered training a generative decoder during this project. However, this would not increase the amount of information compressed in the latent space. The additional small-scale details would be hallucinated by the decoder, without contributing to emulation accuracy, which would be misleading for humans. We never implemented this idea.
>
>     We did not consider introducing physics-based objectives during the training of the auto-encoder, as we assume that the physics are unknown. It is however an interesting research direction.
>
> 2. Our latent diffusion-based emulators are not perfect and often diverge from the ground-truth solution. In this light, it can be considered that individual trajectories are "hallucinated". However, our evaluation (using the spread-skill ratio) demonstrates that, as an ensemble, the trajectories are well calibrated, meaning that they generally cover plausible scenarios.
>
> **Q5 (Generalization)** You are perfectly right, we observed that (L)DMs need to be trained for a long time, and keep improving in the later training stages. Whether these models generate to out-of-distribution tasks is therefore very relevant. We propose to add an experiment where the models are used to emulate with parameters $\theta$ that are not present in the training distribution. Our preliminary results suggest that the models generalize to new parameters $\theta$ when these are close to or in between parameters seen during training. However, they fail for completely new parameters, including new types of boundary conditions.
>
> **Q6 (Repeatability)** Although diffusion models are generative models, there exist deterministic sampling strategies for them, which always lead to the same output when starting from the same initial noise. In fact, our sampling scheme (Adams-Bashforth, see Section 3.3) is one such strategy.
>
> We believe that this rebuttal addresses most of your concerns and, therefore, kindly ask you to reconsider your score.

---

> > ### Comment · Reviewer_v3JQ · 2025-08-02
> >
> > I thank the authors for the detailed rebuttal. I genuinely appreciate the potential and quality of the presented work. However, I rem unconvinced on several critical scientific grounds that are not adequately addressed in the current manuscript or rebuttal. My concerns focus on four core areas: physical fidelity, interpretability, high-frequency dynamics, and hallucination.
> >
> > #### 1. Physical Fidelity and Recovery of Governing Dynamics
> > In data-rich regimes, it is still fundamentally unclear whether the learned latent dynamics truly capture the correct underlying physical processes or merely reproduce statistically plausible behaviors. Without any constraints on the learning process this model may emulate trajectories that similar in appearance but corresponding to incorrect mechanics (see https://arxiv.org/abs/2507.06952) . Without some validation that the model is learning the underlying dynamics of the problem there is no guarantee that the model is recovering the true underlying PDE structure  a crucial distinction for scientific applications beyond black-box forecasting.
> >
> > #### 2. Interpretability and the Role of Physics-Informed Alternatives
> > While I acknowledge the merit of the model being uninformed about the mechanics of the problem. I don't think that a comparison to physics-informed methods is unfair, in fact, in my opinion is necessary. From a practical perspective, in the decision making context of choosing a model for an application if a physics informed model offers comparable accuracy with significantly more interpretability they are preferable especially in high-stakes settings.
> >
> > Given that the diffusion-based model is both computationally intensive and opaque, it must offer an advantage over interpretable alternatives. This threshold is not convincingly met with the current experiments.
> >
> > #### 3. Importance of Reconstruction Errors
> >
> > The authors correctly note that small-scale (high-frequency) content influences the long-term behavior of many physical systems. Yet they simultaneously justify the smoothing and discarding of these features to make learning "more tractable."
> > The proposed tradeoff is problematic in some settings for instance in turbulent flow or low-viscosity regimes, where the high frequency components are important to the correct evolution of the system.
> >
> > It is important to understand if the suppression of high-frequency content is due to the decoder or if the learned latents fail to incorporate this information because the model may fundamentally fail to capture key physical mechanisms such as energy cascades or dissipation dynamics.
> >
> > #### 4. Hallucination and Latent Misinterpretation of Physics
> >
> > The rebuttal acknowledges that the autoencoder and decoder may hallucinate features, and that individual trajectories may not match the ground truth. However, the argument that these hallucinations are acceptable because the ensemble is "well-calibrated" does not resolve the concern.
> >
> > In scientific modeling, the origin of a pattern matters. A flow field generated by the Euler equations may resemble one generated by compressible Navier-Stokes equations, but their physical interpretations (e.g., presence of viscosity, dissipation) are fundamentally different. If the model learns to associate patterns with incorrect physics due to compression or latent-space ambiguities, its outputs risk being physically misleading, even if statistically plausible.
> >
> > I stand with my original opinion of the paper I think it's a good work that could benefit from additional comparison to more traditional benchmarks and a deep study of the learned dynamics. I keep my score.

---

> > > ### Author Response · Authors · 2025-08-04
> > >
> > > Thank you for taking the time and effort to read and answer our rebuttal.
> > >
> > > **Physicality & Reconstruction** Thank you for the reference, it is a very interesting read. We agree that studying what equations are learned by data-driven emulators is an important direction of research, although it is tangent to our work. However, we fail to understand your concern regarding validation. If an emulator is **able to predict the evolution of a system accurately, it is reasonable to assume that it learned its physics**, even if it is not interpretable. The entire field of data-driven emulation rests on this assumption and emulators are generally evaluated with their emulation error (RMSE, VRMSE, ...). We follow this standard in our experiments and demonstrate that latent diffusion-based emulators are more accurate (in VRMSE) than pixel-space baselines, even when increasing compression.
> > >
> > > Obviously, our models, like any emulator (including PINNs), are not perfectly accurate and diverge gradually from the ground-truth. We believe that it is unfair to use this argument against our work as it applies to the entire field of data-driven emulation, and more generally the entire field of machine/deep learning.
> > >
> > > We understand your concern regarding the impact of the degradation of high-frequency content. Indeed, for turbulent flow regimes, high-frequencies are important for the long-term evolution of the system. However, in such chaotic system, it is intractable to achieve accurate long-term prediction as very small errors can lead to entirely different trajectories later on, even with highly accurate numerical simulations (Dryden, 1943; Hu et al., 2020). Instead, following common practice (Dryden, 1943), we evaluate whether our samples **statistically** match the reference. As can be seen in Tables 8 and 9, the frequency content is limited by the autoencoder's reconstruction capabilities.
> > >
> > > We further evaluate our models relative to the auto-encoded states $D_\psi(E_\psi(x^i))$ to demonstrate that the lack of high-frequency content is mainly due to the decoder, and not the latent emulation.
> > >
> > > | Model | Reference | 1:20 | 21:60 | 61:160 |
> > > | --- | --- | --- | --- | --- |
> > > | Diffusion $\div_{64}$ | $x^i$ | 0.064 | 0.208 | 0.202 |
> > > | Diffusion $\div_{64}$ | $D_\psi(E_\psi(x^i))$ | 0.035 | 0.093 | 0.099 |
> > > | Diffusion $\div_{256}$ | $x^i$ | 0.093 | 0.236 | 0.236 |
> > > | Diffusion $\div_{256}$ | $D_\psi(E_\psi(x^i))$ | 0.031 | 0.059 | 0.075 |
> > > | Diffusion $\div_{1024}$ | $x^i$ | 0.108 | 0.282 | 0.287 |
> > > | Diffusion $\div_{1024}$ | $D_\psi(E_\psi(x^i))$ | 0.027 | 0.069 | 0.095 |
> > >
> > > > Table D. Average power spectrum RMSE over the high-frequency band relative to $x^i$ and $D_\psi(E_\psi(x^i))$ of latent-space emulation at different compression rates ($\div$) for the Rayleigh-Bénard dataset.
> > >
> > > * Dryden, 1943. "A Review of the Statistical Theory of Turbulence"
> > > * Hu et al., 2020. "On the Risks of Using Double Precision in Numerical Simulations of Spatio-Temporal Chaos"
> > >
> > > **Interpretability** We agree that some applications may require interpretability and, in these cases, our method would not be applicable. However, we would like to point out that PINNs are not solid baselines for emulation, in addition to requiring knowledge of the physics. Many works demonstrate that data-driven methods (curriculum learning) significantly outperform PINNs in accuracy (error), training cost, and inference speed, while being more stable and easier to train.
> > >
> > > * Krishnapriyan et al., 2021. "Characterizing possible failure modes in physics-informed neural networks"
> > > * Chuang et al., 2022. "Experience report of physics-informed neural networks in fluid simulations: pitfalls and frustration"
> > > * Chuang et al., 2023. "Predictive Limitations of Physics-Informed Neural Networks in Vortex Shedding"
> > > * Grossmann et al., 2024. "Can physics-informed neural networks beat the finite element method?"
> > > * McGreivy et al., 2025. "My disappointing experience with PINNs"
> > >
> > > We note that our LDMs are on-par with pixel-space neural solvers in terms of inference speed, and significantly faster than the numerical solvers used to generate the data [38].
> > >
> > > **Hallucination** We apologize for the confusion regarding hallucinations. We borrowed the term "hallucinate" from your review as we believed we had a common definition of it. **Our models do not hallucinate** statistically plausible states that are not linked to the past states of the trajectory. Rather, each trajectory generated by our LDMs reproduces the dynamics of the system faithfully, but gradually diverges from the ground truth, as expected from any emulator. It is therefore possible to track the "origin of a pattern". In addition, the ensemble of trajectories provides a indicator of uncertainty of the model, which is extremely useful in forecasting tasks.
> > >
> > > We hope our answers address your remaining concerns. If not, we would greatly appreciate if you could describe how to further improve our submission.

---

> ### Comment · Reviewer_v3JQ · 2025-08-05
>
> I thank the authors for the discussion. My main point, as mentioned in the paper I cited, is that even if you have a good metric, the model may fail to recover the actual physical law, which is my primary concern. I believe the validation should be more thorough to demonstrate that the model accurately captures the physical laws.
>
> I propose running the model with this group of equations to obtain the proposed metrics and demonstrate that the latents of these equations (as a trajectory) are not close to each other, indicating that the encoder is not mixing physical phenomena.
>
> The equations I want tested are Euler, Incompressible NS under low and high viscosity, and Compressible NS with low and high $\mu$ (dynamic viscosity). Also, add unrelated equations such as KdV, Burgers, and Shallow Water with varying parameters. If you can show that the model latents can distinguish between parameters of the equation, I would be convinced and raise my score to a full score.

---

> ### Author Response · Authors · 2025-08-05
>
> Thank you for your time and effort during this rebuttal period. We appreciate your suggestion for an additional experiment. We would like to be sure to understand your concern and what you propose.
>
> Let $x_a^{1:L}$ and $x_b^{1:L}$ be two trajectories with different physical parameters $\theta_a$ and $\theta_b$ (e.g. low vs high viscosity), but starting at the same initial state $x_a^1 = x_b^1$. It seems that you are concerned that the latents $z_a^1 = E_\psi(x_a^1)$ and $z_b^1 = E_\psi(x_b^1)$ would be equal, which is correct. Without knowing $\theta_a$ and $\theta_b$, the emulator would therefore produce the same latent trajectories.
>
> However, in our models, **the emulator is conditioned on the parameters $\theta$** and can therefore produce different trajectories with different parameters $\theta_a$ and $\theta_b$ from a single initial state $z^1$.
>
> In this light, we are not sure how to conduct the experiment you suggest. Our framework is not compatible with the framework of the provided reference (Vafa et al., 2025), where the parameters (e.g. mass of planets) are considered to be part of the state.
>
> In addition, there is also a more practical issue with your suggested experiment. The systems you propose do not have the same state representations and (number of) parameters than Euler. We therefore cannot apply it without modifying the architecture of our models and retraining them, at which point we are not sure the experiment makes sense anymore.
>
> Could you please clarify what you expect from us? Would you be satisfied if we took a state $x^i$, and compared the latents $E_\psi(x^i)$ and $E_\psi(x^i + \Delta)$ to show that they are different? Or maybe you are interested in the latents produced by the latent emulator starting from the same initial state $z^1$, but with different parameters $\theta$?

---

> > ### Comment · Reviewer_v3JQ · 2025-08-05
> >
> > Thanks for the response. I understand that for the initial state, the representation is the same; what I want to see is the trajectory of $z$ for each of the equations I mentioned. If there is a clear separation (you can provide a UMAP or t-SNE visualization) of the proposed trajectories for each of the proposed equations (if you want to add other examples, I'm happy with that as well).
> >
> >  What I expect to see is that equations that belong to the same process, such as NS, are together and clearly distinct from the Burgers equation, for instance, and also I want to see the capability of the model to distinguish equations with varying parameters, such as changing the viscosity of the fluid and compressibility in NS.

---

> > > ### Author Response · Authors · 2025-08-05
> > >
> > > Thank you for helping us understand your request. We would like to point out that **it is not possible to provide visualizations** during the rebuttal period, as per NeurIPS guidelines. We can only provide results as tables. In addition, we do not have models trained for the different equations you request. We also don't have the data to train them, nor the numerical solvers to generate this data. Finally, even if we had the data, and resources to train these models, we would not have the time to before the end of this rebuttal.
> > >
> > > We propose the following experiments, which we can conduct in a reasonable time scale:
> > >
> > > 1. Starting from a common initial state $z^1$, we will emulate the Euler equations using our latent diffusion models with different heat capacity $\gamma$ values (this is the only parameter in our Euler dataset). We can then measure and report the pairwise distance (in the latent space) of each trajectories. We expect trajectories with similar heat capacity to be closer to each others.
> > > 2. The parameters $\theta$ of the Rayleigh-Bénard dataset are the Rayleigh and Prantl numbers. We can measure the distance (e.g. in terms of mean and variance) of the distribution of latents for different parameters combination.
> > >
> > > Would this be satisfying for you?

---

> > > > ### Comment · Reviewer_v3JQ · 2025-08-05
> > > >
> > > > That would be ok if, for the camera-ready paper, you include the visualizations. If you can report EMD as you did for the other reviewer, it would be great.

---

> > > > > ### Author Response · Authors · 2025-08-05
> > > > >
> > > > > Thank you for your flexibility. We will make sure to provide visualizations in the camera-ready version.
> > > > >
> > > > > Here are the results for the second experiment we suggested. We split the Rayleigh-Bénard dataset by the Rayleigh number of each trajectory. For each split, we compute the latent representations $z^i = E_\psi(x^i)$ of the $\div_{64}$ auto-encoder for randomly selected states $x^i$. We select one vector $z^i[8, 2] \in \mathbb{R}^{64}$ in each latent state $z^i \in \mathbb{R}^{16 \times 4 \times 64}$. We then compute the Bures-Wasserstein distance between the distribution of vectors of each split.
> > > > >
> > > > > | Rayleigh number | 1e+06 | 1e+07 | 1e+08 | 1e+09 | 1e+10 |
> > > > > |---|---|---|---|---|---|
> > > > > | **1e+06** | 0.000 | 0.687 | 1.099 | 1.520 | 1.764 |
> > > > > | **1e+07** | 0.687 | 0.000 | 0.722 | 1.121 | 1.345 |
> > > > > | **1e+08** | 1.099 | 0.722 | 0.000 | 0.726 | 0.968 |
> > > > > | **1e+09** | 1.520 | 1.121 | 0.726 | 0.000 | 0.678 |
> > > > > | **1e+10** | 1.764 | 1.345 | 0.968 | 0.678 | 0.000 |
> > > > >
> > > > > As can be observed, the distance between each split grows with the gap between Rayleigh numbers.
> > > > >
> > > > > We repeat the same experiment with the Prandtl number and observe the same pattern.
> > > > >
> > > > > | Prandtl number | 1e-01 | 2e-01 | 5e-01 | 1e+00 | 2e+00 | 5e+00 | 1e+01 |
> > > > > |---|---|---|---|---|---|---|---|
> > > > > | **1e-01** | 0.000 | 1.007 | 1.563 | 1.992 | 2.397 | 2.815 | 3.108 |
> > > > > | **2e-01** | 1.007 | 0.000 | 0.991 | 1.375 | 1.778 | 2.195 | 2.500 |
> > > > > | **5e-01** | 1.563 | 0.991 | 0.000 | 0.794 | 1.101 | 1.505 | 1.803 |
> > > > > | **1e+00** | 1.992 | 1.375 | 0.794 | 0.000 | 0.711 | 1.067 | 1.321 |
> > > > > | **2e+00** | 2.397 | 1.778 | 1.101 | 0.711 | 0.000 | 0.682 | 0.919 |
> > > > > | **5e+00** | 2.815 | 2.195 | 1.505 | 1.067 | 0.682 | 0.000 | 0.581 |
> > > > > | **1e+01** | 3.108 | 2.500 | 1.803 | 1.321 | 0.919 | 0.581 | 0.000 |
> > > > >
> > > > > We will provide the results of the first experiment we suggested as soon as possible.

---

> ### Author Response · Authors · 2025-08-06
>
> Thank you for your patience. Here are the results of the first experiment we suggested.
>
> We take one random trajectory $x^{1:L}$ from the test split of the Euler dataset. We compute the inital latent state $z^1 = E_\psi(x^1)$ for the $\div_{80}$ auto-encoder.
>
> Then, for each heat capacity $\gamma$ in $\{1.2, 1.3, 1.4, 1.5, 1.6\}$, we generate one latent trajectory $z^{1:L}$ with the diffusion-based emulator. Afterwards, we compute the distance $||z_a^i - z_b^i||$ for each pair of heat capacities $\gamma_a$ and $\gamma_b$. We report the results for $i \in \{4, 20, 100\}$. We expect the distance between latents to grow with the difference of heat capacity.
>
> We note that we cannot use the EMD in this case. We were able to use the EMD for reviewer's 52YY experiment as the pressure is a non-negative scalar field that we transformed into a density field. The latents are vectors taking real values.
>
> | $i = 4$ | 1.20 | 1.30 | 1.40 | 1.50 | 1.60 |
> |---|---|---|---|---|---|
> | **1.20** | 0.00 | 26.51 | 32.75 | 39.56 | 46.21 |
> | **1.30** | 26.51 | 0.00 | 14.97 | 23.06 | 31.66 |
> | **1.40** | 32.75 | 14.97 | 0.00 | 15.85 | 24.72 |
> | **1.50** | 39.56 | 23.06 | 15.85 | 0.00 | 17.26 |
> | **1.60** | 46.21 | 31.66 | 24.72 | 17.26 | 0.00 |
>
> | $i = 20$ | 1.20 | 1.30 | 1.40 | 1.50 | 1.60 |
> |---|---|---|---|---|---|
> | **1.20** | 0.00 | 57.98 | 65.19 | 72.28 | 78.16 |
> | **1.30** | 57.98 | 0.00 | 41.21 | 55.98 | 64.54 |
> | **1.40** | 65.19 | 41.21 | 0.00 | 43.44 | 59.10 |
> | **1.50** | 72.28 | 55.98 | 43.44 | 0.00 | 47.74 |
> | **1.60** | 78.16 | 64.54 | 59.10 | 47.74 | 0.00 |
>
> | $i = 100$ | 1.20 | 1.30 | 1.40 | 1.50 | 1.60 |
> |---|---|---|---|---|---|
> | **1.20** | 0.00 | 76.48 | 84.62 | 90.31 | 94.48 |
> | **1.30** | 76.48 | 0.00 | 68.17 | 75.35 | 81.86 |
> | **1.40** | 84.62 | 68.17 | 0.00 | 68.57 | 75.51 |
> | **1.50** | 90.31 | 75.35 | 68.57 | 0.00 | 70.30 |
> | **1.60** | 94.48 | 81.86 | 75.51 | 70.30 | 0.00 |
>
> We hope that these new results address your concerns. We remain available if you have any questions.

---

> > ### Comment · Reviewer_v3JQ · 2025-08-06
> >
> > Thanks for these results. I believe this addresses my main concern, as it appears that your model can distinguish between equations with different parameters, and they are encoded in the latent space. I will raise my score, trusting that you will add the visualization of these trajectories.

---

### Official Review · Reviewer_52YY · 2025-07-02

**Clarity:** 3
**Significance:** 3
**Originality:** 2
**Rating:** 6
**Confidence:** 4

**Summary:**

The paper studies how physical emulators can exploit a latent space representationn for achieving an effective speedup. For this, the authors focus on two fluiddynamics datasets and compare different autoencoders (with different compression factors) for latent space representation, as well as diffusion models and a traditional neural solver for the prediction. As a baseline, they consider a neural solver in pixel space. The main result of the paper is to show empirically that latent space representations are robust to compression (up to reasonable compression factors) and that diffusion models are effective in reducing the compounding error problem. This study suggest practitioners to adopt latent space representations and diffusion models as backbone for future research in learning-based physical emulators.

**Questions:**

1. Can the authors elaborate on why the two datasets presented should be sufficient to present a case for the general family of physics emulators? Or otherwise back up their claims with experiments on additional datasets?
2. Is the stride in the datasets introduced because of limitations of the methods during training? Is this something avoidable? Or can these models only work with sufficiently coarse time-steps?
3. Can the authors assess the prediction results with a downstream application metric? E.g., if one is analyzing an airfoil, they might be interested in the resulting lift and drag coefficients. For the datasets you considered, are the predictions accurate enough to be used in downstream applications? This is a much more important metric than VRMSE, power spectrum similarity, etc.
4. Why did the authors use 32x32 patches per token in the latent space model but 16x16 in the pixel space model?
5. Why did the authors introduce the binary mask? Can they argue about the benefits and necessity of training in a different scenario of what the downstream application will be? They write a sentence about this yielding benefits, did they run some ablation studies?
6. They mention the benefits for their sampling strategy: Can they provide evidence? Making these claims without evidence to back up may steer future research in directions not necessarily thoroughly tested.
7. Can the binary mask train the model for “inpainting”? i.e., filling in between a coarser time-scale?
8. Why did the authors not consider a pixel-space diffusion model? What would the performances be?

**Ethical Concerns:**

["NO or VERY MINOR ethics concerns only"]

**Final Justification:**

The authors thoroughly addressed all my concerns. The paper is an empirical study (very clear in the title). Considering it as such, it would be a great contribution to NeurIPS and it is very well developed.
I strongly recommend acceptance

**Limitations:**

In general, the paper discuss its limitations and does not overpromise, which I appreciate. There are however some points that need some extra careful discussion, in my opinion. I believe that by discussing more openly these limitations would position the paper more strongly.

1. The authors only consider two, 2D, fluiddynamics dataset (This point is mentioned by the author, but I think they should expand on it). They argue these are different, but fluiddynamics phenomena are also very rich. Since the study is empirical, providing sufficient empirical evidence is necessary for backing up the claims presented. Furthermore, the title of the work hints at physics emulation in general. The PDE governing fluiddynamics are general, but e.g. a primarily viscous fluid flow is excluded from the study.
2. The quality of the prediction is assessed in terms of a few, different numerical metrics which make sense but are hard to evaluate: What does 0.01 VRMSE mean for the simulation considered? The qualitative images shown seem promising, but it is also clear that the models cut off higher frequency components. Are these important in downstream applications? These questions remain unanswered in the paper, and it remains unclear for which types of applications (if not all) it is a good idea to directly focus on latent space prediction without worrying (too much) about compression.
3. Typically, physics simulator have tradeoff speed-accuracy. It is unclear in the paper where the physics emulator proposed are placing themselves in this axis. How do the performances compare with a lower accuracy, higher speed simulator? Is there a “knob” an user can tune in these physics emulators to reduce speed and improve the accuracy (and vice versa)? These questions remain open, leaving the study somewhat incomplete.
4. There is a vast literature on latent representations (which the authors rightfully cite). But the paper main contribution is an empirical study of latent space for physics emulation, so not discussing and studying thesed representations seem incomplete.

**Paper Formatting Concerns:**

The authors corrected me on the paper formatting, noting that appendices could be added to the main text.

**Quality:**

4

**Strengths And Weaknesses:**

The paper is very well written, focused on a single issue, and has extensive numerical comparisons. It tackles an important and timely problem, and tries to do it empirically but with rigor and only simplifications due to the computational requirements needed for such a task. Given the intensive computational effort, sharing these results could possibly benefit the community and sparing people to repeat the same ablation studies.

Although I believe the problem is important and I appreciate the focus of the paper, I believe there are also several weaknesses to be addressed:

1. Scope

- Dataset Diversity: Only two 2D fluid‐dynamics datasets are used, insufficient to support claims about physics emulators in general; key regimes (e.g., viscous flows, 3D phenomena) are omitted.
- Time-Resolution Constraints: The reason for temporal down‐sampling (striding) is unclear (whether it’s a model limitation or training convenience) and it’s unknown if finer time‐steps would break the approach. A thorough empirical validation should nail all these details.

2. Method

- Patch-Size discrepancy: Latent-space tokens use 32×32 patches, while pixel-space tokens use 16×16, with no rationale or ablation study provided.
- Binary-mask: Introduced without experiments backing up the claimed benefit.
- Sampling strategy claims: Benefits are asserted but unquantified, with no experiments to back up the sampling scheme’s impact.
- Omission of pixel-space Diffusion: Only latent-space diffusion is explored; a pixel-space diffusion model baseline is never evaluated.

3. Metrics

Lack of downstream-task metrics: The study relies on VRMSE and power-spectrum similarity without assessing application-relevant quantities (e.g., in airfoil, lift/drag), so practical adequacy is unknown.

4. Comparisons and baselines

Speed–accuracy trade‐off: No positioning of the emulator on the fidelity vs. runtime axis; no comparison to lower‐fidelity, faster simulators or a mechanism (“knob”) to adjust speed vs. accuracy. These are instead readily available in physics simulators.

5. Literature and context

Latent-representation survey: Despite citing latent-space literature, there is no critical engagement or experimental comparison of different representation strategies, leaving the empirical study feeling incomplete.

---

> ### Author Rebuttal · Authors · 2025-07-29
>
> Thank you for the quality of your review and interest in our work. We greatly appreciate that you read our manuscript with such attention to details. Your concerns are legitimate and we will address them here and in the manuscript.
>
> **Q1 & L1 (Dataset Diversity)** We follow your suggestion and add an additional 3D dataset to our study. This dataset (`turbulence_gravity_cooling` from TheWell [38]) models the interstellar medium as a turbulent fluid subject to gravity and radiative cooling. The 3d state of the system is discretized on a 64x64x64 grid. As for the other datasets, we train several autoencoders with increasing compression rates (from 48 to 768) and latent emulators. The results (see table below) are consistent with the previous experiments and strengthen our conclusions: latent diffusion emulators are robust to compression. Latent neural solvers, conversely, are not as robust.
>
> | Model | VRMSE 01:10 | VRMSE 11:30 | VRMSE 31:50 |
> |:----|:---:|:---:|:---:|
> | Diffusion $\div_{48}$ | 0.253 | 0.514 | 0.667 |
> | Diffusion $\div_{192}$ | 0.256 | 0.506 | 0.647 |
> | Diffusion $\div_{768}$ | 0.267 | 0.535 | 0.663 |
> | Neural solver $\div_{48}$ | 0.259 | 0.598 | 0.834 |
> | Neural solver $\div_{192}$ | 0.283 | 0.634 | 0.856 |
> | Neural solver $\div_{768}$ | 0.339 | 0.708 | 0.94 |
>
> > Table B. Average VRMSE of latent-space emulation at different compression rates (÷) relative to the auto-encoded states $D_\psi(E_\psi(x^i))$ for the TGC dataset.
>
> Unfortunately, we cannot provide plots during the rebuttal, as per NeurIPS guidelines. Further results will be provided in the camera-ready version.
>
> We would also like to mention that the Rayleigh-Bénard dataset models a **viscous** fluid.
>
> **Q2 (Time-Resolution Constraints)** We apologize for the confusion and thank you for pointing this out. The time stride was introduced strictly for convenience: with larger strides, emulation is faster. We conduct the new experiment (see above) without time stride ($\Delta = 1$). We also started to retrain our models with $\Delta = 1$. At the time of this rebuttal, only some Rayleigh-Bénard results are available.
>
> | Model | VRMSE 01:20 | VRMSE 21:60 | VRMSE 61:160 |
> |:----|:---:|:---:|:---:|
> | Diffusion $\div_{64}$ | 0.18 | 0.527 | 0.654 |
> | Diffusion $\div_{256}$ | 0.19 | 0.528 | 0.647 |
> | Diffusion $\div_{1024}$ | 0.22 | 0.536 | 0.652 |
>
> > Table C. Average VRMSE of latent-space emulation at different compression rates (÷) for the Rayleigh-Bénard dataset and with a time stride $\Delta = 1$. The results are very similar to $\Delta = 4$ with $\Delta = 1$, despite the larger ($\times 4$) number of rollout steps.
>
> **Q3 & L2 (Metrics)** (V)RMSE and power spectrum analysis are standard metrics in the data-driven emulation literature. The spread-skill ratio is also standard in Earth sciences. We are not aware of other widespread metrics compatible with the datasets we consider. We would appreciate if you could provide downstream metrics that we could compute in our datasets.
>
> Regarding the high-frequency content, this is definitely true. The use of a deterministic decoder for the auto-encoder leads to a lack of small-scale features. We further evaluate our latent models relative to the auto-encoded states $D_\psi(E_\psi(x^i))$ to demonstrate that this lack is mainly due to the decoder, and not the latent emulation.
>
> We see many applications for latent (compressed) emulation. In weather forecasting, as the lead time increases, large-scale trends and features become more relevant for meteorologist. In engineering/manufacturing, large deformations are typically more important than small ones. In astrophysics, you could care more about total light flux than flux variations over a star surface.
>
> **Q4 (Patch-size Discrepancy)** This discrepancy comes from our desire to give strictly more capacity to the pixel-space models than latent-space models. With 16x16 patches, the pixel-space model has more tokens ($\frac{HW}{16^2}$ vs $\frac{HW}{32^2}$) and more dimensions per token (2048 vs 1024). We also train the pixel-space models with double the compute and for up to 2 times longer. As such, one cannot argue in good faith that our assessment was unfair against the pixel-space baseline.
>
> We would also like to point out that ViTs typically perform better with smaller patches, as demonstrated by Wang et al. (2025) and Mukhopadhyay et al. (2025).
>
> - Wang et al. (2025). "Scaling Laws in Patchification: An Image Is Worth 50,176 Tokens And More"
> - Mukhopadhyay et al. (2025). "Controllable Patching for Compute-Adaptive Surrogate Modeling of Partial Differential Equations"
>
> **Q5 (Binary Mask)** The binary mask is necessary to perform multi-task conditioning. In our case, the tasks are forward and backward temporal prediction, with $c \geq 1$ context states. We vary $c$ in our experiments to determine whether context length has an impact on emulation accuracy. The results, in Table 7, indicate that it does not.
>
> Regarding the benefits, our claim comes from our preliminary experiments and several references that report similar benefits while using randomized/variable masking strategies:
>
> - Zheng et al. (2023). "Fast Training of Diffusion Models with Masked Transformers"
> - Voleti et al. (2022). "MCVD: Masked Conditional Video Diffusion for Prediction, Generation, and Interpolation"
> - Harvey et al. (2022). "Flexible Diffusion Modeling of Long Videos"
>
> **Q7 (Inpainting)** We do not train for this in this study, but masked conditioning can be used for inpainting. However, we note that this task can also be tackled **without any additional training** in the case of diffusion models. As mentioned in Section 4, it is possible to guide the model with partial observations, which includes spatial and temporal "inpainting". We will provide more examples of guidance (e.g. inpainting, super resolution) in the camera-ready version.
>
> **Q6 (Sampling Strategy)** We apologize for the confusion. The Adams-Bashfort sampling method is **not novel**. It was proposed and extensively tested by Zhang et al. [74]. This method is also equivalent to the widespread "linear multi-step" method proposed by Katherine Crowson in their `k-diffusion` repository. We believe that this sampler is sufficiently established that we don't need to evaluate it against others samplers (notably DDPM, DDIM and Heun).
>
> **Q8 (Pixel-space Diffusion)** We do not train pixel-space diffusion baselines as they are too expensive (tens of thousand of GPU hours) to train and evaluate at the resolution and scale of our datasets. We agree that this is a limitation of our work and will discuss it explicitly in the manuscript. We note that our code (which we provided with the submission and will be released publicly) already supports training a pixel-space diffusion model. With enough resources, a third party could easily train these models.
>
> **L3 (Speed/Accuracy Tradeoff)** Indeed we are not comparing our models against numerical solvers at different space and/or time resolutions. We agree that this would be a valuable addition to this study. Unfortunately, we do not have the expertise to implement these solvers ourselves. We will discuss this limitation in the manuscript.
>
> We would like to note, however, that neural networks have other advantages than being fast emulators. They enable other tasks which are impossible with traditional numerical solvers. Notably, neural networks are differentiable, which can be very useful in the case of inverse problems. Diffusion-based emulators can also be guided with partial/noisy observations. Perhaps most importantly, data-driven emulators do not require knowledge of the underlying physics/PDEs, which is extremely valuable in real-world settings.
>
> Regarding the "knob" to reduce speed and improve the accuracy, there isn't one currently. However, that is a very interesting research question. We believe this could be achieved with a tokenizer with a variable number of tokens such as FlexTok (Bachmann et al., 2025) as auto-encoder. Our methodology could then be applied on this "flexible latent space" and we would modify the computational cost by choosing the number of tokens.
>
> - Bachmann et al., 2025. "FlexTok: Resampling Images into 1D Token Sequences of Flexible Length"
>
> **L4 (Latent Representation)** As we explain in the discussion, a broader study of different latent representation strategies would be a valuable addition to this study. However, we believe that a comprehensive assessment of all latent representation methods is vastly outside of the scope of a modest academic conference paper and would require 10 to 100 times more time and compute. Therefore, in this work, we focus on the most widespread latent representation, that is the one learned by an auto-encoder.
>
> **Formatting** As per the paper formatting instructions: "The authors may optionally choose to include some or all of the technical appendices in the same PDF [...]"
>
> We hope that our answers and additional experiments address your concerns. If so, we kindly ask you to reconsider your score.

---

> > ### Comment · Reviewer_52YY · 2025-08-01
> > **Thanks for your rebuttal. Some more comments.**
> >
> > Thanks for the rebuttal and keeping into account my comments. I appreciate the answers and will certainly increase my score. I cannot inspect the revised text (I really think NeurIPS should go back to allowing it) but I assume you will integrate all these new results and discussions.
> >
> > Q3 & L2: You mention some in this answer: "In weather forecasting, as the lead time increases, large-scale trends and features become more relevant for meteorologist. In engineering/manufacturing, large deformations are typically more important than small ones. In astrophysics, you could care more about total light flux than flux variations over a star surface." If you can show that you can use your model for obtaining better performances in one of these tasks, you would provide a much more effective and indisputable metric of effectiveness. I understand this may too much to ask for a review of the paper for a "modest academic conference". But this would make your work extremely solid from the empirical point of view. I am very skeptical of "obscure" mathematical metrics for these kind of things. The downstream applications you mentioned, instead, would be very convincing.
> >
> > Q1-Q2, Q4-Q8: Thanks! Please include these clarifications in the revised text.
> >
> > Re: L3. Nice! Very cool stuff the proposed architecture. Someone should do it... ^^ I'd recommend to include it in your revised text, as outlook. For the solvers, I think you should have spent some time to learn to or involved someone able to: It could have been a very valuable addition to your work. Perhaps consider this for an extension. I know I am a bit direct but I hope you understand that I appreciate your work a lot and I would have really liked to see that as well, in particular to appeal and convince people that work across communities. :) I think a due paragraph in the revised text should address this limitation, and clarify that you focus only on comparisons between a "static" tradeoff. You may want to make sure that the neural solver and the diffusion solver have the same fixed time/computational budget: Do they?
> >
> > Re: L4. I agree with you, but I also like "complete" works. Maybe you can consider complementing the work with additional results in an accompanying website. Or as a journal version.
> >
> > I know L3, L4, Q3 & L2 require some work. I am increasing my score anyway given the other answers. I am somewhat insisting because your work focuses on this very clear question and takes the empirical pathway. The answers are very interesting, important, but somewhat leave space for (smaller and smaller) doubts. With some of these improvements, a third researcher may not have at all to explore on their own and just learn from your work.
> >
> > Progress on one of these directions (with priority on Q3 & L2) would be my requirement for an additional score increase (full score, 6). I am obviously willing to interact this week and give feedback on something that has a reasonable workload but would be convincing (at least for me).
> >
> > Once again, thanks for the good work.

---

> > > ### Author Response · Authors · 2025-08-04
> > >
> > > Thank you very much for reading and answering our rebuttal. We deeply appreciate your interest for our submission. We apologize for the delay of our answer, we took some more time to design and run evaluations with task-oriented metrics under your suggestion.
> > >
> > > Regarding L3 and L4, we will include additional discussion regarding these in the final version. We will also consider providing additional results in a later version, depending on funding.
> > >
> > > Now, regarding Q3 & L2. For each dataset (Euler, Rayleigh-Bénard and Turbulence Quadratic Cooling) we have designed a new metric that could be relevant for downstream applications, which we will include in the manuscript. We would like your feedback on these new metrics. We split our answer on two comments due to the character limit.
> > >
> > > **Euler** The Euler equations are sometimes used in aerodynamics to model flow around objects, like aircraft wings. In this case, the total pressure below and above the wing are what determines the lift/drag. We propose to use an Earth Mover's Distance (EMD) with respect to the ground-truth to quantify the accuracy of the pressure field as a whole, instead of point-wise as with the (V)RMSE. The rationale is that a small shift in the pressure field would not significantly modify lift/drag, while it could affect point-wise metrics heavily. Conversely, the EMD would be small in this case, as there isn't a lot of "dirt" (pressure) to move.
> > >
> > > | Models | 01:20 | 21:60 | 61:100 |
> > > |:---|:---:|:---:|:---:|
> > > | Autoencoder $\div_{80}$     | 0      | 0      |  0      |
> > > | Autoencoder $\div_{320}$    | 0.0001 | 0      |  0      |
> > > | Autoencoder $\div_{1280}$   | 0.0002 | 0.0001 |  0.0001 |
> > > | Diffusion $\div_{80}$       | 0.0006 | 0.001  |  0.0021 |
> > > | Diffusion $\div_{320}$      | 0.0005 | 0.0008 |  0.0018 |
> > > | Diffusion $\div_{1280}$     | 0.0005 | 0.0009 |  0.0019 |
> > > | Neural solver $\div_{1}$    | 0.0014 | 0.0027 |  0.005  |
> > > | Neural solver $\div_{80}$   | 0.0007 | 0.0012 |  0.0024 |
> > > | Neural solver $\div_{320}$  | 0.0006 | 0.0012 |  0.0024 |
> > > | Neural solver $\div_{1280}$ | 0.0012 | 0.002  |  0.0035 |
> > >
> > > > Table E. Average EMD of the pressure field of latent-space emulation at different compression rates (÷) for the Euler dataset.
> > >
> > > The EMD is quite small, especially for diffusion-based models. For reference, the pressure fields of two consecutive states $x^i$ and $x^{i+1}$ (which are very similar in the case of Euler) have an EMD close to $0.005$ on average.
> > >
> > > **Rayleigh-Bénard** One interesting quantity in buoyancy-driven convection is the growth speed of plumes in the fluid. The plumes are initially very slow, they accelerate gradually, and eventually form convection chambers with roughly constant velocity. The distribution of the (vertical) velocity values is a good summary statistic for tracking the growth of plumes. We propose to compute a Wasserstein distance between the distributions of vertical velocity values in the ground-truth and the emulated states. For turbulent fluids, comparing solutions statistically is common as solvers irremediably diverge in finite precision (Dryden, 1943).
> > >
> > > | Model | 01:20 | 21:60 | 61:160 |
> > > |:---|:---:|:---:|:---:|
> > > | Autoencoder $\div_{64}$     | 0      | 0.0002  |  0.002 |
> > > | Autoencoder $\div_{256}$    | 0.0002 | 0.0007 |  0.0005 |
> > > | Autoencoder $\div_{1024}$   | 0.0003 | 0.0019 |  0.0018 |
> > > | Diffusion $\div_{64}$       | 0.0003 | 0.009  |  0.0142 |
> > > | Diffusion $\div_{256}$      | 0.0002 | 0.0063 |  0.0140 |
> > > | Diffusion $\div_{1024}$     | 0.0004 | 0.0059 |  0.0136 |
> > > | Neural solver $\div_{1}$    | 0.0014 | 0.0264 |  0.0200 |
> > > | Neural solver $\div_{64}$   | 0.0010 | 0.0213 |  0.0203 |
> > > | Neural solver $\div_{256}$  | 0.0011 | 0.0215 |  0.0197 |
> > > | Neural solver $\div_{1024}$ | 0.0011 | 0.0210 |  0.0222 |
> > >
> > > > Table E. Average Wasserstein distance of the distribution of vertical velocity values of latent-space emulation at different compression rates (÷) for the Rayleigh-Bénard dataset.
> > >
> > > For reference, the distributions of vertical velocity values of two consecutive states $x^i$ and $x^{i+1}$ have a Wasserstein distance close to $0.005$ on average ($0.02$ for $x^i$ and $x^{i+5}$).

---

> > > > ### Author Response · Authors · 2025-08-04
> > > >
> > > > **Turbulence Gravity Cooling** In the interstellar medium, gravity forms clusters of matter that eventually lead to the birst of stars. The kind of clusters (compact, diffuse, or anything in between) and their proportions is of interest for domain-scientists. We propose to mesure the accuracy of the clustering dynamics with the Wasserstein distance between the distributions of density values in the ground-truth and emulated states.
> > > >
> > > > | Model |   01:10 |   11:30 |   31:50 |
> > > > |:---|:---:|:---:|:---:|
> > > > | Autoencoder $\div_{48}$    | 0.0033 | 0.0047 | 0.0089 |
> > > > | Autoencoder $\div_{192}$   | 0.0082 | 0.011  | 0.0183 |
> > > > | Autoencoder $\div_{768}$   | 0.0182 | 0.0237 | 0.0339 |
> > > > | Diffusion $\div_{48}$      | 0.0047 | 0.0131 | 0.024  |
> > > > | Diffusion $\div_{192}$     | 0.0091 | 0.0169 | 0.0291 |
> > > > | Diffusion $\div_{768}$     | 0.0188 | 0.028  | 0.0435 |
> > > > | Neural solver $\div_{48}$  | 0.0089 | 0.0266 | 0.0522 |
> > > > | Neural solver $\div_{192}$ | 0.0106 | 0.0166 | 0.032  |
> > > > | Neural solver $\div_{768}$ | 0.0233 | 0.0303 | 0.0499 |
> > > >
> > > > For reference, the distributions of density values of two consecutive states $x^i$ and $x^{i+1}$ have a Wasserstein distance close to $0.01$ on average ($0.06$ for $x^i$ and $x^{i+5}$).
> > > >
> > > > Thank you again for your interest and valuable feedback.

---

> > > > > ### Comment · Reviewer_52YY · 2025-08-04
> > > > > **Thanks for the work, I will increase my score.**
> > > > >
> > > > > Thank you for your work. These results are very convincing and I would strongly recommend adding them to the paper.
> > > > > Please discuss these results in the paper, the evident trends and the implications. I think these are very valuable compared to the other numerics.
> > > > > I will increase my score to 6.

---

### Official Review · Reviewer_pmvt · 2025-07-03

**Clarity:** 2
**Significance:** 2
**Originality:** 1
**Rating:** 2
**Confidence:** 4

**Summary:**

This paper compares deterministic neural solvers with latent diffusion models (LDM) for solving partial differential equations (PDEs). Specifically, it compresses 2D image-grid physical simulation data into a latent space using an autoencoder, then reconstructs it through a diffusion model operating within this latent space. The study evaluates performance across varying compression rates and claims that LDM demonstrates robust performance and computational efficiency even at high compression rates.

**Questions:**

Please see the above weaknesses.

**Ethical Concerns:**

["NO or VERY MINOR ethics concerns only"]

**Final Justification:**

After the author rebuttal and discussions, I do agree that the paper provides a thorough empirical analysis of LDMs for physics emulation. My concern was that its scope is narrowly confined to LDMs, despite there are many generative models with competitive potential in this domain (e.g., GAN-based surrogates [1], VAE-based architectures [2], normalizing flows [3], score-based generative models [4][5], etc). This is why I thought the paper limits the generalizability of its findings, and asked for additional experiments with more generative models beyond LDMs. Even though the authors said "*We do not train pixel-space diffusion baselines as they are too expensive...*" but I believe this is not true because, as I wrote in the response, LDMs can be applied in a latent space obtained via an AutoEncoder, rather than pixel-space, which naturally means that other generative models like VAE, GAN and Normalizing Flow could also have been applied in that space.

However, I truly agree that this paper delivers nice work within the boundary of LDMs for physics emulation. While I will not change my original rating, I can agree with acceptance if AC and the other reviewers recommend this paper.

[1] Kastner, Patrick, and Timur Dogan. "A GAN-based surrogate model for instantaneous urban wind flow prediction." Building and Environment 242 (2023).

[2] Clavier, B., et al. "Generative-machine-learning surrogate model of plasma turbulence." Physical Review E (2025).

[3] Yang, Minglei, et al. "Conditional pseudo-reversible normalizing flow for surrogate modeling in quantifying uncertainty propagation." Journal of Machine Learning for Modeling and Computing 6.4 (2025).

[4] Lippe, Phillip, et al. "Pde-refiner: Achieving accurate long rollouts with neural pde solvers." NeurIPS (2023).

[5] Rühling Cachay, Salva, et al. "Dyffusion: A dynamics-informed diffusion model for spatiotemporal forecasting." NeurIPS (2023).

**Limitations:**

Yes

**Quality:**

3

**Strengths And Weaknesses:**

**Strengths**
- Recently, there has been active research applying generative models, especially latent diffusion models, in the field of physics simulation. This paper aligns with that trend by conducting an empirical comparison.

- The systematic experimentation with varying compression rates is meaningful in terms of practical relevance.

- Addressing the trade-off between efficiency and compression through latent space is an important consideration for large-scale simulations.

- The attempt to empirically summarize the potential of LDM-based PDE simulations can be viewed positively.

**Weaknesses**
- Diffusion-based generative models have already been applied to PDE problems in the literature, so the novelty of this paper is very limited.

- Baseline comparisons are highly restricted; additional comparisons with various state-of-the-art methods beyond deterministic neural solvers are needed.

- The fundamental reasons why LDM exhibits superior performance over deterministic solvers across various compression rates are inadequately analyzed.

- It is unclear what practical impact robustness across different compression rates would have in actual simulation applications.

- The proposed method should be tested beyond simple 2D grid-based experiments, such as on irregular meshes or 3D simulations.

- Further validation might be necessary to confirm whether the efficiency advantages can extend to real-world, large-scale problems, and discussion on this aspect is currently lacking.

---

> ### Author Rebuttal · Authors · 2025-07-29
>
> Thank you for your review and the legitimate concerns you have raised. We are thankful for the opportunity to improve and clarify our manuscript.
>
> **W1 (Novelty)** We would like to emphasize that the main takeaway of our work is that **the accuracy of latent diffusion-based emulation is robust to a wide range of compression rates**. This result is unexpected and has practical relevance, as you acknowledge in your review. We are not making any claims regarding the novelty of our method and do not position our models as state-of-the-art.
>
> **W2 (Baselines)** At the time of submission, no emulation results for the datasets of TheWell were available in the literature, other than the preliminary baselines from TheWell's paper [38]. Training our own competing baselines would be unreasonably expensive and lead to fairness concerns. Nevertheless, we agree that a more comprehensive set of baselines would help the reader assess the significance of our results within the context of data-driven emulators. Fortunately, in the past months, several studies have published results for TheWell datasets, which we compile here and will add to the manuscript.
>
> | Source | Method | Dataset | Parameters | lead time -> VRMSE |
> |:---|:---:|:---:|:---:|:---:|
> | Ohana et al. [38] | FNO | Euler | ~20M | 6:12 -> 1.13 |
> | Ohana et al. [38] | U-Net | Euler | ~20M | 6:12 -> 1.02 |
> | Ours | ViT | Euler | 860M | 1:20 -> 0.138 |
> | Ours | LDM $\div_{80}$ | Euler | 240M + 220M | 1:20 -> 0.075 |
> | Ohana et al. [38] | FNO | RB | ~20M | 6:12 -> 10+ |
> | Ohana et al. [38] | U-Net | RB | ~20M | 6:12 -> 10+ |
> | Nguyen et al. (2025) | PhysiX | RB | 4.5B | 2:8 -> 1.067 |
> | Wu et al. (2025) | COAST | RB | ? | 5 -> 0.282 |
> | Mukhopadhyay et al. (2025) | ViT + CSM | RB | 100M | 10 -> 0.140 |
> | Ours | ViT | RB | 860M | 1:20 -> 0.210 |
> | Ours | LDM $\div_{64}$ | RB | 240M + 220M | 1:20 -> 0.162 |
> | Ohana et al. [38] | FNO | TGC | ~20M | 6:12 -> 3.55 |
> | Ohana et al. [38] | U-Net | TGC | ~20M | 6:12 -> 7.14 |
> | Mukhopadhyay et al. (2025) | ViT + CKM | TGC | 100M | 10 -> 0.527 |
> | Ours | LDM $\div_{48}$ | TGC | 720M + 220M | 1:10 -> 0.297 |
>
> > Table A. Average VRMSE results from different studies using TheWell datasets. Even though our latent diffusion models (LDMs) outperform most published baselines, we emphasize that we are not making any claims regarding the novelty of our method and do not position our models as state-of-the-art, notably due to the discrepancies in parameters count, training and evaluation.
>
> **W3 (Explanations)** Indeed, our empirical findings raise many questions, which prompt the need for theoretical explanations. We believe that this is precisely the goal of empirical research, and hence a strength of our work. Nevertheless, we agree that the lack of explanation could leave the reader puzzled. We provide two non-exclusive hypotheses that could explain our findings:
>
> 1. As explained in Section 5, many models, while accurate for short-term prediction, exhibit long-term instabilities as errors accumulate, pushing the predictions out of the training data distribution. Generative models, particularly diffusion models, partially address this problem [18, 19, 22–25] as they produce statistically plausible states, even when they diverge from the ground-truth solution.
>
>     However, distribution shift doesn't explain everything. At the first prediction step, before distribution shift can take effect, diffusion models are already slightly better than deterministic neural solvers. This could be explained by the iterative nature of DM sampling which allows the model to refine its prediction. Validating this hypothesis is an interesting avenue for future research.
>
> 2. For many physics, and especially fluids, dynamics are mainly explained by large-scale/low-frequency content, while small-scale/high-frequency content influences the (very) long-term evolution of the system. Learning to process small-scale content and predict its influence on long-term dynamics might require very large networks, as well as huge amounts of data and compute. When the latter are limited, it becomes more tractable to focus on larger scales. Therefore, by discarding some of the small-scale content, latent diffusion models are able to learn and reproduce the dynamics more easily than pixel-space models.
>
> We would appreciate your feedback on these hypotheses and whether we should include them in the manuscript.
>
> **W4 (Applications)** We see several use cases for using (large) compression in simulation applications. First, some fields (weather, astro) are collecting a lot more data that we can currently process or store. Our findings suggest that we could store or transfer this data in a compressed form, without losing (too much) information/predictability. Second, robustness to compression also means that practitioners do not have to worry (too much) about the compression rate they use: doubling or halving the compression rate will lead to similar emulation accuracy.
>
> **W5 (Dataset Diversity)** We follow your suggestion and add an additional 3D dataset to our study. This dataset (`turbulence_gravity_cooling` from TheWell [38]) models the interstellar medium as a turbulent fluid subject to gravity and radiative cooling. The 3d state of the system is discretized on a 64x64x64 grid. As for the other datasets, we train several auto-encoders with increasing compression rates (from 48 to 768) and latent emulators. The results (see table below) are consistent with the previous experiments and strengthen our conclusions: latent diffusion emulators are robust to compression. Latent neural solvers, conversely, are not as robust.
>
> | Model | VRMSE 01:10 | VRMSE 11:30 | VRMSE 31:50 |
> |:----|:---:|:---:|:---:|
> | Diffusion $\div_{48}$ | 0.253 | 0.514 | 0.667 |
> | Diffusion $\div_{192}$ | 0.256 | 0.506 | 0.647 |
> | Diffusion $\div_{768}$ | 0.267 | 0.535 | 0.663 |
> | Neural solver $\div_{48}$ | 0.259 | 0.598 | 0.834 |
> | Neural solver $\div_{192}$ | 0.283 | 0.634 | 0.856 |
> | Neural solver $\div_{768}$ | 0.339 | 0.708 | 0.94 |
>
> > Table B. Average VRMSE of latent-space emulation at different compression rates (÷) relative to the auto-encoded states $D_\psi(E_\psi(x^i))$ for the TGC dataset.
>
> Unfortunately, we cannot provide plots during the rebuttal, as per NeurIPS guidelines. Further results will be provided in the camera-ready version.
>
> **W6 (Real-World Problems)** Euler and Rayleigh-Bénard are some of the most challenging physics used in the field of data-driven emulators at the moment. They are commonly used for verifying/benchmarking numerical solvers and demonstrate behaviors often found in nature. The new 3D dataset (TGC) also has real-world applications and was originally generated for a domain-science publication by Hirashima et al. (2023).
>
> - Hirashima et al., 2023. "3D-Spatiotemporal forecasting the expansion of supernova shells using deep learning towards high-resolution galaxy simulations"
>
> We believe that this rebuttal addresses most of your concerns and, therefore, kindly ask you to reconsider your score.

---

> > ### Comment · Reviewer_pmvt · 2025-08-04
> >
> > - Novelty: I understand this work's main takeaway; but I think the paper should go beyond the current empirical findings and justifications. The paper should have provided a generalization of the observed phenomena and theoretical justification as I wrote earlier.
> >
> > - Baseline: My original request was for experiments to be conducted using various generative models, not just LDMs. Through such comparative analysis, the paper could explore questions such as whether only LDM exhibits superior performance and, if so, the underlying reasons for this unique advantage. Including such analyses would enhance understanding the significance of the observed phenomenon, providing researchers with valuable insights.
> >
> > - Explanations: It would be beneficial if objective evidences, such as supporting references, could be provided to validate the reasonableness of the suggested hypotheses. Alternatively, building a small-scale synthetic dataset to experimentally test and confirm these hypotheses could also be helpful.
> >
> > Overall, while some of my concerns have been partially addressed, I still believe further validation is required regarding the generalizability of the findings, the theoretical justification for why these outcomes occurred, and their practical implications. Addressing these issues, the revised paper would require another round of review.

---

> ### Author Response · Authors · 2025-08-04
>
> Thank you for answering our rebuttal.
>
> > Novelty: I understand this work's main takeaway; but I think the paper should go beyond the current empirical findings and justifications. The paper should have provided a generalization of the observed phenomena and theoretical justification as I wrote earlier.
>
> > Explanations: It would be beneficial if objective evidences, such as supporting references, could be provided to validate the reasonableness of the suggested hypotheses. Alternatively, building a small-scale synthetic dataset to experimentally test and confirm these hypotheses could also be helpful.
>
> Our empirical findings are novel and valuable, as you acknowledge yourself in your review. The goal of empirical research, which the field of deep learning is largely driven by, is to raise questions, not to find theoretical justifications.
>
> In addition, your original review asks for additional datasets and justifications why LDMS outperform deterministic solvers. We followed your suggestions and provide an additional 3D dataset as well as two hypotheses grounded in previous peer-reviewed research [18, 19, 22–25, 26, 56, 102]. In addition, there is an extensive literature on reduced-order modeling (ROM) [85-99] that provides theoretical justifications for modeling physics within reduced / compressed representations. This literature is already discussed in Section 5 (Related work).
>
> We also believe that it is unfair to ask for other toy experiments that late in the review process.
>
> > Baseline: My original request was for experiments to be conducted using various generative models, not just LDMs. Through such comparative analysis, the paper could explore questions such as whether only LDM exhibits superior performance and, if so, the underlying reasons for this unique advantage. Including such analyses would enhance understanding the significance of the observed phenomenon, providing researchers with valuable insights.
>
> We do not train pixel-space diffusion baselines as they are too expensive (tens of thousand of GPU hours) to train and evaluate at the resolution and scale of our datasets. Training our own baselines would also lead to fairness concerns. We believe that the additional baselines provided in the rebuttal are more than enough to demonstrate that our results are significant. In addition, baselines are not relevant to answer our research question: we are only interested in the accuracy of LDMs when compression increases. We note that our code (which we provided with the submission and will be released publicly) already supports training a pixel-space diffusion model. With enough resources, a third party could easily train these models.
>
> Finally, we would like to note that your only remaining concerns are (a) a lack of theoretical justifications for empirical findings (even though we provide hypotheses) and (b) some missing baselines. We believe that it is unreasonable to recommend "reject" on this basis. Our submission does not have technical flaws, the evaluation is sound and extensive with very challenging datasets, and we provide the code for reproducibility. We also think that our work deserves more than "1: poor" for its originality as our findings are novel and valuable.

---

> > ### Comment · Reviewer_pmvt · 2025-08-08
> >
> > Dear Authors,
> >
> > The additional experiments with more baseline generative models that I have been repeatedly requesting are important because they can clarify how well your findings generalize to other generative models. If the discussion is limited to LDMs alone, I think the significance can be seen as limited.
> >
> > I understand that this study uses very large datasets and that running new experiments during the rebuttal period is not feasible. However this does not indicate that my concern has been resolved. There would be some ways, such as using a different dataset like [1], which would allow experiments to be run much faster (within about four hours on a GPU with specifications similar to an RTX 3090). In addition, LDMs can be applied in a latent space obtained via an AutoEncoder, rather than pixel-space, which naturally means that other generative models like VAE, GAN and Normalizing Flow could also have been applied in that space. Do their performances also change w.r.t. the compression rate? I think answering such question is important. Actually it is not surprising that generative models outperform deterministic models, therefore I consider comparative experiments within generative models themselves essential in this context, and I believe there is no fairness concern here.
> >
> > However, the rebuttal did not provide such evidence, so my question (*if LDM is robust to compression rate are other generative models also robust? Is only the LDM robust while other generative models are not? If so why? etc* ) remains unresolved. LDM is not the only generative model used in the field; other generative models are also actively studied (e.g., [2] compared Diffusion models and VAEs). As in [2], the authors should have conducted similar experiments using more generative models like VAEs (i.e., feeding the latent data obtained from AutoEncoder into the VAE) to determine whether such robustness comes from being a generative model in general, or specifically from being LDM. I believe this is essential for completeness of the research. If such experiments have not been conducted, it would have been desirable to provide some theoretical justification that could help readers assess the generalizability, but this was also not done.
> >
> > [1] Takamoto, Makoto, et al. "Pdebench: An extensive benchmark for scientific machine learning." Advances in Neural Information Processing Systems 35 (2022): 1596-1611.
> >
> > [2] Kohl, Georg, Li-Wei Chen, and Nils Thuerey. "Benchmarking autoregressive conditional diffusion models for turbulent flow simulation." arXiv preprint arXiv:2309.01745 (2023).

---

> ### Author Response · Authors · 2025-08-08
>
> Thank you for taking the time to answer. We agree that applying our methodology to other generative models is interesting and valuable. However, this is outside of the scope of our work. As our title indicates, our study focuses on latent diffusion models and our findings do not need a comparison against other generative models to be valuable and relevant. The experiments your suggest would not modify our findings but provide new findings.
>
> In addition, (latent) diffusion models are state-of-the-art generative models and practitioners are actively using them for (latent) physics emulation, as explained in our introduction. Conversely, GANs are extremely finicky to train and VAEs are poor generative models. We are not aware of any research using VAEs or GANs in the field of physics emulation. **Khol et al. [2] are not comparing VAEs with DMs.** They compare DMs with a transformer trained to emulate in the latent space of a VAE. The VAE plays the same role as our AEs, not as the emulator.
>
> Therefore, we respectfully disagree with your opinion that our work is not significant and original. The very fact that our work raises so many questions, which you formulate in your own review, demonstrates its significance. Even with limited significance, your recommendation (reject) is unjustified.

---

### Note · Authors · 2025-08-12

We thank all reviewers for their engagement during the rebuttal phase. We deeply appreciate the enthusiasm of reviewers **52YY**, **v3JQ** and **mPpC** for our submission and we will make sure to include all of the clarifications and additional results in the camera-ready version. We hope that reviewer **pmvt** can acknowledge the worth of our submission, the scope of which we intentionally limit, in order to conduct clear and extensive experiments.

---

### Decision · Program_Chairs · 2025-09-17

**Decision:**

Accept (poster)

**Comment:**

The paper introduces an empirical study of the effect of the latent space compression of the efficiency of physical emulators. The main finding is that the emulation quality remains high even in high-compression regimes.

Initially, the reviewers appreciated the topic of the paper, clear writing and found the ideas presented in the paper very relevant to the research community. At the same time, several important concerns were brought up that included validation breadth, paper positioning and asked a fair amount of clarification question. With the rebuttal and post rebuttal discussion the authors were able to address almost all of the reviewer’s concerns. The only undressed concern is about the scope of the empirical study (validation breadth). Reviewer pmvt considers the validation of the observations with one generative model (LDM) too narrow and recommends to reject. Reviewers 52YY and v3JQ, agree with the argument of limited validation however, they argue that point out that the study is already very complete and of interest for NeurIPS community. AC would like to point out that the paper is well presented and reduces the scope of model exploration very early on — in the title, as such the AC agrees with the positive reviewers and thus recommends to accept. The authors are kindly asked to include the additions from the discussion and rebuttal in the camera ready (including the promised visualizations).